# Convergence Rates for Gradient Descent on the Edge of Stability for Overparametrised Least Squares

**Lachlan Ewen MacDonald**    **Hancheng Min**[*]   **Leandro Palma**[*]   **Salma Tarmoun**[*]
**Ziqing Xu**[*]   **René Vidal**
Innovation in Data Engineering and Science (IDEAS)
University of Pennsylvania
Pennsylvania, PA 19104
lemacdonald@protonmail.com

## Abstract

Classical optimisation theory guarantees monotonic objective decrease for gradient descent (GD) when employed in a small step size, or "stable", regime. In contrast, gradient descent on neural networks is frequently performed in a large step size regime called the "edge of stability", in which the objective decreases non-monotonically with an observed implicit bias towards flat minima. In this paper, we take a step toward quantifying this phenomenon by providing convergence rates for gradient descent with large learning rates in an overparametrised least squares setting. The key insight behind our analysis is that, as a consequence of overparametrisation, the set of global minimisers forms a Riemannian manifold $M$, which enables the decomposition of the GD dynamics into components parallel and orthogonal to $M$. The parallel component corresponds to Riemannian gradient descent on the objective sharpness, while the orthogonal component is a bifurcating dynamical system. This insight allows us to derive convergence rates in three regimes characterised by the learning rate size: (a) the subcritical regime, in which transient instability is overcome in finite time before linear convergence to a suboptimally flat global minimum; (b) the critical regime, in which instability persists for all time with a power-law convergence toward the optimally flat global minimum; and (c) the supercritical regime, in which instability persists for all time with linear convergence to an orbit of period two centred on the optimally flat global minimum.

## 1 Introduction

The well-known gradient descent (GD) iteration

$$\theta \mapsto \mathrm{GD}^\ell(\eta, \theta) := \theta - \eta \nabla \ell(\theta), \qquad \theta \in \mathbb{R}^p \tag{1}$$

for minimisation of an objective function $\ell : \mathbb{R}^p \to \mathbb{R}$, with learning rate $\eta > 0$, is the foundation of most practical algorithms for training deep neural networks (DNNs). Despite the apparent simplicity of the algorithm and *practical* ease with which DNNs can be trained using GD, *theoretically* GD on DNNs remains poorly understood. In addition to the well-known non-convexity of DNN loss functions $\ell$, our theoretical understanding is hampered by the fact that in practice DNNs tend to be trained with larger learning rates than permissible according to standard optimisation theory.

Specifically, classical optimisation theory requires that the learning rate $\eta$ be less than twice the reciprocal of the largest eigenvalue $\lambda$ (the *sharpness*) of the Hessian $\nabla^2 \ell$. The reason for this is easily

---

[*]Equal contribution, alphabetical order

39th Conference on Neural Information Processing Systems (NeurIPS 2025).

seen by considering the simplest quadratic objective $\ell(\theta) := \frac{1}{2}\lambda\theta^2$ of a single variable $\theta$: if $\eta > 2/\lambda$, then the iterates $\theta_{t+1} = (1 - \eta\lambda)\theta_t$ of $\mathrm{GD}^\ell(\eta, \cdot)$ grow in magnitude like $|1 - \eta\lambda|^t$ leading to rapid divergence. In contrast, DNNs are typically trained in a regime called the *edge of stability* (EOS) [13], in which $\eta$ is often (pointwise) strictly larger than $2/\lambda$. Miraculously, despite this large learning rate, GD is capable of converging *non-monotonically* to a global minimum of the loss, exhibiting an apparent bias toward less sharp (*flatter*) minima.

Despite significant theoretical progress in the past few years, GD at the edge of stability is still poorly understood when compared with its small learning rate counterpart. In particular, while convergence rates for GD in the small learning rate regime can be obtained for DNNs using standard methods [25], convergence rates for GD at the edge of stability have so far only been obtained in settings wherein monotonic loss decrease can be eventually guaranteed [42, 41, 11, 32].

Our purpose in this paper is to provide convergence rates in a *least squares* setting, wherein sustained monotonic decrease of the loss *may never occur*. To our knowledge, these are the first rates of this kind to be provided in the literature.

Our analysis synthesises *Riemannian geometry* and *dynamical systems theory* to formalise an intuition that has been folklore in the literature for some time [16, 12], namely that GD with a large step size implicitly favours flatter global minima. This formalisation is achieved for *codimension 1, overparametrised least squares problems*, including deep scalar factorisation, in which the solutions (global minima) are guaranteed to form a $p - 1$-dimensional Riemannian submanifold $M$ of the $p$-dimensional Euclidean parameter space $\mathbb{R}^p$. Inspired by the "self-stabilisation" idea proposed in [16], we introduce new coordinates $(\theta^\parallel, \theta^\perp)$ for a neighbourhood of $M$, where $\theta^\parallel$ is a point in $M$ and $\theta^\perp$ coordinatises the direction orthogonal to $M$, with respect to which the GD map $(\theta^\parallel, \theta^\perp) \mapsto (\mathrm{GD}^\parallel(\theta^\parallel, \theta^\perp), \mathrm{GD}^\perp(\theta^\parallel, \theta^\perp))$ has approximately the following form:

$$\mathrm{GD}^\parallel : (\theta^\parallel, \theta^\perp) \mapsto \mathrm{GD}_M^\lambda\left(\frac{\eta(\theta^\perp)^2}{2c(\eta, \theta^\parallel)}, \theta^\parallel\right) \qquad \mathrm{GD}^\perp : (\theta^\parallel, \theta^\perp) \mapsto (1 - \eta\lambda(\theta^\parallel))\theta^\perp + (\theta^\perp)^3. \quad (2)$$

The left map $\mathrm{GD}_M^\lambda$ is Riemannian gradient descent on $\lambda$ along $M$ (see (4)), with step size $\eta(\theta^\perp)^2/(2c(\eta, \theta^\parallel))$ for some strictly positive number $c(\eta, \theta^\parallel)$; the right is the normal form for a period-doubling bifurcation [27]. Our analysis thereby clarifies the implicit bias of GD toward flat minima:

> In a neighbourhood of the solution manifold, GD with a large learning rate implicitly performs *Riemannian GD on the sharpness* along the solution manifold, and oscillates as a *bifurcating dynamical system* in the direction orthogonal to the solution manifold.

**Paper contributions.** In the context of codimension 1 least squares problems, we:

1. Make the implicit bias of GD toward flat minima explicit: overparametrisation enables the decomposition of GD into an explicitly sharpness-minimising component along the solution manifold, controlled by an oscillating component orthogonal to the solution manifold.

2. Prove that a class of non-convex scalar factorisation problems fit into our framework. In particular, we prove that despite the non-convexity of the landscape itself, the sharpness in such examples is *geodesically strongly convex* along a geodesic ball in the solution manifold. To our knowledge, this observation has not previously been made.

3. Identify and prove convergence and implicit bias results for large step size GD in three regimes:

   (a) the *subcritical* regime, in which the iterates can be guaranteed to (eventually) decrease monotonically at a linear rate until convergence to a suboptimally flat minimum;
   (b) the *critical* regime, in which the iterates converge non-monotonically with a power law rate to the sharpness-minimising global minimum;
   (c) the *supercritical* regime, in which the iterates converge linearly to a stable period-two orbit orthogonal to the solution manifold at the point of minimum sharpness.

   Although the former and the latter have already been observed in some examples, to our knowledge the middle regime has not yet been identified. Moreover, our work is the first to be able to theoretically quantify the rate of convergence and implicit bias in all three regimes.

4. Verify our theory against numerical experiments.

## 2 Related work

**Overparametrisation and convergence guarantees for DNNs.** Convergence analyses of GD for non-convex loss landscapes such as DNNs are by now well-established [19, 18, 3, 35, 34, 36, 10]. Although none of these works are able to account for non-monotonic convergence of GD using a large step size, they do point to *overparametrisation* as a key feature of DNN loss landscapes which enables a (local) Polyak-Łojasiewicz inequality and thus convergence guarantees in the absence of convexity. Specifically, overparametrisation enables the map $f$ sending parameters to network outputs to be *submersive*, in the sense that the derivative matrix $Df$ of $f$ is pointwise full-rank. Much of the analysis involved in the aforementioned papers is aimed at quantifying this submersivity by lower-bounding the smallest eigenvalue of the "neural tangent kernel" $Df\,Df^T$ [24, 10]. Overparametrisation in this sense plays a key role in our work too, since it guarantees that the global minima form a smooth manifold with a well-defined normal direction, which is the foundation for Equation (2) and all the analysis that follows.

**GD with large learning rates.** Recent work in convex optimisation has demonstrated surprising benefits of large step sizes [21, 22, 23, 4, 5], however the mechanism in these works appears to be distinct from the EOS phenomenon in deep learning. Concerning deep learning, although local stability analysis had already hinted at a relation between step size and sharpness prior to 2020 [43], the seminal work of [13] in 2021 was the first to conduct a systematic empirical study of large step size GD. In particular, [13] exposed the fact that GD on DNNs typically uses a larger step size than admissible by classical theory. Moreover, [13] made the empirical observation that this larger step size implicitly regularises sharpness during the late stages training, and coined the term "edge of stability" for this phase. An explosion of empirical and theoretical work has since been produced in an attempt to account for these empirical observations [2, 7, 40, 38, 39, 16, 30, 44, 1, 12, 26, 42, 41, 11, 17, 32, 20]. At a high level, work up to this point on large step size GD can be classified into one of two categories: *general* analysis attempting to outline the essence of the mechanism of stability in this regime while abstracting from specific architectures [16, 14]; and *specific* analysis focusing on precise architectural details in an attempt to derive more fine-grained results on convergence [42, 41, 11, 32] or implicit bias [38, 39, 12, 17]. The advantage of the former approach is generality and clarity of insight, with the disadvantage of less facility in the proof of quantitative results. The advantage of the latter is the wealth of tools for the proof of quantitative results, with the disadvantage of obfuscating essential mathematical structures with inessential architectural detail.

Our work attempts to achieve the strengths of both approaches with a middle ground of abstraction. On the one hand, in common with and inspired by the seminal work [16], our formulation identifies and hinges on minimal mathematical structures which underlie the dynamics of GD with large step size, independent of architectural details; on the other hand, in common with the latter works, we retain enough mathematical structure in our assumptions to cover some of the specific examples already studied in prior work and obtain quantitative convergence rates and implicit bias characterisations. In particular, we are able to provide convergence rates outside of the "eventually monotonically stable" regimes considered in prior work [42, 41, 11, 32], as well as rigorously quantify oscillatory implicit biases of large step size GD observed empirically and partially accounted for theoretically in [12, 17].

## 3 Theoretical setting

### 3.1 Notation

Given a smooth (i.e., infinitely differentiable, $C^\infty$) manifold $M$ without boundary, and a point $\theta \in M$, we will use $T_\theta M$ to denote the tangent space to $M$ at $\theta$, and $TM := \bigsqcup_{\theta \in M} T_\theta M$ to denote its tangent bundle. Given a smooth map $g : M \to N$ of smooth manifolds, we will denote by $D_M g : TM \to TN$ its derivative, acting at each point $\theta \in M$ as a linear map $D_M g(\theta) : T_\theta M \to T_{g(\theta)} N$ between tangent spaces.

A *Riemannian metric* on $M$ is a smoothly-varying inner product $\langle \cdot, \cdot \rangle_\theta$ on each $T_\theta M$; we will usually drop the subscript and denote the metric by simply $\langle \cdot, \cdot \rangle$. Given Riemannian metrics on $M$ and $N$, any higher derivative $D_M^k g : TM^{\otimes k} \to TN$ is also defined, and at each point $\theta \in M$ acts a symmetric, multilinear map $D_M^k g(\theta) : T_\theta M^{\otimes k} \to T_{g(\theta)} N$ whose value on vectors $v_1, \ldots, v_k \in T_\theta M$ we denote $D_M^k g(\theta)[v_1, \ldots, v_k]$ (respectively, $D_M^k g(\theta)[v^{\otimes k}]$ if $v_i = v \; \forall i$). This $D_M^k g$ may also act on vector fields $V_1, \ldots, V_k : M \to TM$ to give a map $D_M^k g[V_1, \ldots, V_k] : M \to TN$ defined by the formula

$D_M^k g[V_1, \ldots, V_k](\theta) := D_M^k g(\theta)[V_1(\theta), \ldots, V_k(\theta)]$ (resp. $D_M^k g[V^{\otimes k}](\theta) := D_M^k g(\theta)[V(\theta)^{\otimes k}]$ if $V_i = V \ \forall i$) for $\theta \in M$.

If $g : M \to \mathbb{R}$ is scalar-valued, then $\nabla_M^k g : TM^{\otimes k-1} \to TM$ will be used to denote the map obtained by dualising one of the inputs of $D_M^k g$ using the Riemannian metric. Specifically, given any $\theta \in M$ and tangent vectors $v_2, \ldots, v_k \in T_\theta M$, $\nabla_M^k g(\theta)[v_2, \ldots, v_k]$ is the unique element of $T_\theta M$ satisfying

$$\langle \nabla_M^k g(\theta)[v_2, \ldots, v_k], v_1 \rangle = D_M^k g(\theta)[v_1, v_2, \ldots, v_k], \qquad \forall v_1 \in T_\theta M, \tag{3}$$

where $\langle \cdot, \cdot \rangle$ is the Riemannian metric in $T_\theta M$. The objects $\nabla_M g$ and $\nabla_M^2 g$ are the *Riemannian gradient* and *Riemannian Hessian* respectively. In particular, $D_{\mathbb{R}^p}^k g$ and $\nabla_{\mathbb{R}^p}^k g$ will be denoted simply $D^k g$ and $\nabla^k g$; $\nabla g$ and $\nabla^2 g$ then are the ordinary Euclidean gradient and Hessian respectively.

A *geodesic* is a locally length-minimising curve in $M$ (for instance, geodesics in Euclidean space are just straight lines). We will use $d_M(\theta, \theta')$ to denote the geodesic distance between $\theta, \theta' \in M$, which is the length of the shortest geodesic connecting $\theta$ to $\theta'$; $d_M$ makes $M$ into a metric space. For any $\theta \in M$ and any sufficiently small $v \in T_\theta M$, there is a unique geodesic $\gamma : [0, 1] \to M$ such that $\gamma(0) = \theta$ and $\dot{\gamma}(0) = v$; the *exponential map* on $(\theta, v)$ is then by definition $\exp_\theta(v) := \gamma(1)$; in Euclidean space $\exp_\theta(v) = \theta + v$. See Appendix A for more details. Any smooth function $g : M \to \mathbb{R}$ is associated to a *Riemannian gradient descent* map $(\eta, \theta) \mapsto \mathrm{GD}_M^g(\eta, \theta) \in M$ defined in an open neighbourhood of $\{0\} \times M \subset \mathbb{R}_{\geq 0} \times M$ by

$$\mathrm{GD}_M^g(\eta, \theta) := \exp_\theta\big(-\eta \nabla_M g(\theta)\big). \tag{4}$$

This reduces to the familiar $(\eta, \theta) \mapsto \theta - \eta \nabla g(\theta)$ in the Euclidean setting, wherein it is denoted simply $\mathrm{GD}^g$ with the subscript dropped.

Recall that a *regular value* of a function $f : \mathbb{R}^p \to \mathbb{R}^d$ is a point $y \in \mathbb{R}^d$ such that $f^{-1}\{y\}$ is nonempty and $f$ is $C^\infty$ in a neighbourhood of $f^{-1}\{y\}$ with derivative $Df(\theta) : T_\theta \mathbb{R}^p \to T_{f(\theta)} \mathbb{R}^d$ surjective for all $\theta \in f^{-1}\{y\}$. Moreover, if $y$ is a regular value of $f$, then $M := f^{-1}\{y\}$ is a smooth submanifold of $\mathbb{R}^p$, and if $f$ is also analytic in a neighbourhood of $M$, then $M$ is an analytic manifold.

## 3.2 Problem setting

Given a natural number $p > 1$, function $f : \mathbb{R}^p \to \mathbb{R}$ and target value $y \in \mathbb{R}$, we aim to solve the *codimension 1 least squares problem*

$$\min_\theta \ell(\theta) := \frac{1}{2}(f(\theta) - y)^2, \tag{5}$$

using gradient descent with constant step size $\eta > 0$:

$$\theta \mapsto \mathrm{GD}^\ell(\eta, \theta) := \theta - \eta \nabla \ell(\theta). \tag{6}$$

We will make the following assumptions on $f$ and $y$.

**Assumption 3.1** (Regularity and analyticity). The point $y \in \mathbb{R}$ is a regular value of $f$, and $f$ is analytic in a neighbourhood of the pre-image $M := f^{-1}\{y\}$.

The assumption that $y$ is a regular value of $f$ is a strong version of *overparametrisation*: it is equivalent to assuming that the "neural tangent kernel" $Df\,Df^T = \|Df\|^2$ is non-vanishing along $M$, as is often insisted upon in overparametrisation theory for deep learning [24, 10]. Analyticity of $f$ in a neighbourhood of $M := f^{-1}\{y\}$ will be used to invoke powerful results from holomorphic dynamics [6] in our convergence theorems.

The key consequence of Assumption 3.1 is that, by the regular value theorem, the solution set $M$ of (5) is a $(p-1)$-dimensional analytic submanifold of $\mathbb{R}^p$. Points in $M$ will be denoted $\theta^{\|}$ in what follows. For any $\theta^{\|} \in M$, the tangent space $T_{\theta^{\|}} M$ to $M$ at $\theta^{\|}$ is the kernel of $Df(\theta^{\|})$, and any singular value decomposition of $Df(\theta^{\|})$ admits precisely one nonzero singular value corresponding to a singular vector orthogonal to $T_{\theta^{\|}} M$. This singular vector is the *normal vector* $n(\theta^{\|}) := \nabla f(\theta^{\|})/\|\nabla f(\theta^{\|})\|$.

It follows from the chain rule and the fact that $f \equiv y$ along $M$ that the Hessian $\nabla^2 \ell$ of $\ell$ satisfies the identity

$$\nabla^2 \ell|_M \equiv \big(\nabla f \nabla f^T + (f - y)\nabla^2 f\big)|_M \equiv \nabla f \nabla f^T|_M \tag{7}$$

along $M$. We denote by $\lambda$ the largest eigenvalue of $\nabla^2 \ell$, which is equal to $\|\nabla f\|^2$. We assume $M$ to be equipped with the Riemannian metric inherited from its embedding into the Euclidean space $\mathbb{R}^p$.

Our new coordinate representation (2) for gradient descent is essentially a consequence of coordinatising a tubular neighbourhood $N$ of $M$ by a $\|$-coordinate *along $M$*, and a $\perp$-coordinate *orthogonal* to $M$ (Figure 1). However, the rigorous form of (2) requires two additional assumptions on the solution manifold $M$. The first of these provides an invariant line segment about which GD can oscillate, and enables the correct decay rate of the error terms in the $\|$-component of GD.

**Assumption 3.2** (Orthogonal stability). There is $\theta_*^\| \in M$ and a line segment $\mathcal{L}$ through $\theta_*^\|$ orthogonal to $M$ which is invariant under gradient descent on $\ell$ (see Figure 1).

The second assumption necessary for the rigorous form of (2), concerning the $\perp$-equation, is a standard assumption from bifurcation theory enabling the realisation of the $\perp$-equation in a well-known normal form which is easily analysed [27, Theorem 4.3].

**Assumption 3.3** (Genericity). Recall the normal vector field $n := \nabla f / \|\nabla f\|$. At each point $\theta^\| \in M$, for all $\eta$ in a neighbourhood of $2/\lambda(\theta^\|)$, one has

$$c(\eta, \theta^\|) := \left( \frac{\eta}{2} D^3 \ell [n^{\otimes 3}](\theta^\|) \right)^2 - \frac{\eta}{6} D^4 \ell [n^{\otimes 4}](\theta^\|) > 0. \tag{8}$$

Assumptions 3.1, 3.2 and 3.3 are sufficient to obtain the rigorous form of (2) (Theorem 4.1). As is evident from Equation (2), GD with a large step size behaves like gradient descent on the sharpness along $M$. To utilise this fact in our convergence theorems, we will make the following additional assumptions on $M$ and the sharpness $\lambda$.

**Assumption 3.4** (Strong convexity of $\lambda$). There is a ball $B_M(\theta_*^\|, r) \subset M$, centred on $\theta_*^\|$ and of radius $r > 0$ with respect to the geodesic distance, which is *geodesically convex* in the sense that any two points in $B_M(\theta_*^\|, r)$ are connected by a unique distance-minimising geodesic, over which the sharpness $\lambda : M \to \mathbb{R}$ is $\mu$-*geodesically strongly convex* and has $L$-*geodesically Lipschitz gradients*, in the sense that the Riemannian Hessian $\nabla_M^2 \lambda$ of $\lambda$ satisfies

$$\mu I_{TM} \preceq \nabla_M^2 \lambda \preceq L\, I_{TM} \tag{9}$$

Figure 1: $M = \{(x,y) : xy = 1\}$ with tubular neighbourhood $N$ (shaded) and line $\mathcal{L}$ (dotted). Inside $N$, any point $\theta$ is closest to a unique point $\theta^\|$ on $M$, with $\theta - \theta^\| = \theta^\perp n(\theta^\|)$ orthogonal to $M$ for some $\theta^\perp \in \mathbb{R}$. Assumption 3.2 says that $\nabla \ell(\theta')$ is parallel to $\mathcal{L}$ at any point $\theta' \in \mathcal{L}$.

uniformly over $B_M(\theta_*^\|, r)$, where $I_{TM}$ is the identity map on $TM$. We assume moreover that $\lambda$ achieves its minimum value $\lambda(\theta_*^\|)$ in $B_M(\theta_*^\|, r)$ uniquely at $\theta_*^\|$, where one moreover has

$$\nabla_M^2 \lambda(\theta_*^\|) = \nu\, I_{TM}(\theta_*^\|) \tag{10}$$

for some $\nu > 0$, where $I_{TM}(\theta_*^\|)$ is the identity map on the tangent space $T_{\theta_*^\|} M$ of $M$ at $\theta_*^\|$.

These assumptions need not be satisfied for general functions $f$. The geodesic strong convexity of $\lambda$ in particular seems at first to be an alarmingly strong assumption, since it is well-known that DNN loss landscapes are *non*-convex. Surprisingly, these non-trivial assumptions *can be guaranteed to hold* for the following class of toy examples of deep learning non-convex loss landscapes.

**Multilayer scalar factorisation.** The map $f : \mathbb{R}^p \to \mathbb{R}$ defined by $f(\theta^1, \ldots, \theta^p) := \theta^p \cdots \theta^1$ corresponds to a linear network of depth $p$ and width 1 on a single input datum. Any nonzero target value $y \neq 0$ is a regular value for $f$, and $f^{-1}\{y\}$ is then a union of hypersurfaces (Proposition B.1). Assuming without loss of generality that $y > 0$ and setting $\theta_*^\| := y^{\frac{1}{p}} 1_p$, one takes $M$ to be the connected component of $\theta_*^\|$. The line $\mathcal{L}$ is given by the span of the ones-vector $1_p$ (Proposition B.2). Assumption 3.3 holds by Proposition B.3, while Proposition B.6 demonstrates that $\lambda$ is geodesically strongly convex on a geodesic ball centred at $\theta_*^\|$, with

$$\nabla_M^2 \lambda(\theta_*^\|) = 4 y^{2 - 4/p} I_{TM}(\theta_*^\|) \tag{11}$$

so that Assumption 3.4 holds. One may also take $f(\theta^1, \ldots, \theta^p) := \theta^p \phi(\theta^{p-1} \phi(\cdots (\phi(\theta^1)) \cdots))$, with $\phi$ any nonlinearity that is the identity on a neighbourhood of $y^{1/p}$ and still satisfy all assumptions.

Some thought reveals that these assumptions can be relaxed in more-or-less straightforward ways to deep linear networks of sufficient constant width on multi-point datasets. However, the scalar case we consider in this paper is already sufficiently instructive and non-trivial that we leave a more general elaboration of this framework to future work.

## 4    A normal form for gradient descent about the solution manifold

In this section we state a rigorous form (Theorem 4.1) of the approximate update equations alluded to in Equation (2), and give an idea of its proof. This section makes use only of Assumptions 3.1, 3.2 and 3.3. The geodesic convexity of Assumption 3.4 is *not* needed to derive the normal form of GD about $M$, and is necessary only for the convergence theorems to be given in the next section.

Our analysis of the dynamics of GD is inspired by that of [16]; it proceeds from a Taylor expansion[2]. However, while [16] Taylor expands around a set of points with sharpness $\leq 2/\eta$, we Taylor expand around the solution manifold $M$. Our derivation is novel and provides a number of advantages: the assumptions are more transparent and easily checked, and we are guaranteed that $M$ is a Riemannian manifold, while the set of points with sharpness $\leq 2/\eta$ considered in [16] need not be.

Specifically, every point $\theta$ sufficiently close to $M$ admits a unique nearest point $\theta^{\|}$ on $M$. The difference $\theta - \theta^{\|}$ is then a scalar multiple $\theta^{\perp} n(\theta^{\|})$ of the normal vector $n(\theta^{\|}) = \nabla f(\theta^{\|})/\|\nabla f(\theta^{\|})\|$ at $\theta^{\|}$. Thus, pairs $(\theta^{\|}, \theta^{\perp})$ with $\theta^{\|} \in M$ and sufficiently small $\theta^{\perp} \in \mathbb{R}$ suffice to completely coordinatise a neighbourhood of $M$ in $\mathbb{R}^p$ known as a *tubular neighbourhood* $N$ [28, p. 147]. Our first result is then the following characterisation of the GD dynamics in terms of $\theta^{\|}, \theta^{\perp}$, the sharpness $\lambda$ and its Riemannian gradient descent map $\text{GD}_M^{\lambda}$.

**Theorem 4.1** (Normal form for GD about $M$). *There is an analytic change of coordinates* $(\eta, \theta^{\|}, \theta^{\perp}) \mapsto (\eta, \theta)$ *for a tubular neighbourhood $N$ of $M$ in which the gradient descent map* $\text{GD}^{\ell} :$ $(\eta, \theta) \mapsto (\eta, \theta - \eta \nabla \ell(\theta))$ *takes the form* $(\eta, \theta^{\|}, \theta^{\perp}) \mapsto (\eta, \text{GD}^{\|}(\eta, \theta^{\|}, \theta^{\perp}), \text{GD}^{\perp}(\eta, \theta^{\|}, \theta^{\perp}))$, *where*

$$\text{GD}^{\|}(\eta, \theta^{\|}, \theta^{\perp}) = \text{GD}_M^{\lambda} \left( \frac{\eta (\theta^{\perp})^2}{2 c(\eta, \theta^{\|})}, \theta^{\|} \right) + O\big((\theta^{\perp})^3 \min\{1, d_M(\theta^{\|}, \theta_*^{\|})\}\big), \qquad (12)$$

$$\text{GD}^{\perp}(\eta, \theta^{\|}, \theta^{\perp}) = (1 - \eta \lambda(\theta^{\|}))\theta^{\perp} + (\theta^{\perp})^3 + O\big((\theta^{\perp})^4\big), \qquad (13)$$

*with $c(\eta, \theta^{\|})$ defined in Assumption 3.3.*

*Outline of proof.* Recalling the normal vector field $n := \nabla f / \|\nabla f\|$, we approximate the gradient $\nabla \ell(\theta)$ at $\theta \in N$ by the Taylor series

$$\nabla \ell(\theta) = \nabla \ell(\theta^{\|}) + \theta^{\perp} \nabla^2 \ell(\theta^{\|})[n(\theta^{\|})] + \frac{(\theta^{\perp})^2}{2} \nabla^3 \ell(\theta^{\|})[n(\theta^{\|}), n(\theta^{\|})] + O((\theta^{\perp})^3). \qquad (14)$$

Since $M$ consists of global minima for $\ell$, $\nabla \ell(\theta^{\|}) = 0$. Additionally, since $n(\theta^{\|})$ is the sole eigenvector along which $\nabla^2 \ell(\theta^{\|}) = \nabla f(\theta^{\|}) \nabla f(\theta^{\|})^T$ is nontrivial, with eigenvalue $\lambda(\theta^{\|})$, one has $\nabla^2 \ell(\theta^{\|})[n(\theta^{\|})] = \lambda(\theta^{\|}) n(\theta^{\|})$. One thus has:

$$\nabla \ell(\theta) = \lambda(\theta^{\|})\theta^{\perp} n(\theta^{\|}) + \frac{|\theta^{\perp}|^2}{2} \nabla^3 \ell(\theta^{\|})[n(\theta^{\|}), n(\theta^{\|})] + O((\theta^{\perp})^3). \qquad (15)$$

Plugging this into the gradient descent update formula, applying the orthogonal projection $P_{TM}$ of $T\mathbb{R}^p$ onto $TM$ and $n^T$ of $T\mathbb{R}^p$ onto the normal direction respectively, and noting that

$$P_{TM} \nabla^3 \ell[n, n] = \nabla_M \big(\nabla^2 \ell[n, n]\big) = \nabla_M \lambda \qquad (16)$$

---

[2]The idea of higher order terms inducing implicit bias is also present in literature on Sharpness-Aware Minimisation [8].

since $\nabla(n^T n) = \nabla(1) = 0$, yields the following formulae for the GD update in $(\theta^\|, \theta^\perp)$:

$$\theta^\| \mapsto \theta^\| - \frac{\eta(\theta^\perp)^2}{2} \nabla_M \lambda(\theta^\|) + O((\theta^\perp)^3) \tag{17}$$

$$\theta^\perp \mapsto (1 - \eta\lambda(\theta^\|))\theta^\perp - \frac{\eta(\theta^\perp)^2}{2} D^3\ell(\theta^\|)[n(\theta^\|)^{\otimes 3}] + O((\theta^\perp)^3). \tag{18}$$

The $O((\theta^\perp)^3)$ error in the $\|$-update is tightened to $O\big((\theta^\perp)^3 \min\{1, d_M(\theta^\|, \theta_*^\|)\}\big)$ using Assumption 3.2, and $\theta^\| - \frac{\eta(\theta^\perp)^2}{2}\nabla_M\lambda(\theta^\|)$ is the first order approximation of $\exp_{\theta^\|}\big(-\frac{\eta(\theta^\perp)^2}{2}\nabla_M\lambda(\theta^\|)\big)$ by the Taylor expansion of the exponential map [33]. The final form is obtained by invoking Assumption 3.3 and applying a standard transformation from bifurcation theory [27, Theorem 4.3]. See Theorem C.6 for a rigorous proof. $\qquad\square$

We can immediately make the following qualitative observations of the dynamics in Theorem 4.1, with which we will be able to give an idea of how our convergence theorems work. Let $(\theta_t^\|, \theta_t^\perp)$ denote the $t^{th}$ iterate of gradient descent in the $(\theta^\|, \theta^\perp)$ coordinates of Theorem 4.1.

The $\|$-component of the GD dynamics is a perturbed version of *Riemannian gradient descent on the sharpness* along the solution manifold $M$, with time-varying step size determined by the magnitude of the $\perp$-iterates $\theta_t^\perp$. Recalling that we assume $\lambda$ to be geodesically strongly convex as in Assumption 3.4, we can be assured of descent guarantees for $\lambda$ provided $|\theta_t^\perp|$ can be controlled.

The behaviour of $|\theta_t^\perp|$ is governed by the map (13), which is a perturbed, time-varying version of the simpler "flip bifurcation"

$$x \mapsto (1 - \eta\lambda')x + x^3 \tag{19}$$

from bifurcation theory [27]. The iterates $x_t$ of (19) admit the following dynamics.

1. When $\eta < 2/\lambda'$, $0$ is a *hyperbolic* attractor, and the iterates $|x_t|$ go to zero exponentially fast. Were this to hold also for the iterates $|\theta_t^\perp|$ of (13), $\theta_t^\|$ would evolve like gradient descent on $\lambda$ with exponentially decaying step size, hence would converge exponentially fast to a suboptimally flat point.

2. When $\eta = 2/\lambda'$, $0$ is a *parabolic* attractor, and the iterates $|x_t|$ go to zero like $\Theta(t^{-1/2})$. Were this to hold also for the iterates $|\theta_t^\perp|$ of (13), $\theta_t^\|$ would evolve like gradient descent on $\lambda$ with power-law-decaying stepsize, hence would converge with a power-law rate to the optimally flat point.

3. When $\eta > 2/\lambda'$, there is a hyperbolically attracting orbit of period two with amplitude $\approx \sqrt{\eta\lambda' - 2}$ to which the iterates $x_t$ converge exponentially fast. Were the same to be true for the iterates $\theta_t^\perp$ of (13), $\theta_t^\|$ would evolve like gradient descent on $\lambda$ with step size $\Theta\big((\eta\lambda' - 2)\big)$, hence would converge exponentially to the optimally flat point.

In the next section, we provide theorems showing that these intuitions gathered from the unperturbed, time-invariant (19) can be rigorously carried over to the perturbed, time-varying system defined by Theorem 4.1.

## 5 Convergence theorems

In this section, we state our convergence theorems and provide numerical demonstrations of them on a multilayer scalar factorisation problem. In addition to the Assumptions 3.1, 3.2 and 3.3 required for the normal form (Theorem 4.1), we now also invoke Assumption 3.4 to provide rates of convergence. Iterates will be denoted $\theta_t$ (respectively $(\theta_t^\|, \theta_t^\perp)$) for $t \in \mathbb{N} \cup \{0\}$. For convenience, our $|\theta_t^\perp|$ plots concern the orthogonal coordinate of the intermediate Proposition C.5 rather than that of the final Theorem 4.1; since the transformation going between them is analytic (Theorem C.6), this causes no difference in the convergence rate.

## 5.1 Subcritical regime

The subcritical regime is when $2/\lambda(\theta_*^{\parallel}) > \eta > 2/\lambda(\theta_0^{\parallel})$. In this regime, considered also in certain examples in prior works [26, 32], the orthogonal component $\theta_t^{\perp}$ of the iterates exhibits initial, transient oscillation driven by (13) of Theorem 4.1; during this time, the sharpness values $\lambda(\theta_t^{\parallel})$ are monotonically decreasing according to Theorem 4.1 to the point of stability $\eta < 2/\lambda(\theta_t^{\parallel})$ in a finite number of steps. Following the achievement of stability, $\theta_t^{\perp}$ decays exponentially fast, resulting in exponentially less aggressive steps in $\theta_t^{\parallel}$ to decrease $\lambda(\theta_t^{\parallel})$, ultimately resulting in convergence to a suboptimally-flat global minimum of $\ell$. See Figure 2. The theorem below appears as Theorem D.4 in the appendix.

**Theorem 5.1.** *Assume that $\eta < 2/\lambda(\theta_*^{\parallel})$. Then there is a constant $\gamma > 0$ such that for all $\theta_0^{\perp}$ sufficiently close to zero and all $\theta_0^{\parallel} \neq \theta_*^{\parallel} \in B_M(\theta_*^{\parallel}, r)$, if $\eta > 2/\lambda(\theta_0^{\parallel})$ is sufficiently small, then there is*

$$\tau \leq O\left((\theta_0^{\perp})^{-2}\left(\frac{\lambda(\theta_0^{\parallel}) - \lambda(\theta_*^{\parallel})}{2/\eta - \lambda(\theta_*^{\parallel})}\right)^{\gamma}\right) \tag{20}$$

*such that $\eta < 2/\lambda(\theta_t^{\parallel})$ for all $t \geq \tau$ and $\eta \geq 2/\lambda(\theta_t^{\parallel})$ for all $t < \tau$. Consequently, setting $\beta := 1 - (2 - \eta\lambda(\theta_\tau^{\parallel})) < 1$, the iterates $(\theta_t^{\parallel}, \theta_t^{\perp})$ converge to a global minimum $(\theta_\infty^{\parallel}, 0)$ of $\ell$ in $M$ with suboptimal sharpness*

$$\lambda(\theta_\infty^{\parallel}) - \lambda(\theta_*^{\parallel}) \geq \exp\left(-O((\theta_\tau^{\perp})^2(1 - \beta^2)^{-1})\right)(\lambda(\theta_\tau^{\parallel}) - \lambda_*). \tag{21}$$

*at a rate $O(\beta^t)$.*

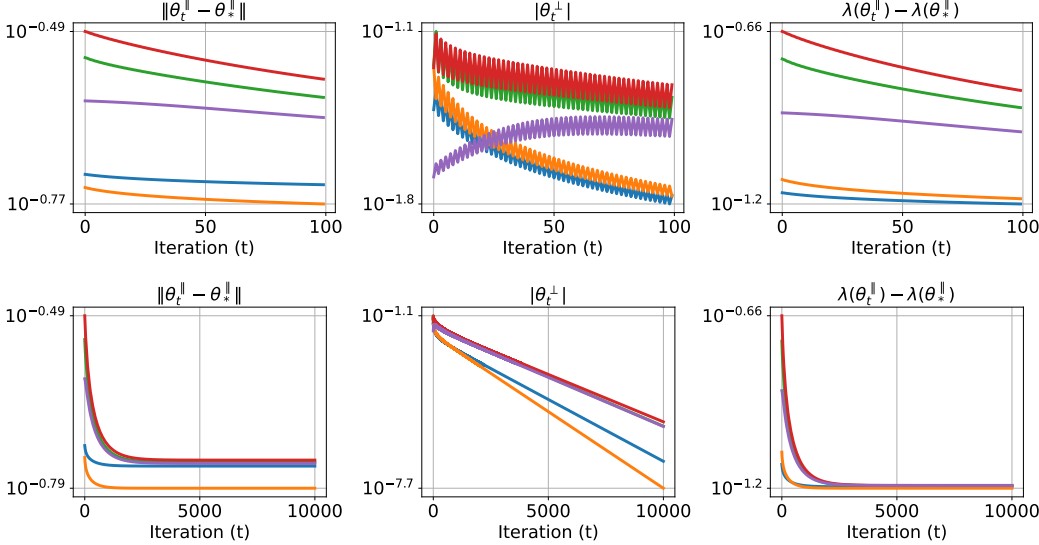

Figure 2: Log-scale plots of distance from $\theta_t^{\parallel}$ to $\theta_*^{\parallel}$ (left), magnitude of $\theta_t^{\perp}$ (centre) and sharpness suboptimality gap (right) for gradient descent on depth 5 scalar factorisation in the **subcritical regime**. Trajectories from five different initialisations shown. Initial instability in $|\theta_t^{\perp}|$ (top) is overcome in finite time with rapid convergence to a suboptimally flat global minimum (bottom).

## 5.2 Critical regime

The critical regime happens when $\eta = 2/\lambda(\theta_*^{\parallel})$. To our knowledge, this regime has not been observed previously in the literature. In this regime, the dynamics of GD are non-hyperbolic, with $\theta_t^{\perp}$ exhibiting 2-periodic decay to zero with a rate $\Theta(t^{-\frac{1}{2}})$. Consequently, the dynamics of $\theta_t^{\parallel}$ are essentially those of gradient descent on $\lambda$ with a step size decaying according to a power-law. Ultimately then, $\theta_t$ does converge to the optimally flat global minimum of $\ell$, but does so at a slow rate. See Figure 3. The theorem below appears as Theorem D.9 in the appendix.

**Theorem 5.2.** *Assume that $\eta = 2/\lambda(\theta_*^{\|})$ and that $\nu/(c(2/\lambda_*, \theta_*^{\|})\lambda_*) < 1$, where $\nu$ and $c(\cdot, \cdot)$ are defined as in Assumptions 3.3 and 3.4 respectively. Then for all $\theta_0^{\|}$ sufficiently close to $\theta_*^{\|}$ and all $\theta_0^{\perp} \neq 0$ sufficiently small, one has $d_M(\theta_t^{\|}, \theta_*^{\|}) = \Theta(t^{-1/2})$ and $|\theta_t^{\perp}| = \Theta(t^{-1/2})$.*

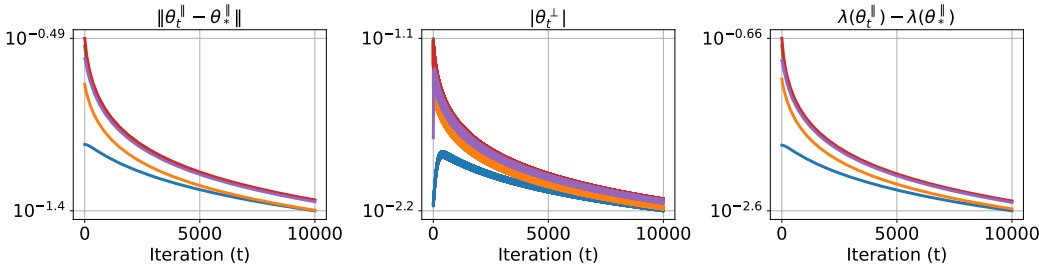

Figure 3: Log-$y$-scale plots of distance from $\theta_t^{\|}$ to $\theta_*^{\|}$ (left), magnitude of $\theta_t^{\perp}$ (centre) and sharpness suboptimality gap (right) on depth 5 scalar factorisation in the **critical regime**. Trajectories from five different initialisations shown. The iterates $|\theta_t^{\perp}|$ may or may not initially *increase*; in both cases, asymptotic, power law decrease of $|\theta_t^{\perp}|$ and $\|\theta_t^{\|} - \theta_*^{\|}\|$ to zero indicates power law convergence to the optimally flat global minimum.

Some further remarks about the critical regime are necessary. First, the additional hypothesis that $\nu/(c(2/\lambda_*, \theta_*^{\|})\lambda_*) < 1$ (which is satisfied for multilayer scalar factorisation, see Proposition B.7) is not strictly necessary for convergence; power-law convergence still occurs to the same minimum without this assumption, but is in that case faster for $d_M(\theta_t^{\|}, \theta_*^{\|})$ than the $\Theta(t^{-1/2})$ rate given in the theorem statement. However, we are unaware of examples having this faster rate. Second, the $|\theta_t^{\perp}|$ iterates may initially increase substantially as in Figure 3 depending on the initialisation; our theorem accommodates this behaviour explicitly (see Theorem D.8).

### 5.3 Supercritical regime

The supercritical regime is when $\eta > 2/\lambda(\theta_*^{\|})$. In this case, when $\eta > 2/\lambda(\theta_*^{\|})$ is sufficiently small, Equation (13) exhibits linear convergence to a stable orbit of period two with amplitude $\approx (\eta\lambda(\theta_*^{\|}) - 2)^{\frac{1}{2}}$. It follows that the step sizes in $\theta_t^{\|}$ in its descent on $\lambda$ are of *approximately constant* size, so that $\theta_t^{\|}$ will converge *linearly* to the optimally flat $\theta_*^{\|}$. However, the complete iterates $\theta_t$ of GD do not converge to a global minimum, but asymptotically oscillate orthogonally to $M$ about the point $\theta_*^{\|}$ along the line $\mathcal{L}$ of Assumption 3.2. See Figure 4. Although this regime has been observed in prior works [12, 17, 20], no general quantitative convergence theorem in this regime has yet been proved. The theorem below appears as Theorem D.10 in the appendix.

**Theorem 5.3.** *Assume that $\eta > 2/\lambda(\theta_*^{\|})$ is sufficiently small. Then there are positive constants $C_1, C_2, C_3$ and a stable orbit of period two of the form $\left(\theta_*^{\|}, \pm(\eta\lambda(\theta_*^{\|}) - 2)^{1/2} + O(\eta\lambda(\theta_*^{\|}) - 2)\right)$ to which the iterates $(\theta_t^{\|}, \theta_t^{\perp})$ starting from any $(\theta_0^{\|}, \theta_0^{\perp})$ satisfying $0 < |\theta_0^{\perp}| \leq C_1(\eta\lambda(\theta_*^{\|}) - 2)^{1/2}$ and $d_M(\theta_0^{\|}, \theta_*^{\|}) \leq C_2(\eta\lambda(\theta_*^{\|}) - 2)^{1/2}$ converge with rate $O\left((1 - C_3(\eta\lambda(\theta_*^{\|}) - 2))^t\right)$.*

## 6 Limitations, discussion and conclusion

Our work has a number of limitations, all of which point toward avenues for further exploration along the lines we have developed.

**Higher order orthogonal dynamics.** Prior work has indicated that beyond the 2-periodic behaviour we studied in this work, higher-order periodicity and chaos emerge as the learning rate increases [17]. Our Theorem 4.1 provides new insight into this behaviour as a manifestation of a bifurcating system oscillating orthogonally to the solution manifold. Our analysis only handles the simplest (namely 2-periodic) case of this supercritical behaviour. We expect our framework to be able to admit convergence proofs of higher oscillatory and chaotic behaviour also, using ideas from dynamical systems theory, however we do not expect this to be easy and have not attempted it in this paper.

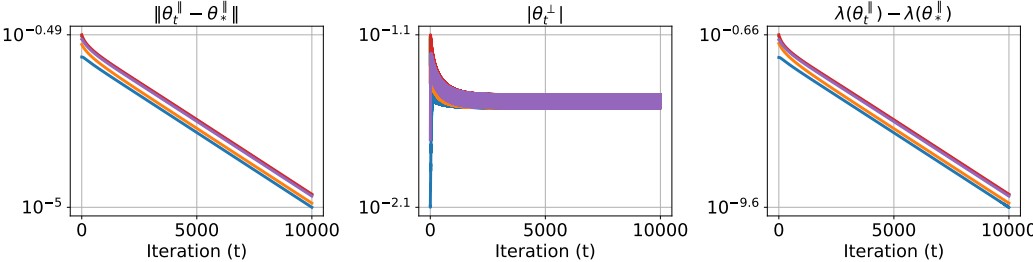

Figure 4: Log-scale plots of distance from $\theta_t^\|$ to $\theta_*^\|$ (left), magnitude of $\theta_t^\perp$ (centre) and sharpness suboptimality gap (right) on depth 5 scalar factorisation in the **supercritical regime**. Trajectories from five different initialisations shown. The iterates $|\theta_t^\perp|$ converge to a stable, period-two orbit driving linear convergence of $\theta_t^\|$ to the optimally flat global minimum $\theta_*^\|$.

**Higher codimension least squares.** Our theory only allows us to treat codimension 1 problems. This is clearly a severe limitation, ruling out for instance overparametrised regression on multiple datapoints or multiple outputs. It is not difficult to extend our definitions to higher codimension, however proving anything in this setting seems very challenging, as it would require an understanding of higher-dimensional bifurcating dynamical systems. We leave this question to future work.

**Geodesic convexity of sharpness for more general solution manifolds.** One of the most surprising results of our work is that the sharpness of the loss is geodesically strongly convex over a geodesic ball in the solution manifold for a number of overparametrised non-convex problems. Should this prove to be a more general fact, which we suspect may be the case, it would go a long way to explaining the still-mysterious ease of optimisation and implicit bias of GD on DNNs. We leave a more thorough investigation of this question to future work.

**Finding the tubular neighbourhood.** Our theory is premised on the *assumption* that GD is initialised in a tubular neighbourhood of the solution manifold. While there is no reason to think this is the case with standard initialisation schemes used in practice, we conjecture that GD does converge rapidly to a tubular neighbourhood from a standard initialisation during the phase known as progressive sharpening [13], after which the familiar dynamics described here would apply. We leave exploration of this question to future work.

**Relation to stochasticity.** Prior works [9, 15, 31] studying *stochastic* gradient descent have demonstrated an implicit bias toward flat minimisers arising from stochasticity. All of these works consider a smaller learning rate than those considered in our work; consequently, the mechanism behind the implicit bias considered in [9, 15, 31] is fundamentally different than that considered in our work. Studying how these distinct implicit biases interact is an important direction for future research.

## Acknowledgments and Disclosure of Funding

This research was supported by NSF 2031985 and Simons Foundation 814201 (Theorinet), ONR MURI 503405-78051 and University of Pennsylvania Startup Funds.

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

# A  Differential geometry background

The purpose of this section is to give a brief overview of essential notions from Riemannian geometry. A good source for the first subsection is [29]. We have not been able to locate a good source for the latter subsection; in particular, to our knowledge no attempt has yet been made to consider the Riemannian Hessian of the square gradient norm of a function defining a hypersurface, however the calculation is elementary.

## A.1  Riemannian metrics, connections and geodesics

Recall that a *Riemannian metric* on a manifold $M$ is a smoothly-varying family of positive-definite inner products $\langle \cdot, \cdot \rangle_\theta$ on the tangent spaces $T_\theta M$ of $M$, with associated norm $\| \cdot \|_\theta$. Any submanifold $M$ of a Riemannian manifold $(N, g_N)$ inherits a Riemannian metric from $N$; precisely, given tangent vectors $v, w \in T_\theta M \subset T_\theta N$, one defines

$$g_M(v, w) := g_N(v, w). \tag{22}$$

Since we work with submanifolds of Euclidean space, whose tangent spaces are all themselves subspaces of Euclidean space equipped with the Euclidean inner product, we will often use the notation $\langle \cdot, \cdot \rangle$ and $\| \cdot \|$ without further decoration when considering the metric of a submanifold of Euclidean space.

Denote by $T^*M$ the cotangent bundle of $M$, whose fibre over $\theta \in M$ is the space of linear functionals $T_\theta M \to \mathbb{R}$. Given $(k, l) \in \mathbb{N} \cup \{0\}$ a $(k, l)$-tensor at the point $\theta \in M$ is an element of $T_\theta M^{\otimes k} \otimes T_\theta^* M^{\otimes l}$, and a $(k, l)$-*tensor field* is a smooth section of the bundle $TM^{\otimes k} \otimes T^*M^{\otimes l}$, that is, a smooth assignment of a tensor $T(\theta) \in T_\theta M^{\otimes k} \otimes T_\theta^* M^{\otimes l}$ to each $\theta \in M$. The space of such sections will be denoted $\Gamma(TM^{\otimes k} \otimes T^*M^{\otimes l})$.

Whether or not $M$ carries a Riemannian metric, the derivative of a map $f : M \to \mathbb{R}$ makes sense as a map $D_M f : TM \to T\mathbb{R}$ which is fibrewise-linear in the sense that its restriction $D_M f(\theta) : T_\theta M \to T_{f(\theta)} \mathbb{R}$ to each fibre is a linear map. The derivative $D_M f$ can in this sense be thought of as a $(0, 1)$-tensor field. Higher derivatives $D_M^k f$ of $f$ are thus derivatives of tensor fields, and can be defined using a Riemannian metric. Specifically, assuming that $M$ has a Riemannian metric, there is a distinguished derivative operator $D_M : \Gamma(TM^{\otimes k} \otimes T^*M^{\otimes l}) \to \Gamma(TM^{\otimes k} \otimes T^*M^{\otimes l+1})$ defined for all $(k, l) \in \mathbb{N} \cup \{0\}$ called the *Levi-Civita connection*. The additional $T^*M$-slot gained by applying $D_M$ to a tensor $T$ is to be thought of as a *direction* in which $T$ is differentiated by $D_M$. In particular, given $T \in \Gamma(TM^{\otimes k} \otimes T^*M^{\otimes l})$ and $X \in \Gamma(TM)$, the notation $D_M T[X] \in \Gamma(TM^{\otimes k} \otimes T^*M^{\otimes l})$ is the *directional derivative of $T$ in the direction $X$*. Note that when $M = \mathbb{R}^p$, $D := D_{\mathbb{R}^p}$ is the usual multivariate derivative, and the higher derivatives $D^k := D_{\mathbb{R}^p}^k$ coincide with the usual higher-order derivatives in Euclidean space. If $g : M \to \mathbb{R}$ is a smooth, scalar-valued function, then $\nabla_M^k g \in \Gamma(TM \otimes T^*M^{\otimes k-1})$ will be used to denote $D_M^k g$ with one of the $T^*M$-slots dualised to a $TM$-slot as in (3). In particular, $\nabla_M g$ and $\nabla_M^2 g$ are the *Riemannian gradient* and *Riemannian Hessian* respectively; they coincide with the usual gradient $\nabla g$ and Hessian $\nabla^2 g$ respectively when $M = \mathbb{R}^p$.

A special case of the Levi-Civita connection is that inherited by a submanifold. If $M$ is a submanifold of a Riemannian manifold $N$, then the Levi-Civita connection associated to the induced metric (22) is $D_M := P_{TM} D_N$, where $P_{TM}$ is the orthogonal projection $TN \to TM$ induced by the metric on $N$.

Given $(\theta, v) \in TM$, classical ordinary differential equation (ODE) theory guarantees the existence of a unique $M$-valued solution $\gamma$ to the $TM$-valued second order ODE

$$(D_M \dot{\gamma})[\dot{\gamma}] = 0, \qquad \gamma(0) = \theta, \quad \dot{\gamma}(0) = v \tag{23}$$

on some interval $[0, \epsilon]$. This unique solution is called the *geodesic* through $(\theta, v)$. In particular, for all $v \in T_\theta M$ sufficiently small, $\gamma(1)$ makes sense and is the value of the *exponential map* $\exp_\theta(v)$. The exponential map $\exp_\theta : B_{T_\theta M}(0, R) \to M$ is invertible onto its image $\exp_\theta\left(B_{T_\theta M}(0, R)\right) \subset M$ for all $R > 0$ sufficiently small, and its inverse, denoted $\log_\theta : \exp_\theta\left(B_{T_\theta M}(0, R)\right) \to B_{T_\theta M}(0, R)$, is called the *logarithm*. When $M = \mathbb{R}^p$ with the Euclidean metric, $\exp_\theta(v)$ is defined for all $(\theta, v) \in T\mathbb{R}^p$ and is equal to $\theta + v$. The *length* of a geodesic $\gamma$ on $[0, b]$ is given by $\int_0^b \|\dot{\gamma}(t)\| dt$, and the *geodesic distance* $d_M(\theta, \theta')$ between two points $\theta, \theta' \in M$ is the length of the shortest

geodesic connecting $\theta$ to $\theta'$. Associated to any geodesic $\gamma : [0, b] \to M$ and any $t \in [0, b]$ is a unique orthogonal map $\Pi(\gamma)_t : T_{\gamma(0)}M \to T_{\gamma(t)}M$ such that $\Pi(\gamma) : t \mapsto \Pi(\gamma)t$ satisfies the ODE

$$\big(D_M \Pi(\gamma)\big)[\dot{\gamma}] = 0, \qquad \Pi(\gamma)_0 = I_{T_{\gamma(0)}M}. \tag{24}$$

This map $\Pi(\gamma)_t$ is called the *parallel transport* morphism, and extends to a map $T_{\gamma(0)}M^{\otimes k} \otimes T_{\gamma(0)}^* M^{\otimes l} \to T_{\gamma(t)}M^{\otimes k} \otimes T_{\gamma(t)}^* M^{\otimes l}$ of arbitrary tensors. In particular, if $\theta, \theta' \in M$ admit a unique geodesic $\gamma : [0, b] \to M$ with $\gamma(0) = \theta$ and $\gamma(b) = \theta'$, $\Pi(\gamma)_b : T_\theta M \to T_{\theta'} M$ will be denoted simply $\Pi_{\theta \to \theta'}$.

Tensor fields admit *covariant Taylor expansions* on $M$ defined as follows. Fix $\theta \in M$ and suppose that $\theta' \in M$ is sufficiently close to $\theta$ that there is a unique geodesic $\gamma : [0, 1] \to M$ such that $\gamma(0) = \theta$ and $\gamma(1) = \theta'$. Consider then the function $\widetilde{T}(t) := \Pi(\gamma)_t^{-1} T(\gamma(t)) \in T_\theta M^{\otimes k} \otimes T_\theta^* M^{\otimes l}$. Taylor's theorem in a single variable implies that

$$\widetilde{T}(1) = \widetilde{T}(0) + D\widetilde{T}(0) + \frac{1}{2} D^2 \widetilde{T}(0) + \dots. \tag{25}$$

Since $\big(D_M \Pi(\gamma)\big)[\dot{\gamma}] = 0$ and $(D_M \dot{\gamma})[\dot{\gamma}] = 0$, one has:

$$D^k \widetilde{T}(0) = D^k (\Pi(\gamma)^{-1} \circ T \circ \gamma)(t)|_{t=0} \tag{26}$$

$$= \big(\Pi(\gamma)_t^{-1} D_M^k T(\gamma(t))[\dot{\gamma}(t)^{\otimes k}]\big)|_{t=0} \tag{27}$$

$$= D_M^k T(\theta)[\log_\theta(\theta')^{\otimes k}]. \tag{28}$$

Substituting these formulae into (25) yields the *covariant Taylor expansion*

$$T(\theta') = \Pi_{\theta \to \theta'} \left( T(\theta) + D_M T(\theta)[\log_\theta(\theta')] + \frac{1}{2} D_M^2 T(\theta)[\log_\theta(\theta')^{\otimes 2}] + O\big(\|\log_\theta(\theta')\|^3\big) \right). \tag{29}$$

## A.2 Riemannian geometry of submanifolds and hypersurfaces in Euclidean space

When $M$ is a hypersurface Euclidean space $\mathbb{R}^p$, with normal vector field $n$, the Hessian of a smooth function takes the following form.

**Lemma A.1.** *Let $M$ be a smooth hypersurface in $\mathbb{R}^p$ with normal vector field $n$. The Riemannian Hessian $\nabla_M^2 \lambda$ of a smooth map $\lambda : \mathbb{R}^p \to \mathbb{R}$ along $M$ is the $p \times p$ matrix-valued function*

$$P_{TM}(\nabla^2 \lambda - \langle n, \nabla \lambda \rangle Dn) P_{TM}, \tag{30}$$

*on $M$.*

*Proof.* When $M$ is an embedded submanifold of $\mathbb{R}^p$, the action of the Levi-Civita connection on a function $\lambda$ (i.e., the Riemannian gradient operator) is given by $P_{TM} \nabla \lambda$, where $\nabla$ is the ordinary Euclidean gradient. The action of the Levi-Civita connection on a vector field $X : \mathbb{R}^p \to \mathbb{R}^p$ is given by $P_{TM} DX P_{TM}$. Thus:

$$\nabla_M^2 \lambda = P_{TM} D\big(P_{TM} \nabla \lambda\big) P_{TM} = P_{TM}(\nabla^2 \lambda - Dn \langle n, \nabla \lambda \rangle - n((Dn)^T \nabla \lambda)^T - nn^T \nabla^2 \lambda) P_{TM} \tag{31}$$

$$= P_{TM}(P_{TM} \nabla^2 \lambda - \langle n, \nabla \lambda \rangle Dn - n((Dn)^T \nabla \lambda)^T) P_{TM} \tag{32}$$

$$= P_{TM}(\nabla^2 \lambda - \langle n, \nabla \lambda \rangle Dn) P_{TM}, \tag{33}$$

where the final line follows from

$$P_{TM}(nv^T) = (I - nn^T)(nv^T) = nv^T - nv^T = 0 \tag{34}$$

for any vector $v$. $\qquad \square$

We will be particularly interested in the case of a hypersurface $M = f^{-1}\{y\} \subset \mathbb{R}^p$, where $f : \mathbb{R}^p \to \mathbb{R}$ is a smooth submersion, with metric $g$ inherited from the ambient Euclidean space. Specifically, when $n$ is the unit normal vector field defined by

$$n(\theta) := \frac{\nabla f(\theta)}{\|\nabla f(\theta)\|}, \qquad \theta \in M, \tag{35}$$

the Riemannian metric $g$ on $M$ is defined by

$$g(\theta) := P_{TM}(\theta) := I - n(\theta)n(\theta)^T, \qquad \theta \in M \tag{36}$$

where we use $P_{TM}(\theta)$ to denote the projection onto $T_\theta M$. We will be particularly interested in computing the Riemannian Hessian of the function $\lambda := \|\nabla f\|^2$, which admits the following formula entirely in terms of $f$ and its derivatives.

**Lemma A.2.** *For a hypersurface $M := f^{-1}\{y\}$ of Euclidean space defined by a smooth function $f : \mathbb{R}^p \to \mathbb{R}$, the Riemannian Hessian of $\lambda := \|\nabla f\|^2$ is given by*

$$\nabla_M^2 \lambda = 2P_{TM}\big((\nabla^3 f \nabla f + (\nabla^2 f)^2)\big) - \langle n, \nabla^2 f n\rangle \nabla^2 f\big)P_{TM} \tag{37}$$

*where $\nabla^3 f$ is regarded as a map $\mathbb{R}^p \to \mathbb{R}^{p \times p}$, or as the map sending a direction vector $v$ to the directional derivative of the Hessian matrix $\nabla^2 f$ in the direction $v$.*

*Proof.* By Lemma A.1, we have that

$$\nabla_M^2 \lambda = P_{TM}\big(\nabla^2 \lambda - \langle n, \nabla\lambda\rangle Dn\big)P_{TM}, \tag{38}$$

so it remains only to compute the various quantities. The following identities are elementary to verify:

$$\nabla\lambda = 2\nabla^2 f \nabla f, \tag{39}$$

$$\nabla^2\lambda = 2\big(\nabla^3 f \nabla f + (\nabla^2 f)^2\big), \tag{40}$$

and

$$Dn = P_{TM}\frac{\nabla^2 f}{\|\nabla f\|}. \tag{41}$$

The result now follows. $\qquad\square$

Our next lemma allows us to lower-bound the *convexity radius* $\mathrm{conv}(\theta)$ at $\theta \in M$ which is the largest $r > 0$ such that the geodesic ball $B_M(\theta, r)$ is geodesically convex. This will be used in verifying Assumption 3.4 for multilayer scalar factorisation. If $M$ is a hypersurface in $\mathbb{R}^p$ with normal vector field $n$, then the *second fundamental form* of $M$ is the map $h : TM \otimes TM \to \mathbb{R}$ defined by

$$h(\theta)[v, w] := -\langle Dn(\theta)[v], w\rangle \tag{42}$$

for all $\theta \in M$ and $v, w \in T_\theta M$.

**Lemma A.3.** *Let $M$ be a smooth hypersurface in $\mathbb{R}^p$ with normal vector field $n$, and fix $\theta_* \in M$. Assume that for all $R > 0$, there exists $C_R > 0$ such that*

$$|h(\theta)[u, v]| \le C_R \tag{43}$$

*for all $\theta \in B_M(\theta_*, 2R)$ and all orthonormal vectors $u, v \in T_\theta M$. Then for all $R > 0$, one has*

$$\mathrm{conv}(\theta_*) \ge \frac{1}{2}\min\left\{\frac{\pi}{C_R}, R\right\}. \tag{44}$$

*Proof.* The bound follows from a well-known relationship between the convexity radius and the *injectivity radius*, which, at a point $\theta \in M$, is the largest $r > 0$ such that $\exp_\theta : B_{T_\theta M}(0, r) \to M$ is injective. We thus first bound the injectivity radius. Fix $R > 0$. By the Gauss equation and the hypothesis, at any point $\theta \in B(\theta_*, 2R)$, the sectional curvature $K(\theta)$ of $M$ satisfies

$$|K(\theta)[u, v]| = \big|h(\theta)[u, u]\, h(\theta)[v, v] - h(\theta)[u, v]^2\big| \le C_R^2 \tag{45}$$

for all unit, orthogonal pairs $u, v \in T_\theta M$. Consequently, by Klingenberg's formula [37, Lemma 6.4.7], the injectivity radius $\mathrm{inj}(\theta)$ at $\theta$, namely the largest value of $r$ such that $\exp_\theta : B_{T_\theta M}(0, r) \to M$ is injective, satisfies

$$\mathrm{inj}(\theta) \ge \min\left\{\frac{\pi}{C_R}, \frac{1}{2}L(\theta)\right\}, \tag{46}$$

where $L(\theta)$ is the length of the shortest geodesic loop based at $\theta$. If $L(\theta) \geq 2R$, then this simply yields inj $\geq \min\{\pi/C_R, R\}$. If $L(\theta) < 2R$, then the shortest geodesic loop $\gamma : [0, L(\gamma)] \to M$ at $\theta$ is contained entirely in $B(\theta_*, 2R)$. Fenchel's theorem then combines with the hypothesis to give

$$C_R \geq \int_0^{L(\theta)} \left| h(\gamma(t))[\dot{\gamma}(t), \dot{\gamma}(t)] \right| dt \geq \frac{2\pi}{L(\gamma)}, \tag{47}$$

so that $(1/)L(\theta) \geq \pi/C_R$. Consequently, one has the bound

$$\inf_{\theta \in B(\theta_*, R)} \text{inj}(\theta) \geq \left\{ \frac{\pi}{C_R}, R \right\} \tag{48}$$

in general.

Now, setting

$$r_R := \frac{1}{2} \min \left\{ \frac{\pi}{C_R}, R \right\}, \tag{49}$$

one sees that the hypotheses of [37, Theorem 6.4.8] hold, implying that $B(\theta_*, r_R)$ is geodesically convex. $\qquad\square$

# B  Case study: deep scalar factorisation

In this section, we verify that all of our assumptions are satisfied for deep scalar factorisation, where $f : \mathbb{R}^p \to \mathbb{R}$ is defined by

$$f(\theta_1, \dots, \theta_p) := \theta_1 \cdots \theta_p, \qquad \theta := (\theta_1, \dots, \theta_p) \in \mathbb{R}^p. \tag{50}$$

For $\theta \in \mathbb{R}^p$ having all entries nonzero, it will be convenient to denote

$$v(\theta) := (\theta_1^{-1}, \dots, \theta_p^{-1}). \tag{51}$$

We first demonstrate that Assumption 3.1 is satisfied.

**Proposition B.1.** *Any $y \neq 0$ is a regular value of $f$. Consequently, Assumption 3.1 is satisfied.*

*Proof.* Since $y \neq 0$, any point $\theta^\| \in f^{-1}\{y\}$ has all coordinates being nonzero. Hence $Df(\theta^\|) = y\,v(\theta^\|) \neq 0$ making $y$ a regular value of $f$. $\qquad\square$

The pre-image manifold $f^{-1}\{y\}$ of any $y \neq 0$ has several connected components. Without loss of generality, we will assume from hereon that $y > 0$ and denote by $M$ the component of $f^{-1}\{y\}$ contained in the positive orthant. We next demonstrate that Assumption 3.2 is satisfied.

**Proposition B.2.** *Set $\theta_*^\| := y^{1/p} 1_p$, where $1_p$ is the vector of 1s. Then the line spanned by the normal vector $n(\theta_*^\|)$ is invariant under gradient descent on $\ell = (1/2)(f - y)^2$ for any $\eta$, so that Assumption 3.2 is satisfied.*

*Proof.* At $\theta_*^\| := y^{1/p} 1_p$, one has $n(\theta_*^\|) = p^{-1/2} 1_p$. For any $\alpha \in \mathbb{R}$, one sees that

$$\nabla \ell(\alpha\, n(\theta_*^\|)) = \left( f(\alpha p^{-1/2} 1_p) - y \right) \nabla f(\alpha p^{-1/2} 1_p) \tag{52}$$

$$= \left( f(\alpha p^{-1/2} 1_p) - y \right) \alpha^{p-1} p^{-(p-1)/2} 1_p \tag{53}$$

is a multiple of $n(\theta_*^\|)$, implying that the span of $n(\theta_*^\|)$ is invariant under gradient descent on $\ell$. $\qquad\square$

We will continue to denote $\theta_*^\| := y^{1/p} 1_p$, which is often referred to in the literature as the "balanced" solution. We next come to Assumption 3.3.

**Proposition B.3.** *Assumption 3.3 holds along $M$. In particular, at $\theta_*^\|$,*

$$c(2/\lambda(\theta_*^\|), \theta_*^\|) = \left( \frac{32}{p^3} \binom{p}{2}^2 - \frac{8}{p^2} \binom{p}{3} \right) y^{-2/p}, \tag{54}$$

*where we use the convention $\binom{p}{3} = 0$ if $p < 3$.*

*Proof.* Given symmetric, multilinear functionals $A, B$ taking $k$ and $l$ inputs respectively, let $A \odot B$ denote their symmetric product:

$$(A \odot B)[v_1, \ldots, v_{k+l}] = \frac{1}{(k+l)!} \sum_{\sigma \in \text{Perm}(k+l)} A[v_{\sigma(1)}, \ldots, v_{\sigma(k)}] B[v_{\sigma(k+1)}, \ldots, v_{\sigma(k+l)}], \quad (55)$$

where $\text{Perm}(k+l)$ is the group of permutations of $k+l$ elements. Since $\ell \equiv (1/2)(f-y)^2$, one has

$$D\ell = (f-y)Df \Rightarrow D\ell|_M = 0, \quad (56)$$

$$D^2\ell = Df \odot Df + (f-y)D^2 f \Rightarrow D^2\ell|_M = Df \odot Df \quad (57)$$

$$D^3\ell = 3D^2 f \odot Df + (f-y)D^3 f \Rightarrow D^3\ell|_M = 3D^2 f \odot Df \quad (58)$$

$$D^4\ell = 4D^3 f \odot Df + 3D^2 f \odot D^2 f + (f-y)D^4 f \Rightarrow D^4\ell|_M = 4D^3 f \odot Df + 3D^2 f \odot D^2 f. \quad (59)$$

One then sees that

$$c(\eta, \theta^{\|}) = \left( \frac{\eta}{2} D^3\ell[n^{\otimes 3}](\theta^{\|}) \right)^2 - \frac{\eta}{6} D^4\ell[n^{\otimes 4}](\theta^{\|}) \quad (60)$$

$$= \left( \frac{3\eta}{2} D^2 f[n^{\otimes 2}] Df[n] \right)^2 (\theta^{\|}) - \frac{\eta}{6} \left( 4D^3 f[n^{\otimes 3}] Df[n] + 3(D^2 f[n^{\otimes 2}])^2 \right)(\theta^{\|}) \quad (61)$$

$$= \left( \frac{9\eta^2}{4} \|Df(\theta^{\|})\|^2 - \frac{\eta}{2} \right) \left( D^2 f(\theta^{\|})[n(\theta^{\|})^{\otimes 2}] \right)^2 - \frac{4\eta}{6} \|Df(\theta^{\|})\| D^3 f(\theta^{\|})[n(\theta^{\|})^{\otimes 3}]. \quad (62)$$

By continuity, it suffices to prove the claim at $\eta = 2/\lambda(\theta^{\|}) = 2/\|Df(\theta^{\|})\|^2$; substituting this into the above yields

$$c(2/\lambda(\theta^{\|}), \theta^{\|}) = \frac{8}{\|Df(\theta^{\|})\|^2} D^2 f(\theta^{\|})[n(\theta^{\|})^{\otimes 2}]^2 - \frac{8}{6\|Df(\theta^{\|})\|} D^3 f(\theta^{\|})[n(\theta^{\|})^{\otimes 3}]. \quad (63)$$

One computes

$$Df_l(\theta^{\|}) = \frac{y}{\theta_l}, \quad (64)$$

$$D^2 f_{l_1 l_2}(\theta^{\|}) = (1 - \delta_{l_1 l_2}) \frac{y}{\theta_{l_1} \theta_{l_2}}, \quad (65)$$

$$D^3 f_{l_1 l_2 l_3}(\theta^{\|}) = \begin{cases} (1 - \delta_{l_1 l_2})(1 - \delta_{l_1 l_3})(1 - \delta_{l_2 l_3}) \frac{y}{\theta_{l_1} \theta_{l_2} \theta_{l_3}} & \text{if } L \geq 3 \\ 0 \text{ if } L = 2 \end{cases}. \quad (66)$$

Consequently,

$$D^2 f(\theta^{\|})[n(\theta^{\|})^{\otimes 2}] = \frac{1}{\|Df(\theta^{\|})\|^2} D^2 f(\theta^{\|})[Df(\theta^{\|})^{\otimes 2}] \quad (67)$$

$$= \frac{1}{\|Df(\theta^{\|})\|^2} \sum_{l_1, l_2} (1 - \delta_{l_1 l_2}) \frac{y}{\theta_{l_1} \theta_{l_2}} \frac{y}{\theta_{l_1}} \frac{y}{\theta_{l_2}} \quad (68)$$

$$= \frac{2}{\|Df(\theta^{\|})\|^2} \sum_{l_1 < l_2} \frac{y^3}{\theta_{l_1}^2 \theta_{l_2}^2} \neq 0. \quad (69)$$

If $p = 2$, then, since $D^3 f \equiv 0$, Assumption 3.3 trivially holds for all $\theta^{\|} \in M$ and all $\eta$ close to $2/\lambda(\theta^{\|})$. Let us assume then that $p \geq 3$. One has:

$$D^3 f(\theta^{\|})[n(\theta^{\|})^{\otimes 3}] = \frac{1}{\|Df(\theta^{\|})\|^3} D^3 f(\theta^{\|})[Df(\theta^{\|})^{\otimes 3}] \quad (70)$$

$$= \frac{1}{\|Df(\theta^{\|})\|^3} \sum_{l_1, l_2, l_3} (1 - \delta_{l_1 l_2})(1 - \delta_{l_1 l_3})(1 - \delta_{l_2 l_3}) \frac{y}{\theta_{l_1} \theta_{l_2} \theta_{l_3}} \frac{y}{\theta_{l_1}} \frac{y}{\theta_{l_2}} \frac{y}{\theta_{l_3}} \quad (71)$$

$$= \frac{6}{\|Df(\theta^{\|})\|^3} \sum_{l_1 < l_2 < l_3} \frac{y^4}{\theta_{l_1}^2 \theta_{l_2}^2 \theta_{l_3}^2}. \quad (72)$$

Thus one has

$$c(2/\lambda(\theta^\|), \theta^\|) = \frac{32y^6}{\|Df(\theta^\|)\|^6}\left(\sum_{l_1<l_2}\frac{1}{\theta_{l_1}^2\theta_{l_2}^2}\right)^2 - \frac{8y^4}{\|Df(\theta^\|)\|^4}\sum_{l_1<l_2<l_3}\frac{1}{\theta_{l_1}^2\theta_{l_2}^2\theta_{l_3}^2}. \tag{73}$$

Substituting

$$\|Df(\theta^\|)\|^2 = \sum_{l=1}^{p}\frac{y^2}{\theta_l^2} \tag{74}$$

then yields

$$c(2/\lambda(\theta^\|), \theta^\|) = 32\left(\sum_{l_1<l_2}\theta_{l_1}^{-2}\theta_{l_2}^{-2}\right)^2\left(\sum_l\theta_l^{-2}\right)^{-3} - 8\left(\sum_{l_1<l_2<l_3}\theta_{l_1}^{-2}\theta_{l_2}^{-2}\theta_{l_3}^{-2}\right)\left(\sum_l\theta_l^{-2}\right)^{-2} \tag{75}$$

$$= \left(\sum_l\theta_l^{-2}\right)^{-3}\left(32\left(\sum_{l_1<l_2}\theta_{l_1}^{-2}\theta_{l_2}^{-2}\right)^2 - 8\left(\sum_{l_1<l_2<l_3}\theta_{l_1}^{-2}\theta_{l_2}^{-2}\theta_{l_3}^{-2}\right)\left(\sum_l\theta_l^{-2}\right)\right) \tag{76}$$

$$= (pS_1(\theta^\|))^{-3}\left(32\binom{p}{2}^2 S_2(\theta^\|)^2 - 8p\binom{p}{3}S_1(\theta^\|)S_3(\theta^\|)\right), \tag{77}$$

where $S_i(\theta^\|)$ are the elementary symmetric means in the variables $\theta_l^{-2}$. Newton's inequality for the elementary symmetric means gives $S_2(\theta^\|)^2 \geq S_1(\theta^\|)S_3(\theta^\|)$, so that $c(2/\lambda(\theta^\|), \theta^\|) > 0$ since $\binom{p}{2}^2 = p^2(p-1)^2/4 > p^2(p-1)(p-2)/6 = p\binom{p}{3}$.

To complete the proof, we evaluate at $\theta_*^\| = y^{1/p}1_p$. One has:

$$\sum_l\theta_l^{-2} = py^{-2/p}, \qquad \sum_{l_1<l_2}\theta_{l_1}^{-2}\theta_{l_2}^{-2} = \binom{p}{2}y^{-4/p}, \qquad \sum_{l_1<l_2<l_3}\theta_{l_1}^{-2}\theta_{l_2}^{-2}\theta_{l_3}^{-3} = \binom{p}{3}y^{-6/p}, \tag{78}$$

from which it follows that

$$c(2/\lambda(\theta_*^\|), \theta_*^\|) = \frac{32}{p^3}\binom{p}{2}^2 y^{-2/p} - \frac{8}{p^2}\binom{p}{3}y^{-2/p}, \tag{79}$$

thus yielding the claimed formula. $\qquad\square$

We finally come to verifying Assumption 3.4. This requires us both to demonstrate the existence of a geodesically convex ball containing $\theta_*^\|$, and to demonstrate that $\lambda$ is geodesically strongly convex in this ball.

We first give an explicit ball about $\theta_*^\|$ which is geodesically convex using Lemma A.3.

**Lemma B.4.** *The geodesic ball of radius*

$$r := \begin{cases} \infty & \text{if } p = 2 \\ y^{1/p}\left(\frac{4\pi+1-\sqrt{16\pi+1}}{16\pi-8}\right) & \text{if } p \geq 3 \end{cases} \tag{80}$$

*centred on $\theta_*^\|$ is geodesically convex.*

*Proof.* That $r$ can be taken to be $\infty$ if $p = 2$ follows from the fact that $M$ in this case is a copy of the real line.

Suppose that $p \geq 3$. Observe that for any $\theta^{\|} = (\theta_1, \ldots, \theta_p) \in M$ and any $u, v \in T_{\theta^{\|}} M$, one has

$$h(\theta^{\|})[u,v] = -\langle Dn(\theta^{\|})[u], v \rangle \tag{81}$$

$$= -\|\nabla f(\theta^{\|})\|^{-1} \langle u, \nabla^2 f(\theta^{\|}) \rangle \tag{82}$$

$$= -\left( \sum_i \theta_i^{-2} \right)^{-\frac{1}{2}} \langle u, \left( n(\theta^{\|}) n(\theta^{\|})^T - \mathrm{diag}((\theta_i^{-2})_{i=1}^p) \right) v \rangle \tag{83}$$

$$= -\left( \sum_i \theta_i^{-2} \right)^{-\frac{1}{2}} \left( \sum_i u_i v_i \theta_i^{-2} \right), \tag{84}$$

since $n(\theta^{\|})^T w = 0$ for any vector $w \in T_{\theta^{\|}} M$. Consequently, for any $R \in [0, (1/2) y^{1/p})$, since $B_M(\theta_*^{\|}, 2R) \subset B_{\mathbb{R}^p}(\theta_*^{\|}, 2R)$ by the distance minimising property of geodesics, if $\theta^{\|} \in B_M(\theta_*^{\|}, 2R)$ then $\theta_i \in [y^{1/p} - 2R, y^{1/p} + 2R]$ for all $i$, so that

$$|h(\theta^{\|})[u,v]| \leq (y^{1/p} - 2R)^{-2} (y^{1/p} + 2R) =: C_R. \tag{85}$$

By Lemma A.3, for all $R \in [0, (1/2) y^{1/p})$ the ball of radius

$$r_R := \frac{1}{2} \min \left\{ \frac{\pi}{C_R}, R \right\} \tag{86}$$

is geodesically convex; it thus remains only to optimise this $r_R$. Since $R \mapsto R$ is monotonically increasing and $R \mapsto \pi/C_R$ is monotonically decreasing, it suffices to find the first smallest value of $R$ for which $\pi/C_R = R$. This equality gives a quadratic whose solution is

$$R = y^{1/p} \left( \frac{4\pi + 1 - \sqrt{16\pi + 1}}{8\pi - 4} \right), \tag{87}$$

from which the result follows. □

We next use Lemma A.2 to compute the Riemannian Hessian of $\lambda$ along $M$.

**Lemma B.5.** *For $\theta^{\|} = (\theta_1, \ldots, \theta_p) \in M$, define $s_1(\theta^{\|}) := \sum_i \theta_i^{-2}$ and $s_2(\theta^{\|}) := \sum_i \theta_i^{-4}$. Then the function $\lambda$ has Riemannian Hessian*

$$\nabla_M^2 \lambda = \begin{cases} 2y^2 P_{TM} \left( 3\, diag(v^{\odot 4}) - \frac{s_2}{s_1} diag(v^{\odot 2}) \right) P_{TM} & \text{if } p > 2 \\ 2y^2 P_{TM} \left( diag(v^{\odot 4}) + \left( s_1 - \frac{s_2}{s_1} \right) diag(v^{\odot 2}) \right) P_{TM} & \text{if } p = 2 \end{cases}. \tag{88}$$

*In particular, at $\theta_*^{\|} \in U$,*

$$\nabla_M^2 \lambda(\theta_*^{\|}) = 4y^{2-4/p} I_{TM}(\theta_*^{\|}). \tag{89}$$

*Moreover, if $p = 2$, then $\lambda$ is 2-geodesically strongly convex over all of $M$; if $p > 2$, then for any $\delta \in [0, 2 - \sqrt{3})$, $\lambda$ is $2y^{2-4/p}(1+\delta)^{-4}(1-\delta)^{-2} \left( 3(1-\delta)^2 - (1+\delta)^2 \right)$-geodesically strongly convex over the closed geodesic-distance ball $B_M(\theta_*^{\|}, \delta y^{1/p})$ of radius $\delta y^{1/p}$ about $\theta_*^{\|}$ in $M$.*

*Proof.* For notational convenience, set $V_1 := \mathrm{diag}(v^{\odot 2})$ and $V_2 := \mathrm{diag}(v^{\odot 4})$, where $\odot$ denotes the Hadamard (componentwise) product and $v$ is the vector field defined in (51). From the identities (64), (65) and (66), one sees that along $M$ one has

$$\nabla f \equiv y\, v, \qquad \nabla^2 f \equiv y(vv^T - V_1), \qquad n \equiv s_1^{-1/2} v. \tag{90}$$

The pointwise-bilinear map $\nabla^3 f$ is zero if $p = 2$, and if $p > 2$ it acts on a vector $z$ to give the matrix with components

$$(\nabla^3 f[z])_{ij} = y(1 - \delta_{ij}) \sum_k (1 - \delta_{jk})(1 - \delta_{ik}) v_i v_j v_k z_k \tag{91}$$

$$= y(1 - \delta_{ij}) \Big( \sum_k (v_i v_j v_k z_k - \delta_{jk} v_i v_j v_k z_k - \delta_{ik} v_i v_j v_k z_k + \delta_{jk} \delta_{ik} v_i v_j v_k z_k) \Big) \tag{92}$$

$$= y(1 - \delta_{ij}) \big( (v^T z) v v^T - v(V_1 z)^T - (V_1 z) v^T + \mathrm{diag}(v^{\odot 3} \odot z) \big)_{ij} \tag{93}$$

$$= y \big( (v^T z) v v^T - v(V_1 z)^T - (V_1 z) v^T + \mathrm{diag}(v^{\odot 3} \odot z) + \tag{94}$$

$$- (v^T z) V_1 + \mathrm{diag}(v^{\odot 3} \odot z) + \mathrm{diag}(v^{\odot 3} \odot z) - \mathrm{diag}(v^{\odot 3} \odot z) \big)_{ij} \tag{95}$$

$$= y \big( (v^T z)(v v^T - V_1) - v(V_1 z)^T - (V_1 z) v^T + 2\mathrm{diag}(v^{\odot 3} \odot z) \big)_{ij}, \tag{96}$$

where the fourth line follows from the fact that $(1 - \delta_{ij}) A_{ij} = (A - \mathrm{diag}(A))_{ij}$ for any matrix $A$ and the fact that $\mathrm{diag}(v(V_1 z)^T) = \mathrm{diag}((V_1 z) v^T) = \mathrm{diag}(v^{\odot 3} \odot z)$. Substituting $z = \nabla f = y v$ one sees that

$$\nabla^3 f \nabla f = \begin{cases} y^2 \big( s_1(v v^T - V_1) - v v^T V_1 - V_1 v v^T + 2V_2 \big) & \text{if } p > 2 \\ 0 & \text{if } p = 2 \end{cases}. \tag{97}$$

$$(\nabla^2 f)^2 = y^2 (s_1 v v^T - v v^T V_1 - V_1 v v^T + V_2) \tag{98}$$

and

$$\langle n, \nabla^2 f\, n \rangle \nabla^2 f = \frac{y^2}{s_1} \langle v, v v^T v - V_1 v \rangle (v v^T - V_1) = y^2 (s_1 - s_2/s_1)(v v^T - V_1), \tag{99}$$

so that, by Lemma A.2,

$$\nabla_M^2 \lambda = 2P_{TM} \big( \nabla^3 f \nabla f + (\nabla^2 f)^2 - \langle n, \nabla^2 f \rangle \nabla^2 f \big) P_{TM} \tag{100}$$

$$= 2y^2 P_{TM} \big( 3V_2 - (s_2/s_1) V_1 + (s_1 + (s_2/s_1)) v v^T - 2v v^T V_1 - 2V_1 v v^T \big) P_{TM} \tag{101}$$

if $p > 2$ and

$$\nabla_M^2 \lambda = 2y^2 P_{TM} \big( V_2 + (s_1 - (s_2/s_1)) V_1 - v v^T V_1 - V_1 v v^T + (s_2/s_1) v v^T \big) P_{TM} \tag{102}$$

if $p = 2$. Now, let $\tau$ be any tangent vector field to $M$. Since $v$ is normal to $M$ one has $v^T \tau = 0$, so that

$$\big( (s_1 + (s_2/s_1)) v v^T - 2v v^T V_1 - 2V_1 v v^T \big) \tau = -2v v^T V_1 \tau \tag{103}$$

and

$$\big( -v v^T V_1 - V_1 v v^T + (s_2/s_1) v v^T \big) \tau = -v v^T V_1 \tau \tag{104}$$

are both multiples of the normal vector $v$; it follows that

$$\nabla_M^2 \lambda = \begin{cases} 2y^2 P_{TM} \big( 3V_2 - (s_2/s_1) V_1 \big) P_{TM} & \text{if } p > 2 \\ 2y^2 P_{TM} \big( V_2 + (s_1 - (s_2/s_1)) V_1 \big) P_{TM} & \text{if } p = 2 \end{cases} \tag{105}$$

as claimed. In either case, evaluating this matrix at $\theta_*^{\|} = y^{1/p} 1_p$ gives the claimed identity

$$\nabla_M^2 \lambda(\theta_*^{\|}) = 4y^{2-4/p} I_{TM}(\theta_*^{\|}). \tag{106}$$

In particular, if $p = 2$, evaluating $\nabla_M^2 \lambda$ on the unit tangent vector field $\hat{\tau} : (\theta_1, \theta_2) \mapsto (-\theta_1, \theta_2)/\sqrt{\theta_1^2 + \theta_2^2}$ yields

$$\big( \hat{\tau}^T \nabla_M^2 \lambda\, \hat{\tau} \big)(\theta^{\|}) = \frac{2y^2}{(\theta_1^2 + \theta_2^2)} \Big( \theta_1^{-2} + \theta_2^{-2} + \frac{4\theta_1^{-2} \theta_2^{-2}}{\theta_1^{-2} + \theta_2^{-2}} \Big) \tag{107}$$

$$= \frac{2y^2}{\theta_1^2 + \theta_2^2} \Big( \frac{\theta_1^2 + \theta_2^2}{\theta_1^2 \theta_2^2} + \frac{4}{\theta_1^2 + \theta_2^2} \Big) = 2 + \frac{8y^2}{(\theta_1^2 + \theta_2^2)^2} \geq 2 \tag{108}$$

for any $\theta^{\|} = (\theta_1, \theta_2) \in M$, so that $\lambda$ is 2-geodesically strongly convex over all of $M$ as claimed.

We now assume that $p > 2$ and come to determining a neighbourhood of $\theta_*^{\|}$ on which $\nabla_M^2 \lambda$ is positive definite. It is clear that a sufficient condition for $\nabla_M^2 \lambda$ to be positive-definite is to have all entries of the diagonal matrix $3V_2 - (s_2/s_1)V_1$ be positive; we will show that this sufficient condition can be guaranteed over a geodesic ball in $M$ centred at $\theta_*^{\|}$. Since the geodesic ball of radius $r$ at a point in $M$ is contained in the Euclidean ball of radius $r$ at the same point by the distance-minimising property of geodesics, it suffices to show that this sufficient condition can be guaranteed over a Euclidean ball centred at $\theta_*^{\|}$.

Fix $\delta \in [0, 1)$: for any $\theta = (\theta_1, \ldots, \theta_p)$ in the Euclidean ball $B_{\mathbb{R}^p}(\theta_*^{\|}, \delta y^{1/p})$, one has $\theta_i \in [(1 - \delta)y^{1/p}, (1 + \delta)y^{1/p}]$. Consequently, for any $\theta \in B_{\mathbb{R}^p}(\theta_*^{\|}, \delta y^{1/p})$ one has

$$\min_i \theta_i^{-2} \geq (1 + \delta)^{-2} y^{-2/p}, \qquad \frac{s_2}{s_1}(\theta) \leq \max_i \theta_i^{-2} \frac{s_1}{s_1}(\theta) \leq (1 - \delta)^{-2} y^{-2/p}. \tag{109}$$

It follows that for all $\theta \in B_{\mathbb{R}^p}(\theta_*^{\|}, \delta y^{1/p})$ one has

$$\lambda_{\min}(3V_2 - (s_2/s_1)V_1)(\theta) \geq \min_i \theta_i^{-2}\big(3\min_i \theta_i^{-2} - (s_2/s_1)(\theta)\big) \tag{110}$$

$$\geq (1 + \delta)^{-2} y^{-2/p}\big(3(1 + \delta)^{-2} y^{-2/p} - (1 - \delta)^{-2} y^{-2/p}\big) \tag{111}$$

$$= (1 + \delta)^{-4}(1 - \delta)^{-2} y^{-4/p}\big(3(1 - \delta)^2 - (1 + \delta)^2\big), \tag{112}$$

which is strictly greater than zero provided that $3(1 - \delta)^2 - (1 + \delta)^2 > 0$, which holds provided that $0 \leq \delta < 2 - \sqrt{3}$. This proves the result. $\qquad\square$

Note that $\lambda$ is *not* globally convex over $M$ in general. For instance, when $p = 3$, along the family $\theta^{\|}(\epsilon) := (\epsilon, 1, y\epsilon^{-1}) \in M$, elementary calculations show that $\nabla_M^2 \lambda(\theta^{\|}(\epsilon))$ exhibits negative eigenvalues as $\epsilon \to 0_+$.

Combining Lemmas B.4 and B.5 one obtains explicit bounds under which Assumption 3.4 holds for multilayer scalar factorisation.

**Proposition B.6.** *If $p = 2$, then $\lambda$ is 2-geodesically strongly convex over the geodesically convex set $M$. If $p \geq 3$, then $\lambda$ is $1.33y^{2-4/p}$-geodesically strongly convex over the geodesically convex ball $B_M(\theta_*^{\|}, 0.15y^{1/p})$. In all cases, one has*

$$\nabla_M^2 \lambda(\theta_*^{\|}) = 4y^{2-4/p} I_{TM}(\theta_*^{\|}). \tag{113}$$

*Proof.* Numerical computation reveals that the radius of Lemma B.4 is lower-bounded by $0.15y^{1/p}$. Substituting $\delta = 0.15$ into the strong convexity bound of Lemma B.5 and rounding down to two decimal places yields the claimed result. $\qquad\square$

We conclude the section by proving that the additional hypothesis required for Theorem 5.2 is satisfied.

**Proposition B.7.** *For any $p \geq 2$, with $\nu = 4y^{2-4/p}$ as in Proposition B.6, one has*

$$\frac{\nu}{c(2/\lambda(\theta_*^{\|}))\lambda(\theta_*^{\|})} \leq \frac{1}{2} < 1, \tag{114}$$

*so that the additional hypothesis of Theorem 5.2 holds.*

*Proof.* Since $\lambda(\theta_*^{\|}) = \|Df(\theta_*^{\|})\|^2 = py^{2-2/p}$, one calculates using Proposition B.3 that

$$c(2/\lambda(\theta_*^{\|}), \theta_*^{\|})\lambda(\theta_*^{\|}) = y^{2-4/p}\left(\frac{32}{p^2}\binom{p}{2}^2 - \frac{8}{p}\binom{p}{3}\right) \tag{115}$$

$$= y^{2-4/p} 8(p - 1)\big(p - 1 - (1/6)(p - 2)\big) \tag{116}$$

$$\geq y^{2-4/p} 8(p - 1)\big(p - 1 - (p - 2)\big) \tag{117}$$

$$\geq 8y^{2-4/p} \tag{118}$$

for any $p \geq 2$. The result follows. $\qquad\square$

## C The normal form of gradient descent

To prove Theorem 4.1 rigorously requires a number of steps. **First**, one must give leading order formulae for the expression of gradient descent in tubular neighbourhood coordinates. It is essential here that the errors in these leading order formulae are appropriately sharpened from their naive estimates using the structure of the tubular neighbourhood map (Lemma C.2) and the orthogonal stability assumption (Lemma C.4); without this sharpening our convergence theorems are impossible. These considerations culminate in an intermediate coordinate expression for gradient descent in Proposition C.5. **Second**, one must invoke Assumption 3.3 to transform the $\perp$-dynamics of Proposition C.5 into the easily-analysed normal form quoted in Theorem 4.1. This is formalised in Theorem C.6 and concludes this section.

Throughout this section, given functions $g, h_1, \ldots, h_s : \mathbb{R}^k \to \mathbb{R}^l$ we will use the notation

$$g = O(h_1, \ldots, h_s) \tag{119}$$

to mean that there exist constants $C_1, \ldots, C_s > 0$ such that

$$\|g(w)\| \leq \sum_{i=1}^{s} C_i \|h_i(w)\| \tag{120}$$

for all $w$ sufficiently close to zero.

In addition to this asymptotic notation, we will continue the notation of the main body of the paper, denoting by $M := f^{-1}\{y\}$ the submanifold of $\mathbb{R}^p$ obtained as the preimage of a regular value $y \in \mathbb{R}$ of an analytic map $f : \mathbb{R}^p \to \mathbb{R}$. We will also denote by $n := \nabla f / \|\nabla f\|$ the unit normal along $M$. Assumptions 3.1, 3.2 and 3.3 are assumed in all of what follows. Assumption 3.4 is *not* needed for this section.

Consider the map $E : M \times \mathbb{R} \to \mathbb{R}^p$ defined by

$$E(\theta^{\|}, \theta^{\perp}) := \theta^{\|} + \theta^{\perp} n(\theta^{\|}). \tag{121}$$

Observe that

$$DE(\theta^{\|}, \theta^{\perp}) = (I_{TM}(\theta^{\|}) + \theta^{\perp} Dn(\theta^{\|}), n(\theta^{\|})), \tag{122}$$

where $I_{TM}(\theta^{\|}) : T_{\theta^{\|}} M \to T_{\theta^{\|}} M$ is the identity map. The derivative $DE(\theta^{\|}, \theta^{\perp})$ acts as the linear map

$$T_{\theta^{\|}} M \oplus T_{\theta^{\perp}} \mathbb{R} \ni (v^{\|}, v^{\perp}) \mapsto v^{\|} + \theta^{\perp} Dn(\theta^{\|})[v^{\|}] + v^{\perp} n(\theta^{\|}) \in T_{E(\theta^{\|}, \theta^{\perp})} \mathbb{R}^p \tag{123}$$

between tangent spaces, and can equivalently be written in terms of the *shape operator* $S(\theta^{\|})$ : $T_{\theta^{\|}} M \ni v \mapsto -Dn(\theta^{\|})[v] \in T_{\theta} M$ as

$$DE(\theta^{\|}, \theta^{\perp}) = (I_{TM}(\theta^{\|}) - \theta^{\perp} S(\theta^{\|}), n(\theta^{\|})). \tag{124}$$

Observe that since $n(\theta^{\|})$ is orthogonal to $T_{\theta^{\|}} M$, $DE(\theta^{\|}, \theta^{\perp})$ is invertible whenever $|\theta^{\perp}| < \|S(\theta^{\|})\|^{-1}$, where $\|S(\theta^{\|})\|$ denotes the operator norm of $S(\theta^{\|}) : T_{\theta^{\|}} M \to T_{\theta^{\|}} M$ with respect to the Riemannian metric. Applying the inverse function theorem proves the following.

**Proposition C.1.** *The map $E : M \times \mathbb{R} \to \mathbb{R}^p$ is invertible on an open neighbourhood $U$ of $M \times \{0\}$. The image of $U$ in $\mathbb{R}^p$ is a tubular neighbourhood $N$ of $M$, and we denote the inverse of $E$ thereon by $E^{-1}$.*

The coordinates $E : (\theta^{\|}, \theta^{\perp}) \mapsto E(\theta^{\|}, \theta^{\perp})$ are called *tubular neighbourhood coordinates*. The next step is to derive a formula for $E^{-1}$ and thereafter derive a formula for the conjugate of gradient descent in the coordinates $(\theta^{\|}, \theta^{\perp})$ induced by $E$. The following technical lemma will be necessary to get the correct asymptotics for the error terms in our formula for $E^{-1}$.

**Lemma C.2.** *Let $U \subset \mathbb{R}^{m_1} \times \mathbb{R}^{m_2}$ be an open set and let $g : U \ni (x, y) \mapsto g(x, y) \in \mathbb{R}^p$ be a $C^3$ function. Assume that $D_y^k g \equiv 0$ on $U$ for $k = 2, 3$ and that $g$ is invertible on $U$. Then for any $(x, y) \in U$ and all $v \in \mathbb{R}^p$ sufficiently small, setting $(v_x, v_y) = Dg(x, y)^{-1}[v]$ and $z := g(x, y)$, the point $z + v$ is contained in the domain of $g^{-1}$ and one has*

$$g^{-1}(z + v) = g^{-1}(z) + Dg^{-1}(z)[v] + \frac{1}{2} D^2 g^{-1}(z)[v, v] + O(\|v_x\|^3, \|v_x\|^2 \|v_y\|, \|v_x\| \|v_y\|^2). \tag{125}$$

*Proof.* Since $g$ is invertible on $U$ and $U$ is open, $g(U) = \text{domain}(g^{-1})$ is also open. Fix $(x, y) \in U$ and set $z := g(x, y)$. Then since $g(U)$ is open, $z + v \in g(U)$ for all sufficiently small $v$ so that $g^{-1}(z + v)$ makes sense. Moreover, for all such $v$, there exists $u \in \mathbb{R}^{m_1} \times \mathbb{R}^{m_2}$ such that $g((x, y) + u) = z + v$ so that $g^{-1}(z + v) = (x, y) + u = g^{-1}(z) + u$. Hence one has the following formula for the remainder $R(v)$ of the Taylor expansion of $g^{-1}(z + v)$ about $z$:

$$R(v) := g^{-1}(z + v) - g^{-1}(z) - Dg^{-1}(z)[v] - \frac{1}{2}D^2 g^{-1}(z)[v, v] \tag{126}$$

$$= u - Dg^{-1}(z)[v] - \frac{1}{2}D^2 g^{-1}(z)[v, v]. \tag{127}$$

To bound $R(v)$, therefore, we seek a formula $u(v)$ for $u$ in terms of $v$.

From $g((x, y) + u) = z + v$, one obtains

$$v = G(u) := Dg(x, y)[u] + \frac{1}{2}D^2 g(x, y)[u, u] + S(u), \tag{128}$$

where the remainder $\tilde{u} \mapsto S(\tilde{u})$ satisfies $S(\tilde{u}) = O\big(\|\tilde{u}_x\|^3, \|\tilde{u}_x\|^2 \|\tilde{u}_y\|\big)$ since $D_y^k g \equiv 0$ for $k = 2, 3$ by hypothesis. Since $Dg(x, y)$ is invertible by the invertibility of $g$ on $U$, $G$ is smoothly invertible on a neighbourhood $V$ of $0$ by the inverse function theorem. Consider now the second-order approximation

$$u' := Dg(x, y)^{-1}\left(v - \frac{1}{2}D^2 g(x, y)\big[Dg(x, y)^{-1}[v]^{\otimes 2}\big]\right) \tag{129}$$

to $u$. Since $u' = O(\|v\|)$, taking $v$ smaller if necessary we are guaranteed that $u' \in V$. Thus there is $C > 0$ such that

$$\|u' - u\| \leq C\|G(u') - G(u)\| = C\|G(u') - v\|. \tag{130}$$

We will show that the right side of (130) admits a sufficiently tight bound $b(v)$ that $u(v) = u'(v) + O(b(v))$ suffices to give the desired decay in $R(v)$.

For notational convenience, set $w := (1/2)\,Dg(x, y)^{-1}D^2 g(x, y)\big[Dg(x, y)^{-1}[v]^{\otimes 2}\big]$. Then $u' = Dg(x, y)^{-1}[v] - w$, and substituting this expression into (128) yields

$$G(u') - v = -D^2 g(x, y)\big[Dg(x, y)^{-1}[v], w\big] + \frac{1}{2}D^2 g(x, y)[w^{\otimes 2}] + S\left(Dg(x, y)^{-1}[v] - \frac{1}{2}w\right). \tag{131}$$

Since $D_y^2 g \equiv 0$ by hypothesis, one has $w = O(\|v_x\|^2, \|v_x\|\|v_y\|)$ as well as

$$D^2 g(x, y)\big[Dg(x, y)^{-1}[v], w\big] = O(\|v_x\|^3, \|v_x\|^2\|v_y\|, \|v_x\|\|v_y\|^2), \tag{132}$$

$$D^2 g(x, y)[w^{\otimes 2}] = O(\|w\|^2) = O(\|v_x\|^4, \|v_x\|^3\|v_y\|, \|v_x\|^2\|v_y\|^2) \tag{133}$$

and

$$S\left(Dg(x, y)^{-1}[v] - \frac{1}{2}w\right) = O(\|v_x\|^3, \|v_x\|^2\|v_y\|). \tag{134}$$

Thus, by (130),

$$u = u' + O\big(\|v_x\|^3, \|v_x\|^2\|v_y\|, \|v_x\|\|v_y\|^2\big) \tag{135}$$

$$= Df(x, y)^{-1}[v] - \frac{1}{2}Df(x, y)^{-1}D^2 f(x, y)\big[Df(x, y)^{-1}[v]^{\otimes 2}\big] + \tag{136}$$

$$+ O\big(\|v_x\|^3, \|v_x\|^2\|v_y\|, \|v_x\|\|v_y\|^2\big). \tag{137}$$

Finally, applying the chain rule to the equation $g^{-1} \circ g = \text{Identity}$ shows that $Dg^{-1}(z) = Dg(x, y)^{-1}$ and $D^2 g^{-1}(z) = -Dg(x, y)^{-1}D^2 g(x, y)[Dg(x, y)^{-1}, Dg(x, y)^{-1}]$. It thus follows from substituting (137) into (127) that

$$R(v) = u' - Df^{-1}(z)[v] - \frac{1}{2}D^2 f^{-1}(z)[v, v] + O\big(\|v_x\|^3, \|v_x\|^2\|v_y\|, \|v_x\|\|v_y\|^2\big) \tag{138}$$

$$= O\big(\|v_x\|^3, \|v_x\|^2\|v_y\|, \|v_x\|\|v_y\|^2\big) \tag{139}$$

as claimed. $\qquad\square$

A formula can now be given for $E^{-1}$ using a Taylor expansion as follows.

**Proposition C.3.** *For all $\theta^\| \in M$ and all $v = v^\| + v^\perp n(\theta^\|) \in T_{\theta^\|} M \oplus span(n(\theta^\|)) = T_{\theta^\|} \mathbb{R}^p$ sufficiently small, one has*

$$E^{-1}(\theta^\| + v) = \begin{pmatrix} \theta^\| + v^\| + Dn(\theta^\|)[v^\|]v^\perp \\ v^\perp \end{pmatrix} + O(\|v^\|\|^3, \|v^\|\|^2\|v^\perp\|, \|v^\|\|\|v^\perp\|^2). \quad (140)$$

*Proof.* The result follows from a second-order Taylor expansion with remainder:

$$E^{-1}(\theta^\| + v) = E^{-1}(\theta^\|) + DE^{-1}(\theta^\|)[v] + \frac{1}{2} D^2 E^{-1}(\theta^\|)[v, v] + \text{error}(\theta^\|, v). \quad (141)$$

Since the $\theta^\perp$-derivatives of $E(\theta^\|, \theta^\perp) = \theta^\| + \theta^\perp n(\theta^\|)$ vanish for all orders $> 1$, by Lemma C.2 error$(\theta^\|, v)$, satisfies

$$\text{error}(\theta^\|, v) = O(\|v^\|\|^3, \|v^\|\|^2\|v^\perp\|, \|v^\|\|\|v^\perp\|^2). \quad (142)$$

It thus remains only to calculate the leading order terms. One easily computes

$$DE(\theta^\|, \theta^\perp)\left[\begin{pmatrix} v^\| \\ v^\perp \end{pmatrix}\right] = v^\| + \theta^\perp Dn(\theta^\|)[v^\|] + v^\perp n(\theta^\|) \quad (143)$$

$$D^2 E(\theta^\|, \theta^\perp)\left[\begin{pmatrix} v^\| \\ v^\perp \end{pmatrix}^{\otimes 2}\right] = D\left(DF(\theta^\|, \theta^\perp)\left[\begin{pmatrix} v^\| \\ v^\perp \end{pmatrix}\right]\right)\left[\begin{pmatrix} v^\| \\ v^\perp \end{pmatrix}\right] \quad (144)$$

$$= D_{\theta^\|}\left(v^\| + \theta^\perp Dn(\theta^\|)[v^\|] + v^\perp n(\theta^\|)\right)[v^\|] + \quad (145)$$

$$+ D_{\theta^\perp}\left(v^\| + \theta^\perp Dn(\theta^\|)[v^\|] + v^\perp n(\theta^\|)\right)[v^\perp] \quad (146)$$

$$= \theta^\perp D^2 n(\theta^\|)[v^\|, v^\|] + v^\perp Dn(\theta^\|)[v^\|] + v^\perp Dn(\theta^\|)[v^\|] \quad (147)$$

$$= \theta^\perp D^2 n(\theta^\|)[v^\|, v^\|] + 2v^\perp Dn(\theta^\|)[v^\|], \quad (148)$$

for any $(\theta^\|, \theta^\perp) \in M \times \mathbb{R}$ and $(v^\|, v^\perp) \in T_{\theta^\|} M \times \mathbb{R}$. Recalling that $E$ is invertible when restricted to an open neighbourhood $U$ of $M \times \{0\}$ in $M \times \mathbb{R}$, by differentiating the identity $E^{-1} \circ E = \text{Identity}_{U \subset M \times \mathbb{R}}$ one computes

$$DE^{-1}(\theta^\|) = DE^{-1} \circ E(\theta^\|, 0) = (DE(\theta^\|, 0))^{-1} = \begin{pmatrix} P_{TM}(\theta^\|) \\ n(\theta^\|)^T \end{pmatrix}, \quad (149)$$

where $P_{TM}(\theta^\|) : T_{\theta^\|} \mathbb{R}^p \to T_{\theta^\|} M$ is the orthogonal projection, so that

$$DE^{-1}(\theta^\|)[v] = \begin{pmatrix} v^\| \\ v^\perp \end{pmatrix}. \quad (150)$$

Again differentiating the identity $E^{-1} \circ E = \text{Identity}_U$ yields

$$D^2 E^{-1}(\theta^\|) = D^2 E^{-1} \circ E(\theta^\|, 0) = -DE(\theta^\|, 0)^{-1} D^2 E(\theta^\|, 0)[DE(\theta^\|, 0)^{-1}, DE(\theta^\|, 0)^{-1}], \quad (151)$$

so that from (148) one obtains

$$D^2 E^{-1}(\theta^\|)[v, v] = \begin{pmatrix} P_{TM}(\theta^\|) \\ n(\theta^\|)^T \end{pmatrix} (2v^\perp Dn(\theta^\|)[v^\|]) = \begin{pmatrix} 2v^\perp Dn(\theta^\|)[v^\|] \\ 0 \end{pmatrix}, \quad (152)$$

with the final equality being a consequence of $n^T Dn = -(Dn)^T n = -n^T Dn = 0$ as follows from differentiating the identity $n^T n = 1$. The result follows. $\square$

The next result characterises the scaling of the $\|$-component of $\nabla \ell$ in a neighbourhood of $\theta_*^\|$. It is essential for obtaining the correct asymptotic decay in the error terms.

**Lemma C.4.** *Assumption 3.2 implies that for all $\theta^\| \in M$ sufficiently close to $\theta_*^\|$ and all $\theta^\perp \in \mathbb{R}$ sufficiently small, one has $P_{TM}(\theta^\|)\nabla\ell\left(E(\theta^\|, \theta^\perp)\right) = O(d_M(\theta^\|, \theta_*^\|))$, where $P_{TM}(\theta^\|) : T_{\theta^\|} \mathbb{R}^p \to T_{\theta^\|} M$ is the orthogonal projection onto $T_{\theta^\|} M$.*

*Proof.* By Assumption 3.2, one has

$$P_{TM}(\theta_*^{\parallel})\nabla\ell\big(E(\theta_*^{\parallel},\theta^{\perp})\big) = 0 \tag{153}$$

for all $\theta^{\perp} \in \mathbb{R}$ sufficiently small. Since for all sufficiently small $\theta^{\perp}$ the map $\theta^{\parallel} \mapsto P_{TM}(\theta^{\parallel})\nabla\ell\big(E(\theta^{\parallel},\theta^{\perp})\big)$ is smooth, the result follows. $\qquad\square$

We are now able to give the tubular neighbourhood coordinate expression for gradient descent.

**Proposition C.5.** *In the tubular neighbourhood coordinates* $E : (\theta^{\parallel},\theta^{\perp}) \mapsto E(\theta^{\parallel},\theta^{\perp})$ *for a tubular neighbourhood of* $M$, *the gradient descent map* $\mathrm{GD}_{\mathbb{R}^p}^{\ell}(\eta,\cdot) : \theta \mapsto \theta - \eta\nabla\ell(\theta)$ *is conjugated to the analytic map*

$$\begin{pmatrix}\theta^{\parallel}\\\theta^{\perp}\end{pmatrix} \mapsto \begin{pmatrix}\theta^{\parallel} - \frac{\eta(\theta^{\perp})^2}{2}\nabla_M\lambda(\theta^{\parallel}) + O\big((\theta^{\perp})^3\min\{1,d_M(\theta^{\parallel},\theta_*^{\parallel})\}\big)\\(1-\eta\lambda(\theta^{\parallel}))\theta^{\perp} - \frac{\eta}{2}D^3\ell[n^{\otimes 3}](\theta^{\parallel})(\theta^{\perp})^2 - \frac{\eta}{6}D^4\ell[n^{\otimes 4}](\theta^{\parallel})(\theta^{\perp})^3 + O\big((\theta^{\perp})^4\big)\end{pmatrix} \tag{154}$$

*Proof.* Since $f$ is analytic by Assumption 3.1, so too are $\mathrm{GD}_{\mathbb{R}^p}^{\ell}(\eta,\cdot)$ and $E$ and hence, since the inverse function theorem preserves analyticity, so too is $E^{-1}$. Hence the conjugate $E^{-1}\circ\mathrm{GD}_{\mathbb{R}^p}^{\ell}(\eta,\cdot)\circ E$ is analytic wherever defined. Recalling the open neighbourhood $U$ of $M\times\{0\}$ in $M\times\mathbb{R}$ from Proposition C.1 on which $E$ is a diffeomorphism, given $(\theta^{\parallel},\theta^{\perp})\in U$

$$\mathrm{GD}_{\mathbb{R}^p}^{\ell}\big(\eta, E(\theta^{\parallel},\theta^{\perp})\big) = E(\theta^{\parallel},\theta^{\perp}) - \eta\nabla\ell(E(\theta^{\parallel},\theta^{\perp})) \tag{155}$$

$$= \theta^{\parallel} + \theta^{\perp}n(\theta^{\parallel}) - \eta\nabla\ell(\theta^{\parallel} + \theta^{\perp}n(\theta^{\parallel})). \tag{156}$$

Applying Proposition C.3 with $v = \theta^{\perp}n(\theta^{\parallel}) - \eta\nabla\ell(\theta^{\parallel} + \theta^{\perp}n(\theta^{\parallel}))$, one then obtains

$$E^{-1}\big(\mathrm{GD}_{\mathbb{R}^p}^{\ell}\big(\eta, E(\theta^{\parallel},\theta^{\perp})\big)\big) = \begin{pmatrix}\theta^{\parallel} + v^{\parallel} + O(\|v^{\parallel}\|^3, \|v^{\parallel}\|\|v^{\perp}\|)\\v^{\perp} + O(\|v^{\parallel}\|^3, \|v^{\parallel}\|^2\|v^{\perp}\|, \|v^{\parallel}\|\|v^{\perp}\|^2)\end{pmatrix}. \tag{157}$$

Note the crucial role played by the estimate on the error in Proposition C.3 here: were there to be a pure $O(\|v^{\perp}\|^k)$ term in the error for some $k \geq 1$, then the same error term would have appeared in the the formula (157) and, as we shall now see, would have prevented the $\parallel$-component of the dynamics from having an $O(d_M(\theta^{\parallel},\theta_*^{\parallel}))$ error term.

We now compute formulae for $v^{\parallel}$ and $v^{\perp}$. Using the fact that $\nabla\ell(\theta^{\parallel}) = 0$, one has the Taylor expansion

$$\nabla\ell(E(\theta^{\parallel},\theta^{\perp})) = \theta^{\perp}\nabla^2\ell[n](\theta^{\parallel}) + \frac{(\theta^{\perp})^2}{2}\nabla^3\ell[n^{\otimes 2}](\theta^{\parallel}) + \frac{(\theta^{\perp})^3}{6}\nabla^4\ell[n^{\otimes 3}](\theta^{\parallel}) + O\big((\theta^{\perp})^4\big) \tag{158}$$

$$= \theta^{\perp}\lambda\, n(\theta^{\parallel}) + \frac{(\theta^{\perp})^2}{2}\nabla^3\ell[n^{\otimes 2}](\theta^{\parallel}) + \frac{(\theta^{\perp})^3}{6}\nabla^4\ell[n^{\otimes 3}](\theta^{\parallel}) + O\big((\theta^{\perp})^4\big), \tag{159}$$

where the second line follows from the fact that $n(\theta^{\parallel})$ is an eigenvector of $\nabla^2\ell(\theta^{\parallel})$ with eigenvalue $\lambda(\theta^{\parallel})$. Taking the inner-product of (159) with $n(\theta^{\parallel})$ then reveals that

$$v^{\perp} = \langle n(\theta^{\parallel}), v\rangle \tag{160}$$

$$= (1-\eta\lambda(\theta^{\parallel}))\theta^{\perp} - \frac{\eta}{2}D^3\ell[n^{\otimes 3}](\theta^{\parallel})(\theta^{\perp})^2 - \frac{\eta}{6}D^4\ell[n^{\otimes 4}](\theta^{\parallel})(\theta^{\perp})^3 + O\big((\theta^{\perp})^4\big). \tag{161}$$

One obtains $v^{\parallel}$ by projecting (159) onto $T_{\theta^{\parallel}}M$:

$$v^{\parallel} = P_{TM}(\theta^{\parallel})v = -\eta P_{TM}(\theta^{\parallel})\nabla\ell(E(\theta^{\parallel},\theta^{\perp})) \tag{162}$$

$$= -\frac{\eta(\theta^{\perp})^2}{2}P_{TM}\nabla^3\ell[n^{\otimes 2}](\theta^{\parallel}) + O\big((\theta^{\perp})^3\min\{1,d_M(\theta^{\parallel},\theta_*^{\parallel})\}\big) \tag{163}$$

$$= -\frac{\eta(\theta^{\perp})^2}{2}\nabla_M\big(\nabla^2\ell[n^{\otimes 2}]\big)(\theta^{\parallel}) + O\big((\theta^{\perp})^3\min\{1,d_M(\theta^{\parallel},\theta_*^{\parallel})\}\big) \tag{164}$$

$$= -\frac{\eta(\theta^{\perp})^2}{2}\nabla_M\lambda(\theta^{\parallel}) + O\big((\theta^{\perp})^3\min\{1,d_M(\theta^{\parallel},\theta_*^{\parallel})\}\big), \tag{165}$$

where the second line follows from Lemma C.4, the third line follows from the definition $\nabla_M = P_{TM}\nabla$ of the Levi-Civita connection on $M \subset \mathbb{R}^p$, and the final line follows from the identity $\nabla^2\ell[n,n] \equiv \lambda$ that holds along $M$. Substituting these expressions into (157) yields the result. $\quad\square$

By finally invoking Assumption 3.3, we may now transform the tubular neighbourhood coordinate expression of gradient descent from Proposition C.5 into the normal form expression of Theorem 4.1.

**Theorem C.6.** *Recall the function $c : \mathbb{R} \times M \to \mathbb{R}_{>0}$*

$$c(\eta, \theta^\|) := \left(\frac{\eta}{2}D^3\ell[n^{\otimes 3}](\theta^\|)\right)^2 - \frac{\eta}{6}D^4\ell[n^{\otimes 4}](\theta^\|) > 0 \tag{166}$$

*from Assumption 3.3. There is an open neighbourhood $W$ of $\{(2/\lambda(\theta^\|), \theta^\|, 0) : \theta^\| \in M\}$ in $\mathbb{R} \times M \times \mathbb{R}$ on which $\varphi : W \to \mathbb{R} \times M \times \mathbb{R}$ defined by*

$$\varphi(\eta, \theta^\|, \theta^\perp) = \left(\varphi^\eta(\eta, \theta^\|, \theta^\perp), \varphi^\|(\eta, \theta^\|, \theta^\perp), \varphi^\perp(\eta, \theta^\|, \theta^\perp)\right) \tag{167}$$

$$:= \left(\eta, \theta^\|, \frac{\theta^\perp}{c(\eta, \theta^\|)^{1/2}} - \frac{\frac{\eta}{2}D^3\ell[n^{\otimes 3}]}{(1 - \eta\lambda(\theta^\|))^2 - (1 - \eta\lambda(\theta^\|))}\frac{(\theta^\perp)^2}{c(\eta, \theta^\|)}\right) \tag{168}$$

*is an analytic diffeomorphism onto its image. Moreover, the composite $(I_\mathbb{R} \times E) \circ \varphi : W \to \mathbb{R}^p \times \mathbb{R}$, where $I_\mathbb{R}$ is the identity map, conjugates $(I_\mathbb{R}, \mathrm{GD}^\ell_{\mathbb{R}^p}) : \mathbb{R} \times \mathbb{R}^p \to \mathbb{R} \times \mathbb{R}^p$ to the map*

$$\begin{pmatrix} \eta \\ \theta^\| \\ \theta^\perp \end{pmatrix} \mapsto \begin{pmatrix} \eta \\ \mathrm{GD}^\lambda_M\left(\frac{\eta(\theta^\perp)^2}{2c(\eta,\theta^\|)}, \theta^\|\right) + O\left((\theta^\perp)^3\min\{1, d_M(\theta^\|, \theta^\|_*)\}\right) \\ (1 - \eta\lambda(\theta^\|))\theta^\perp + (\theta^\perp)^3 + O\left((\theta^\perp)^4\right) \end{pmatrix}. \tag{169}$$

*Proof.* That an open neighbourhood $W$ of $\{(2/\lambda(\theta^\|), \theta^\|, 0) : \theta^\| \in M\}$ exists on which $\varphi$ is a diffeomorphism follows from the fact that $c(\eta, \theta^\|)^{-1/2} \neq 0$ for all $\theta^\| \in M$ and $\eta$ in a neighbourhood of $2/\lambda(\theta^\|)$ and the inverse function theorem. That $\varphi$ is analytic is a consequence of the analyticity of $f$ and the functions used in defining $\varphi$ in terms of $\ell = (1/2)(f - y)^2$.

The formula (169) pertains to the conjugation

$$\varphi^{-1} \circ (I_\mathbb{R} \times E^{-1}) \circ \mathrm{GD}^\ell_{\mathbb{R}^p} \circ (I_\mathbb{R} \times E) \circ \varphi \tag{170}$$

of $\mathrm{GD}^\ell_{\mathbb{R}^p}$. Proposition C.5 already gives us a formula for $(I_\mathbb{R} \times E^{-1}) \circ \mathrm{GD}^\ell_{\mathbb{R}^p} \circ (I_\mathbb{R} \times E)$, so it suffices merely to conjugate this formula by $\varphi$.

Since $\varphi^\|(\eta, \theta^\|, \theta^\perp) = \theta^\|$, the $\|$-component of (170) is given by substituting $\varphi^\perp(\eta, \theta^\|, \theta^\perp)$ in place of $\theta^\perp$ in the $\|$-component of (154), which yields the map

$$\begin{pmatrix} \eta \\ \theta^\| \\ \theta^\perp \end{pmatrix} \mapsto \theta^\| - \frac{\eta(\theta^\perp)^2}{2c(\eta,\theta^\|)}\nabla_M\lambda(\theta^\|) + O\left((\theta^\perp)^3\min\{1, d_M(\theta^\|, \theta^\|_*)\}\right). \tag{171}$$

As demonstrated in [33], however, one has

$$\exp_{\theta^\|}\left(-\frac{\eta(\theta^\perp)^2}{2c(\eta,\theta^\|)}\nabla_M\lambda(\theta^\|)\right) = \theta^\| - \frac{\eta(\theta^\perp)^2}{2c(\eta,\theta^\|)}\nabla_M\lambda(\theta^\|) + O\left((\theta^\perp)^4\|\nabla_M\lambda(\theta^\|)\|^2\right) \tag{172}$$

$$= \theta^\| - \frac{\eta(\theta^\perp)^2}{2c(\eta,\theta^\|)}\nabla_M\lambda(\theta^\|) + O\left((\theta^\perp)^4\min\{1, d_M(\theta^\|, \theta^\|_*)^2\}\right) \tag{173}$$

since $\nabla_M\lambda(\theta^\|_*) = 0$. Thus the $\|$-component of (170) may be written as

$$\begin{pmatrix} \eta \\ \theta^\| \\ \theta^\perp \end{pmatrix} \mapsto \mathrm{GD}^\lambda_M\left(\frac{\eta(\theta^\perp)^2}{2c(\eta,\theta^\|)}, \theta^\|\right) + O\left((\theta^\perp)^3\min\{1, d_M(\theta^\|, \theta^\|_*)\}\right) \tag{174}$$

as claimed.

Finally, the $\perp$-component of (169) follows from the same argument used in [27, Theorem 4.3]. $\quad\square$

# D   Convergence theorems

The purpose of this section is to provide proofs of the main convergence theorems presented in the paper. In addition to assuming Assumptions 3.1, 3.2 and 3.3 as in the previous section, in this section we will also assume Assumption 3.4, which states that there is a geodesically convex ball $B_M(\theta_*^{\|}, r)$ centred on $\theta_*^{\|}$ over which $\lambda$ is geodesically smooth, geodesically strongly convex, and has unique global minimum at $\theta_*^{\|}$ with Riemannian Hessian at $\theta_*^{\|}$ being a multiple of the identity.

## D.1   Subcritical regime

Our convergence theorem in the subcritical regime has two components. First, it is proved that gradient descent implicitly descends $\lambda(\theta_t^{\|})$ from being initially $> 2/\eta$ to eventually being $< 2/\eta$. For this, it is necessary to have a descent lemma for the perturbed Riemannian gradient descent component of Theorem 4.1 (Lemma D.1) as well as upper and lower bounds on $|\theta_t^{\perp}|$ during this phase (Lemma D.3). Following this transient phase, convergence is easily proved using the fact that $|\theta_t^{\perp}|$ contracts exponentially to zero.

**Lemma D.1** (Descent lemma for perturbed Riemannian gradient descent). *For the $\mu$-geodesically strongly convex, $L$-geodesically smooth function $\lambda$ on the geodesically convex ball $B_M(\theta_*^{\|}, r)$ centred on $\theta_*^{\|}$ in $M$, consider the map*

$$\mathrm{GD}^{\|}(\eta, \theta^{\|}, \theta^{\perp}) = \exp_{\theta^{\|}}\left(\frac{-\eta(\theta^{\perp})^2}{2c(\eta, \theta^{\|})}\nabla_M\lambda(\theta^{\|})\right) + O\big((\theta^{\perp})^3 \min\{1, d_M(\theta^{\|}, \theta_*^{\|})\}\big) \tag{175}$$

*of Theorem C.6. Then for any $\eta > 0$ and all $\theta^{\perp}$ sufficiently close to zero, $\mathrm{GD}^{\|}(\eta, \cdot, \theta^{\perp})$ maps $B_M(\theta_*^{\|}, r)$ into $B_M(\theta_*^{\|}, r)$. Moreover, for any $\eta > 0$, any $\theta^{\|} \in B_M(\theta_*^{\|}, r)$ and all $\theta^{\perp}$ sufficiently close to zero one has*

$$\lambda\big(\mathrm{GD}^{\|}(\eta, \theta^{\|}, \theta^{\perp})\big) - \lambda(\theta_*^{\|}) \leq \left(1 - \frac{\mu\eta(\theta^{\perp})^2}{4c(\eta, \theta^{\|})}\right)(\lambda(\theta^{\|}) - \lambda(\theta_*^{\|})). \tag{176}$$

*Proof.* For $\theta^{\perp}$ sufficiently small, there is unique $\delta(\theta^{\|}, \theta^{\perp}) = O\big((\theta^{\perp})^3 \min\{1, d_M(\theta^{\|}, \theta_*^{\|})\}\big)$ such that

$$\mathrm{GD}^{\|}(\eta, \theta^{\|}, \theta^{\perp}) = \exp_{\theta^{\|}}\left(-\frac{\eta(\theta^{\perp})^2}{2c(\eta, \theta^{\|})}\nabla_M\lambda(\theta^{\|}) + \delta(\theta^{\|}, \theta^{\perp})\right). \tag{177}$$

To see that $\mathrm{GD}^{\|}(\eta, \cdot, \theta^{\perp})$ preserves the ball $B_M(\theta_*^{\|}, r)$, consider the function $\rho(\theta^{\|}) := (1/2)d_M(\theta^{\|}, \theta_*^{\|})^2$. By the Gauss lemma [29, Corollary 6.10], $\nabla_M\rho(\theta^{\|}) = -\log_{\theta^{\|}}(\theta_*^{\|})$. Fixing $\theta^{\|} \in B_M(\theta_*^{\|}, r)$, for any vector $w \in T_{\theta^{\|}}M$ one therefore has

$$\frac{d}{dt}\bigg|_{t=0}\rho\left(\exp_{\theta^{\|}}\big(-t\nabla_M\lambda(\theta^{\|}) + t^{3/2}w\big), \theta_*^{\|}\right) = -\langle\nabla_M\rho(\theta^{\|}), \nabla_M\lambda(\theta^{\|})\rangle \tag{178}$$

$$= \langle\log_{\theta^{\|}}(\theta_*^{\|}), \nabla_M\lambda(\theta^{\|})\rangle \tag{179}$$

$$\leq -2\mu\rho(\theta^{\|}) < 0, \tag{180}$$

where the third line follows from Lemma E.2. Thus, for all $\theta^{\perp}$ sufficiently small, we are assured that $\rho\big(\mathrm{GD}^{\|}(\eta, \theta^{\|}, \theta^{\perp})\big) < \rho(\theta^{\|})$, implying that $\mathrm{GD}^{\|}(\eta, \theta^{\|}, \theta^{\perp}) \in B_M(\theta_*^{\|}, r)$.

We now come to the descent lemma. Since $B_M(\theta_*^\parallel, r)$ is geodesically convex, we may perform a covariant Taylor expansion and invoke the geodesic smoothness of $\lambda$ to obtain

$$\lambda\big(\mathrm{GD}^\parallel(\eta, \theta^\parallel, \theta^\perp)\big) \leq \lambda(\theta^\parallel) + \left\langle \nabla_M \lambda(\theta^\parallel), -\frac{\eta(\theta^\perp)^2}{2c(\eta, \theta^\parallel)} \nabla_M \lambda(\theta^\parallel) + \delta(\theta^\parallel, \theta^\perp) \right\rangle + \quad (181)$$

$$+ \frac{L}{2} \left\| \frac{\eta(\theta^\perp)^2}{2c(\eta, \theta^\parallel)} \nabla_M \lambda(\theta^\parallel) - \delta(\theta^\parallel, \theta^\perp) \right\|^2 \quad (182)$$

$$\leq \lambda(\theta^\parallel) - \frac{\eta(\theta^\perp)^2}{2c(\eta, \theta^\parallel)} \left( 1 - \frac{\eta(\theta^\perp)^2 L}{4c(\eta, \theta^\parallel)} \right) \|\nabla_M \lambda(\theta^\parallel)\|^2 + \quad (183)$$

$$+ \left( 1 - \frac{\eta(\theta^\perp)^2 L}{2c(\eta, \theta^\parallel)} \right) \langle \nabla_M \lambda(\theta^\parallel), \delta(\theta^\parallel, \theta^\perp) \rangle + \frac{L}{2} \|\delta(\theta^\parallel, \theta^\perp)\|^2. \quad (184)$$

From Cauchy-Schwarz and the estimate $0 \leq (\epsilon^{1/2} a - \epsilon^{-1/2} b)^2$ (implying that $ab \leq (1/2)(\epsilon a^2 + \epsilon^{-1} b^2)$) with $\epsilon = \eta(\theta^\perp)^2 / (4c(\eta, \theta^\parallel))$, one obtains

$$\langle \nabla_M \lambda(\theta^\parallel), \delta(\theta^\parallel, \theta^\perp) \rangle \leq \|\nabla_M \lambda(\theta^\parallel)\| \|\delta(\theta^\parallel, \theta^\perp)\| \quad (185)$$

$$\leq \frac{\eta(\theta^\perp)^2}{8c(\eta, \theta^\parallel)} \|\nabla_M \lambda(\theta^\parallel)\|^2 + \frac{2c(\eta, \theta^\parallel)}{\eta(\theta^\perp)^2} \|\delta(\theta^\parallel, \theta^\perp)\|^2. \quad (186)$$

Taking $|\theta^\perp|$ sufficiently small that $\eta(\theta^\perp)^2 L / (4c(\eta, \theta^\parallel)) \leq 1/4$, therefore, one obtains

$$\lambda\big(\mathrm{GD}^\parallel(\eta, \theta^\parallel, \theta^\perp)\big) \leq \lambda(\theta^\parallel) - \frac{\eta(\theta^\perp)^2}{4c(\eta, \theta^\parallel)} \|\nabla_M \lambda(\theta^\parallel)\|^2 + \left( \frac{2c(\eta, \theta^\parallel)}{\eta(\theta^\perp)^2} + \frac{L}{2} \right) \|\delta(\theta^\parallel, \theta^\perp)\|^2 \quad (187)$$

$$= \lambda(\theta^\parallel) - \frac{\eta(\theta^\perp)^2}{4c(\eta, \theta^\parallel)} \|\nabla_M \lambda(\theta^\parallel)\|^2 + O\big((\theta^\perp)^4 \min\{1, d_M(\theta^\parallel, \theta_*^\parallel)^2\big). \quad (188)$$

Subtracting $\lambda(\theta_*^\parallel)$ from both sides and applying geodesic strong convexity ($\|\nabla_M \lambda(\theta^\parallel)\|^2 \geq 2\mu(\lambda(\theta^\parallel) - \lambda(\theta_*^\parallel))$ by Lemma E.3) yields

$$\lambda\big(\mathrm{GD}^\parallel(\eta, \theta^\parallel, \theta^\perp)\big) - \lambda(\theta_*^\parallel) \leq \left( 1 - \frac{\mu\eta(\theta^\perp)^2}{2c(\eta, \theta^\parallel)} \right) (\lambda(\theta^\parallel) - \lambda(\theta_*^\parallel)) + O\big((\theta^\perp)^4 \min\{1, d_M(\theta^\parallel, \theta_*^\parallel)^2\}\big). \quad (189)$$

Finally, invoking geodesic strong convexity once more to estimate $d_M(\theta^\parallel, \theta_*^\parallel)^2 \leq (2/\mu)\big(\lambda(\theta^\parallel) - \lambda(\theta_*^\parallel)\big)$ as in Lemma E.1 and taking $\theta^\perp$ sufficiently small allows one to absorb the remainder term on the left side into the $\lambda(\theta^\parallel) - \lambda(\theta_*^\parallel)$ factor and yields the result.. $\qquad \square$

The next result will be used in giving uniform bounds on the magnitudes of the $\theta^\perp$ iterates.

**Lemma D.2.** *For $\alpha \in \mathbb{R}$, define $f_\alpha : \mathbb{R} \to \mathbb{R}$ by*

$$f_\alpha(x) := -(1 + \alpha)x + x^3 + O(x^4). \quad (190)$$

*For $\alpha_0, \alpha_1 \in \mathbb{R}$, consider the composite $f_{\alpha_1 \alpha_0} := f_{\alpha_1} \circ f_{\alpha_0}$. Then for all $\gamma > 0$ sufficiently small and all $\alpha_0, \alpha_1 \in [0, \gamma]$:*

1. *$f_{\alpha_1 \alpha_0}$ is monotonically increasing on $[-2\sqrt{\gamma}, 2\sqrt{\gamma}]$.*

2. *$f_{\alpha_1 \alpha_0}$ admits the sole fixed points $0$,*

$$\xi_- = -\sqrt{\frac{\alpha_0 + \alpha_1}{2}} + O(\alpha_0 + \alpha_1), \qquad \xi_+ = \sqrt{\frac{\alpha_0 + \alpha_1}{2}} + O(\alpha_0 + \alpha_1) \quad (191)$$

   *in the interval $[-2\sqrt{\gamma}, 2\sqrt{\gamma}]$.*

*Proof.* For $\gamma > 0$ small and all $\alpha_0, \alpha_1 \in [0, \gamma]$, one computes

$$f_{\alpha_1 \alpha_0}(x) = (1 + \alpha_1 + \alpha_0)x - 2x^3 + O(\alpha_0 \alpha_1 x, \alpha_1 x^3, \alpha_0 x^3, x^4), \quad (192)$$

$$Df_{\alpha_1\alpha_0}(x) = 1 + \alpha_1 + \alpha_0 - 6x^2 + O(\alpha_0\alpha_1, \alpha_1 x^2, \alpha_0 x^2, x^3). \tag{193}$$

In particular, for all $|x| \leq 2\sqrt{\gamma}$, one has

$$Df_{\alpha_1\alpha_0}(x) \geq 1 + \alpha_1 + \alpha_0 - 24\gamma + O(\gamma^{3/2}) > 0 \tag{194}$$

for all $\gamma$ sufficiently small, so that $f_{\alpha_1\alpha_0}|_{[-2\sqrt{\gamma}, 2\sqrt{\gamma}]}$ is monotonically increasing. Set $\beta := (\alpha_0 + \alpha_1)/2$ for notational convenience, and denote

$$\Phi(x) := f_{\alpha_1\alpha_0}(x) - x = 2\beta x - 2x^3 + O(\beta^2 x, \beta x^3, x^4). \tag{195}$$

Fixed points of $f_{\alpha_1\alpha_0}$ are precisely roots of $\Phi$, one of which is clearly $x = 0$. Moreover, if $\beta = 0$, then $\Phi(x) = -2x^3 + O(x^4)$ so that in this case $x = 0$ is the only root of $\Phi$ on $[-2\sqrt{\gamma}, 2\sqrt{\gamma}]$ for all $\gamma$ sufficiently small. Suppose now that $\beta > 0$. We will demonstrate the existence of a unique root $\xi_+$ of $\Phi$ in $(0, 2\sqrt{\gamma}]$ with the claimed formula. Since $2\beta x - 2x^3 > 0$ for all $0 < x \leq 2^{-1/2}\sqrt{\beta}$, we see that for all $\gamma$ sufficiently small, $\Phi(x)$ is strictly positive on $(0, 2^{-1/2}\sqrt{\beta}]$ and so cannot admit any roots therein. However, observe that for all $\gamma$ sufficiently small one has

$$D\Phi(x) = 2\beta - 6x^2 + O(\beta^2, \beta x^2, x^3) < 0 \tag{196}$$

for all $2^{-1/2}\sqrt{\beta} \leq x \leq 2\sqrt{\gamma}$, implying that $\Phi$ is monotonically decreasing on $[2^{-1/2}\sqrt{\beta}, 2\sqrt{\gamma}]$, and thus admits *at most* one root therein. Since moreover $\Phi(2^{-1/2}\sqrt{\beta}) > 0$ and $\Phi(2\sqrt{\beta}) = -12\beta^{3/2} + O(\beta^2) < 0$, by the intermediate value theorem $\Phi$ admits *precisely* one root $\xi_+$ in $[2^{-1/2}\sqrt{\beta}, 2\sqrt{\gamma}]$. By the mean value theorem, there is $z$ between $\sqrt{\beta}$ and $\xi_+ \leq 2\sqrt{\beta}$ such that

$$0 = \Phi(\xi_+) = \Phi(\sqrt{\beta}) + D\Phi(z)(\xi_+ - \sqrt{\beta}), \tag{197}$$

so that, since $|D\Phi(z)| \geq 4\beta + O(\beta^{3/2}) = \Omega(\beta)$ and $\Phi(\sqrt{\beta}) = O(\beta^2)$,

$$\xi_+ = \sqrt{\beta} - \frac{\Phi(\sqrt{\beta})}{D\Phi(z)} = \sqrt{\beta} + O(\beta) \tag{198}$$

as claimed. A similar argument can be used to show there exists a unique root $\xi_- = -\sqrt{\beta} + O(\beta)$ of $\Phi$ in $[-2\sqrt{\gamma}, 0)$, thus proving the lemma. $\square$

**Lemma D.3** (Upper and lower bounds for $T^\perp$ iterates). *Let $\{\lambda_t\}_{t\in\mathbb{N}}$ be a monotonically decreasing sequence of numbers. Then for all $\eta > 0$ such that $\eta\lambda_0 > 2$ is sufficiently small, there is a constant $C > 0$ such that for any $\theta_0^\perp$ sufficiently small, the iterates defined by*

$$\theta_{t+1}^\perp = (1 - \eta\lambda_t)\theta_t^\perp + (\theta_t^\perp)^3 + O((\theta_t^\perp)^4) \tag{199}$$

*satisfy the bounds*

$$\frac{|\theta_0^\perp|}{\sqrt{1 + 3(\theta_0^\perp)^2 t}} \leq |\theta_t^\perp| \leq 2\sqrt{\eta\lambda(\theta_0^\|) - 2} \tag{200}$$

*for all $t \in \mathbb{N}$ such that $\eta\lambda_t \geq 2$.*

*Proof.* We prove the upper bound first. For each $s \in \mathbb{N}$ such that $\eta\lambda_s > 2$, denote $\alpha_s := \eta\lambda_s - 2$ and denote by $f_s$ the map

$$f_s : \theta^\perp \mapsto (1 - \eta\lambda_s)\theta^\perp + (\theta^\perp)^3 + O((\theta^\perp)^4) \tag{201}$$
$$= -(1 + \alpha_s)\theta^\perp + (\theta^\perp)^3 + O((\theta^\perp)^4). \tag{202}$$

Supposing that $s \in \mathbb{N}$ is such that both $\eta\lambda_s, \eta\lambda_{s+1} \geq 2$, since $\{\lambda_t\}_{t\in\mathbb{N}}$ is monotonically decreasing Lemma D.2 guarantees that $f_{s+1,s} := f_{s+1} \circ f_s$ admits attractive fixed points $\xi_{s+1,s,\pm} = \pm\sqrt{(\alpha_{s+1} + \alpha_s)/2} + O(\alpha_{s+1} + \alpha_s)$ and is monotonically increasing on $[-2\sqrt{\alpha_0}, 2\sqrt{\alpha_0}]$ provided $\alpha_0 = \eta\lambda_0 - 2$ is sufficiently small. Suppose now that $\theta_0^\perp \in [-2\sqrt{\alpha_0}, 2\sqrt{\alpha_0}] \setminus \{0\}$. The iterates $\theta_t^\perp$ as defined in (199) satisfy

$$\theta_{2t+2}^\perp = f_{2t+1,2t}(\theta_{2t}^\perp), \qquad \theta_{2t+1}^\perp = f_{2t,2t-1}(\theta_{2t-1}^\perp), \qquad t \in \mathbb{N}. \tag{203}$$

Assuming without loss of generality that $\theta_0^\perp > 0$, we will prove by induction that $\theta_{2t}^\perp \leq 2\sqrt{\alpha_0}$ for all $t \in \mathbb{N}$ such that $\alpha_s \geq 0$ for all $s \leq 2t - 1$. Clearly $\theta_0^\perp \leq 2\sqrt{\alpha_0}$ by assumption. Suppose as an inductive hypothesis that $0 < \theta_{2t}^\perp \leq 2\sqrt{\alpha_0}$ and that $\alpha_{2t+1}, \alpha_{2t} \geq 0$. Using the monotonicity of $f_{2t+1,2t}$ from Lemma D.2 and the fact that $\xi_{2t+1,2t,+}$ is its sole fixed point in $(0, 2\sqrt{\alpha_0}]$, either

$$\theta_{2t}^\perp \leq \xi_{2t+1,2t,+} \Rightarrow \theta_{2t}^\perp \leq f_{2t+1,2t}(\theta_{2t}^\perp) = \theta_{2t+2}^\perp \leq \xi_{2t+1,2t,+} \leq 2\sqrt{\alpha_0}, \tag{204}$$

or

$$\theta_{2t}^\perp \geq \xi_{2t+1,2t,+} \Rightarrow \xi_{2t+1,2t,+} \leq f_{2t+1,2t}(\theta_{2t}^\perp) = \theta_{2t+2}^\perp \leq \theta_{2t}^\perp \leq 2\sqrt{\alpha_0} \tag{205}$$

by the inductive hypothesis; in either case $\theta_{2t+2}^\perp \leq 2\sqrt{\alpha_0}$, thus proving the claim. A symmetric argument shows that the odd iterates satisfy $|\theta_{2t+1}^\perp| \leq 2\sqrt{\alpha_0}$ for all $t \in \mathbb{N}$ such that $\alpha_s \geq 0$ for all $s \leq 2t$, thus proving the upper-bound in (200).

We now prove the lower bound. Invoking the upper bound, $\alpha_0$ can be taken sufficiently small such that the recursive estimate

$$|\theta_{t+1}^\perp| = |\theta_t^\perp|(1 + \alpha_t - (\theta_t^\perp)^2 + O((\theta_t^\perp)^3)) \tag{206}$$

$$\geq |\theta_t^\perp|(1 - 2(\theta_t^\perp)^2) \tag{207}$$

holds as long as $\alpha_t \geq 0$. By taking $\alpha_0$ even smaller if necessary, squaring, taking the reciprocal of both sides and applying $1/(1-x)^2 \leq 1 + 3x$ for $x$ sufficiently small one deduces that

$$\frac{1}{(\theta_{t+1}^\perp)^2} \leq \frac{1}{(\theta_t^\perp)^2}(1 + 6(\theta_t^\perp)^2) \Rightarrow \frac{1}{(\theta_{t+1}^\perp)^2} - \frac{1}{(\theta_t^\perp)^2} \leq 6, \tag{208}$$

from which it follows by a telescoping sum that

$$|\theta_t^\perp| \geq \frac{|\theta_0^\perp|}{\sqrt{1 + 6(\theta_0^\perp)^2 t}} \tag{209}$$

for all $t$ such that $\alpha_t \geq 0$. This proves the claim. $\square$

**Theorem D.4.** *Assume that $\eta < 2/\lambda(\theta_*^\|)$, and let $0 < c, C < \infty$ satisfy $c \leq c(\eta, \theta^\|) \leq C$ for all $\theta^\| \in B_M(\theta_*^\|, r)$. Then for all $\theta_0^\perp$ sufficiently close to zero and all $\theta_0^\| \neq \theta_*^\| \in B_M(\theta_*^\|, r)$, if $\eta > 2/\lambda(\theta_0^\|)$ is sufficiently small, then there is $\tau \in \mathbb{N}$, with*

$$\tau \leq \tau(\theta_0^\|, \theta_0^\perp) := \left\lceil \frac{1}{6(\theta_0^\perp)^2} \left( \frac{2/\eta - \lambda(\theta_*^\|)}{\lambda(\theta_0^\|) - \lambda(\theta_*^\|)} \right)^{-\frac{24C}{\mu\eta}} \right\rceil, \tag{210}$$

*such that $\eta < 2/\lambda(\theta_t^\|)$ for all $t \geq \tau$ and $\eta \geq 2/\lambda(\theta_t^\|)$ for all $t < \tau$. Consequently, setting $\beta := 1 - (2 - \eta\lambda(\theta_\tau^\|)) < 1$, the iterates $(\theta_t^\|, \theta_t^\perp)$ converge to a global minimum $(\theta_\infty^\|, 0)$ of $\ell$ in $M$ with suboptimal sharpness*

$$\lambda(\theta_\infty^\|) - \lambda(\theta_*^\|) \geq \exp\left( -\frac{4\eta L^2}{c\mu} \frac{(\theta_\tau^\perp)^2}{1 - \beta^2} \right)(\lambda(\theta_\tau^\|) - \lambda_*). \tag{211}$$

*at a rate $O(\beta^t)$.*

*Proof.* For any $\theta_0^\| \neq \theta_*^\|$, take $\theta_0^\perp \neq 0$ sufficiently close to zero that the hypotheses of Lemma D.1 and Lemma D.3 are satisfied, with $\lambda_0$ in Lemma D.3 taken to be equal to $\lambda(\theta_0^\|)$, and assume that $\eta\lambda_0 > 0$ is sufficiently small as in Lemma D.3. Then we may assume that $\lambda_t := \lambda(\theta_t^\|)$ is monotonically decreasing and the bounds of Lemma D.3 apply to $\theta_t^\perp$. Since $\{\lambda_t\}_{t \in \mathbb{N}}$ is monotonically decreasing, any $\tau$ satisfying $\eta < 2/\lambda(\theta_t^\|)$ for all $t \geq \tau$ and $\eta \geq 2/\lambda(\theta_t^\|)$ for all $t < \tau$ is necessarily unique. We claim that this $\tau$ satisfies the upper bound $\tau \leq \tau(\theta_0^\|, \theta_0^\perp)$. Indeed, suppose for a contradiction that $\tau > \tau(\theta_0^\perp, \theta_0^\|)$. Then for all $t < \tau$, denoting $\lambda_* := \lambda(\theta_*^\|)$, one would have

$$\lambda_{t+1} - \lambda_* \leq \left( 1 - \frac{\mu\eta}{4c(\eta, \theta_t^\|)} \frac{(\theta_0^\perp)^2}{(1 + 6(\theta_0^\perp)^2 t)} \right)(\lambda_t - \lambda_*) \tag{212}$$

by Lemmas D.1 and D.3. Invoking $c(\eta, \theta_t^{\parallel}) \le C$, taking logarithms, applying the upper-bound $\ln(1-x) \le -x$ and applying a telescoping sum then yield

$$\ln(\lambda_t) - \ln(\lambda_0) \le -\frac{\mu\eta}{24C}\sum_{s=0}^{t-1}\frac{1}{\frac{1}{6(\theta_0^{\perp})^2}+s} \tag{213}$$

$$\le -\frac{\mu\eta}{24C}\ln(1+6(\theta_0^{\perp})^2 t) \tag{214}$$

by Lemma E.4, so that

$$\lambda_t \le (\lambda_0 - \lambda_*)(1+6(\theta_0^{\perp})^2 t)^{-\frac{\mu\eta}{24C}} + \lambda_*. \tag{215}$$

However, plugging in $t = \tau(\theta_0^{\perp}, \theta_0^{\parallel})$ then reveals that $\lambda_t < 2/\eta$, thus contradicting the assumption that $\tau > \tau(\theta_0^{\perp}, \theta_0^{\parallel})$ and implying that $\tau \le \tau(\theta_0^{\perp}, \theta_0^{\parallel})$ as claimed.

We now come to the final convergence and sharpness suboptimality claim. For all $t \ge \tau$, Lemma D.1 continues to imply that $\lambda_t < 2/\eta$ decreases monotonically, while $\theta_t^{\perp}$ contracts exponentially according to the estimate

$$|\theta_{t+1}^{\perp}| = |\theta_t^{\perp}||1 + (\eta\lambda_t - 2) - (\theta_t^{\perp})^2 + O(\theta_t^{\perp})^3| \le |\theta_t^{\perp}||1 - (2 - \eta\lambda_t)| \le |\theta_t^{\perp}||1 - (2 - \eta\lambda_{\tau})|. \tag{216}$$

This proves the claim that the iterates $(\theta_t^{\parallel}, \theta_t^{\perp})$ converge to a global minimum $\theta_{\infty}^{\parallel}$ of $\ell$ at rate $O\big((1-(2-\eta\lambda_{\tau}))^t\big)$. We now turn to proving the suboptimality of the sharpness at $\theta_{\infty}^{\parallel}$. The geodesic strong convexity of $\lambda$ in the form of Lemma E.2 implies that

$$\mu d_M(\theta^{\parallel}, \theta_*^{\parallel})^2 \le \langle \Pi_{\theta^{\parallel} \to \theta_*^{\parallel}} \nabla_M \lambda(\theta^{\parallel}), \log_{\theta_*^{\parallel}}(\theta^{\parallel}) \rangle \le \|\nabla_M \lambda(\theta^{\parallel})\| d_M(\theta^{\parallel}, \theta_*^{\parallel}) \tag{217}$$

for any $\theta^{\parallel} \in B_M(\theta_*^{\parallel}, r)$, so that the $\parallel$-update can be written

$$\mathrm{GD}^{\parallel}(\eta, \theta^{\parallel}, \theta^{\perp}) = \exp_{\theta^{\parallel}}\left(-\frac{\eta(\theta^{\perp})^2}{2c(\eta, \theta^{\parallel})}\nabla_M \lambda(\theta^{\parallel}) + O\big((\theta^{\perp})^3 \min\{1, d_M(\theta^{\parallel}, \theta_*^{\parallel})\}\big)\right) \tag{218}$$

$$= \exp_{\theta^{\parallel}}\left(-\frac{\eta(\theta^{\perp})^2}{2c(\eta, \theta^{\parallel})}\nabla_M \lambda(\theta^{\parallel}) + O\big((\theta^{\perp})^3 \min\{1, \|\nabla_M \lambda(\theta^{\parallel})\|\}\big)\right). \tag{219}$$

Consequently, invoking the geodesic strong convexity of $\lambda$ once again in the form of Lemma E.1,

$$\lambda_{t+1} \ge \lambda_t + \langle \nabla_M \lambda(\theta_t^{\parallel}), \log_{\theta_t^{\parallel}}(\theta_{t+1}^{\parallel}) \rangle + \frac{\mu}{2}\|\log_{\theta_t^{\parallel}}(\theta_{t+1}^{\parallel})\|^2 \tag{220}$$

$$= \lambda_t - \left\langle \nabla_M \lambda(\theta_t^{\parallel}), \frac{\eta(\theta_t^{\perp})^2}{2c(\eta, \theta_t^{\parallel})}\nabla_M \lambda(\theta_t^{\parallel}) + O\big((\theta_t^{\perp})^3 \min\{1, \|\nabla_M \lambda(\theta_t^{\parallel})\|\}\big) \right\rangle \tag{221}$$

$$+ \frac{\mu}{2}\left\|\frac{\eta(\theta_t^{\perp})^2}{2c(\eta, \theta_t^{\parallel})}\nabla_M \lambda(\theta_t^{\parallel}) + O\big((\theta_t^{\perp})^3 \min\{1, \|\nabla_M \lambda(\theta_t^{\parallel})\|\}\big)\right\|^2 \tag{222}$$

$$= \lambda_t - \frac{\eta(\theta_t^{\perp})^2}{2c(\eta, \theta_t^{\parallel})}\|\nabla_M \lambda(\theta_t^{\parallel})\|^2(1 + O(\theta_t^{\perp})) \tag{223}$$

$$\ge \lambda_t - \frac{\eta(\theta_t^{\perp})^2}{c(\eta, \theta_t^{\parallel})}\|\nabla_M \lambda(\theta_t^{\parallel})\|^2 \tag{224}$$

for sufficiently small $2\sqrt{\eta\lambda_0 - 2} > |\theta_t^{\perp}|$. Subtracting $\lambda_*$ from both sides and invoking the Lipschitz gradients ($\|\nabla_M \lambda(\theta^{\parallel})\| \le L d_M(\theta^{\parallel}, \theta_*^{\parallel})$) and strong convexity ($(\mu/2)d_M(\theta^{\parallel}, \theta_*^{\parallel})^2 \le \lambda(\theta^{\parallel}) - \lambda_*$ from Lemma E.1) properties of $\lambda$ then yield

$$\lambda_{t+1} - \lambda_* \ge \lambda_t - \lambda_* - \frac{\eta(\theta_t^{\perp})^2}{c(\eta, \theta_t^{\parallel})}L^2 d_M(\theta_t^{\parallel}, \theta_*^{\parallel})^2 \tag{225}$$

$$\ge (\lambda_t - \lambda_*)\left(1 - \frac{2\eta(\theta_t^{\perp})^2 L^2}{c(\eta, \theta_t^{\parallel})\mu}\right) \tag{226}$$

$$\ge (\lambda_t - \lambda_*)\left(1 - \frac{2\eta L^2(\theta_{\tau}^{\perp})^2}{c\mu}\beta^{2t}\right), \tag{227}$$

where in the final line we have invoked $c(\eta, \theta_t^{\|}) \geq c$. It follows that

$$\lambda_t - \lambda_* \geq \prod_{s=\tau}^{t-1} \left(1 - \frac{2\eta L^2 (\theta_\tau^\perp)^2}{c\mu} \beta^{2s}\right)(\lambda_\tau - \lambda_*). \tag{228}$$

Taking $2\sqrt{\eta\lambda_0 - 2} > |\theta_\tau^\perp|$ smaller if necessary and using the bound $\ln(1 - x) \geq -2x$ valid for small $x$ one has

$$\ln\left(\prod_{s=0}^{t-1}\left(1 - \frac{2\eta L^2(\theta_\tau^\perp)^2}{c\mu}\beta^{2s}\right)\right) \geq -\frac{4\eta L^2(\theta_\tau^\perp)^2}{c\mu}\sum_{s=0}^{t-1}\beta^{2s} \tag{229}$$

$$\geq -\frac{4\eta L^2(\theta_\tau^\perp)^2}{c\mu}\frac{1}{1 - \beta^2} \tag{230}$$

from which the result follows. $\qquad\square$

## D.2 Critical regime

The convergence theorem in the critical regime is the most difficult to prove. This is because in this regime, the fixed point to which the iterates converge is merely *parabolic* rather than *hyperbolic* as is (at least asymptotically) true in the subcritical and supercritical regimes. The convergence theorem in the critical regime will follow from a careful analysis of the following parabolic system.

Fix $0 < a < c$ and $b > 0$. Let $A(x, y) = O(x, y)$, $B(x) = O(x)$ and $C(y) = O(y)$ be analytic functions, and consider the analytic functions

$$T_x(x, y) = \left(1 - (a + A(x, y))y^2\right)x, \qquad T_y(x, y) = \left(1 + (b + B(x))x^2 - (c + C(y))y^2\right)y. \tag{231}$$

Denote by $T = (T_x, T_y) : \mathbb{R}^2 \to \mathbb{R}^2$ the corresponding analytic map. Clearly $0$ is a fixed point of $T$; we will be concerned with proving convergence of the iterates of $T$ to this fixed point. The following result guarantees that $T$ admits an invariant manifold approaching the origin. This result is essential, since our final convergence proof is a two-stage proof; first, convergence to a neighbourhood of the invariant manifold is proved, following which, second, convergence to the origin is easy.

**Lemma D.5.** *There exists $r > 0$ and a function $u : \{x > 0 : |x^2 - r| < r\} \to \mathbb{R}$ whose graph $\Gamma_u := \{(x, u(x)) : x \in dom(u)\}$ is invariant under $T$, and for which $u(x) = \sqrt{\frac{b}{c-a}}x + O(x^2)$.*

*Proof.* The analytic map $T : \mathbb{R}^2 \to \mathbb{R}^2$ extends uniquely to a holomorphic map $\tilde{T}$ defined on a neighbourhood of $0$ in $\mathbb{C}^2$. Like $T$, at the origin the map $\tilde{T}$ is the identity to first order, vanishes at order two, and has third order Taylor coefficients given by

$$\partial_{xyy}^3 T_x(0,0) = -2a, \quad \partial_{yxx}^3 T_y(0,0) = 2b, \quad \partial_{yyy}^3 T_y(0,0) = -6c, \tag{232}$$

with all other third order coefficients being zero. Observe that on the vector $v$ given by

$$v = \left(1, \sqrt{\frac{b}{c-a}}\right) \tag{233}$$

one has

$$\partial^3 T(0,0)[v,v,v] = \left(-6a\frac{b}{c-a}, 6b\sqrt{\frac{b}{c-a}} - 6c\sqrt{\frac{b}{c-a}}^3\right) \tag{234}$$

$$= -6a\sqrt{\frac{b}{c-a}}^3 v, \tag{235}$$

so that $v$ is a non-degenerate characteristic direction of $\tilde{T}$ [6, Definition 4.1], which is moreover attracting since $-6a\sqrt{b/(c-a)} < 0$. By [6, Section 6], therefore, there exists $r > 0$ and a holomorphic function $u : \{x \in \mathbb{C} : |x^2 - r| < r\} \to \mathbb{C}$ such that $u(x) = \sqrt{\frac{b}{c-a}}x + O(x^2)$ whose graph is invariant under $T$. Restricting this $u$ to the intersection of its domain with the positive real axis yields the result. $\qquad\square$

The next lemma will be key in establishing convergence to the invariant manifold.

**Lemma D.6.** *Let* $u(x) = \sqrt{\frac{b}{c-a}}x + O(x^2)$ *be the function from Lemma D.5, and define* $\Delta(x,y) :=$ $y - u(x)$. *Then* $\widetilde{\Delta}(x,y) := \Delta(x,y)/x$ *satisfies*

$$|\widetilde{\Delta} \circ T(x,y)| \le |\widetilde{\Delta}(x,y)| \left(1 - \frac{c-a}{2}y^2\right) \tag{236}$$

*for all sufficiently small* $x > 0$, $y \ge 0$.

*Proof.* We estimate $\frac{\widetilde{\Delta} \circ T}{\widetilde{\Delta}}$. Define

$$D(x,y) := \frac{T_x(x,y)}{x}, \qquad N(x,y) := \frac{\Delta \circ T(x,y)}{\Delta(x,y)}, \qquad (x,y) \in \mathrm{dom}(u) \times \mathbb{R}_{\ge 0}. \tag{237}$$

Observe that both $D$ and $N$ are analytic on the domain of $u$. Observe moreover that

$$\frac{\widetilde{\Delta} \circ T(x,y)}{\widetilde{\Delta}(x,y)} = \frac{\Delta \circ T(x,y)}{T_x(x,y)} \cdot \frac{x}{\Delta(x,y)} = \frac{N(x,y)}{D(x,y)} = 1 - \frac{D(x,y) - N(x,y)}{D(x,y)}, \tag{238}$$

so that estimation of $\frac{\widetilde{\Delta} \circ T}{\widetilde{\Delta}}$ is reduced to estimating $\frac{D-N}{D}$. We first estimate $D - N$. For this, observe that since $T(x,0) = (x,0)$, one has $D(x,0) - N(x,0) = 0$ for all $x \in \mathrm{dom}(u)$. Thus, by analyticity of $D$ and $N$, there is a unique analytic function $H$ on $\mathrm{dom}(u) \times \mathbb{R}$ such that

$$D(x,y) - N(x,y) = yH(x,y). \tag{239}$$

We compute $H(x,y)$ to second order. For notational convenience, set

$$\kappa := \sqrt{\frac{b}{c-a}} \tag{240}$$

so that $u(x) = \kappa x + O(x^2)$. Differentiate (239) with respect to $y$ and evaluate at $y = 0$ to obtain

$$H(x,0) = \partial_y D(x,0) - \partial_y N(x,0), \tag{241}$$

Observe that

$$\partial_y D(x,0) = \partial_y(1 - ay^2 + A(x,y)y^2)|_{y=0} = 0 \tag{242}$$

while, from $N = (\Delta \circ T)/\Delta$, one has

$$\partial_y N(x,0) = \frac{\Delta \, \partial_y(\Delta \circ T) - (\Delta \circ T) \, \partial_y \Delta}{\Delta^2}(x,0). \tag{243}$$

From $\Delta(x,y) = y - u(x)$, is easily seen that

$$\Delta(x,0) = -u(x), \qquad \partial_y \Delta(x,0) = 1, \qquad \Delta \circ T(x,0) = -u(T_x(x,0)) = -u(x) \tag{244}$$

while

$$\partial_y(\Delta \circ T)(x,0) = \partial_y T_y(x,0) - u'(x)\partial_y T_x(x,0) = 1 + bx^2 + O(x^3) \tag{245}$$

so that (243) yields

$$H(x,0) = -\partial_y N(x,0) = \frac{bx^2 + O(x^3)}{u(x)} = (c-a)\kappa x + O(x^2). \tag{246}$$

To compute the second order component of $H$, differentiate (239) twice with respect to $y$ and evaluate at $y = 0$ to obtain

$$\partial_y H(x,0) = \frac{1}{2}\left(\partial_y^2 D(x,0) - \partial_y^2 N(x,0)\right). \tag{247}$$

One sees that

$$\partial_y^2 D(x,0) = \partial_y^2(1 - ay^2 + A(x,y)y^2)|_{y=0} = -2a + O(x), \tag{248}$$

while

$$\partial_y^2 N(x,0) = \left( \frac{\partial_y^2(\Delta \circ T)}{\Delta} - \frac{(\Delta \circ T)\partial_y^2\Delta}{\Delta^2} - 2\frac{\partial_y(\Delta \circ T)\,\partial_y\Delta}{\Delta^2} + 2\frac{(\Delta \circ T)\,(\partial_y\Delta)^2}{\Delta^3} \right)(x,0).$$
(249)

Since

$$\partial_y^2(\Delta \circ T)(x,0) = \partial_y^2 T_y(x,0) - u''(x)(\partial_y T_x(x,0))^2 - u'(x)\partial_y^2 T_x(x,0) = 2a\kappa x + O(x^2)$$
(250)

one obtains

$$\partial_y^2 N(x,0) = -\frac{2a\kappa x + O(x^2)}{\kappa x + O(x^2)} - 2\frac{1 + bx^2 + O(x^3)}{\kappa^2 x^2 + O(x^3)} + 2\frac{1}{\kappa^2 x^2 + O(x^3)}$$
(251)

$$= -2a - 2b/\kappa^2 + O(x) = -2c + O(x).$$
(252)

Thus

$$\partial_y H(x,0) = \frac{1}{2}(-2a + 2c + O(x)) = (c - a) + O(x).$$
(253)

We deduce from (246), (253) and (239) that

$$D(x,y) - N(x,y) = (c - a)(y^2 + \kappa yx) + O(yx^2, y^2 x, y^3).$$
(254)

It follows that for all sufficiently small $x > 0, y \geq 0$, one has $D(x,y) - N(x,y) \geq \frac{(c-a)}{2}y^2$ and $1 \geq D(x,y) > 0$ so that

$$\frac{\widetilde{\Delta} \circ T(x,y)}{\widetilde{\Delta}(x,y)} = 1 - \frac{D(x,y) - N(x,y)}{D(x,y)} \leq 1 - \frac{c - a}{2}y^2$$
(255)

as claimed. $\qquad\square$

Lemma D.7 below gives uniform upper-bounds on the magnitudes of iterates, which are used in enabling certain estimates in the convergence theorem.

**Lemma D.7.** *Define $W(x,y) := bx^2 + \frac{a}{2}y^2$. Then for all $(x_0, y_0)$ in a sufficiently small neighbourhood of $0$, with $y_0 > 0$, the sequence $x_t$ is monotonically non-increasing and the sequence $y_t$ satisfies the uniform bound $y_t^2 \leq 2W(x_0, y_0)/a$ for all $t \in \mathbb{N}$.*

*Proof.* In any sufficiently small neighbourhood $U$ of $0 \in \mathbb{R}^2_{\geq 0}$, one has

$$W(T(x,y)) = bx^2\big(1 - ay^2 + O(xy^2, y^3)\big)^2 + \frac{1}{2}ay^2\big(1 + bx^2 - cy^2 + O(x^3, y^3)\big)^2$$
(256)

$$= bx^2(1 - 2ay^2) + O(x^3y^2, x^2y^3) + \frac{1}{2}ay^2(1 + 2bx^2 - 2cy^2) + O(x^3y^2, y^5)$$
(257)

$$= W(x,y) - abx^2y^2 - acy^4 + O(x^3y^2, x^2y^3, y^5)$$
(258)

$$\leq W(x,y) - \frac{1}{2}y^2(abx^2 + acy^2)$$
(259)

for all $(x,y) \in U$, so that $W$ is non-increasing as a function of the iterates of the map close to zero.

Now fix $(x_0, y_0)$ sufficiently small that, with $W_0 := W(x_0, y_0)$ one has $(x_0, \sqrt{2W_0/a})$ contained in $U$. Then in particular $(x_0, y_0)$ is contained in $U$ since $y_0 \leq \sqrt{2W_0/a}$. Suppose as an inductive hypothesis that $y_t \leq \sqrt{2W_0/a}$, $x_t \leq x_0$ and $W_t \leq W_0$. Shrinking $U$ yet further if necessary, one sees that

$$1 > 1 - y_t^2 + O(x_t y_t^2, y_t^3) > 0$$
(260)

thus yielding $x_{t+1} < x_t \leq x_0$. Moreover, the inductive hypothesis implies that $(x_t, y_t) \in U$; thus, denoting $W_t := W(x_t, y_t)$ and $W_{t+1} := W(x_{t+1}, y_{t+1})$, one sees that

$$W_{t+1} \leq W_t - \frac{1}{2}ay_t^2(bx_t^2 + cy_t^2) \leq W_0,$$
(261)

so that $y_{t+1} \leq \sqrt{2W_{t+1}/a} \leq \sqrt{2W_0/a}$. This completes the proof. $\qquad\square$

Finally we can prove convergence of the iterates of $T$ toward zero.

**Theorem D.8.** *Set $\kappa := \sqrt{b/(c-a)}$. For all sufficiently small $x_0 \geq 0$ and $y_0 > 0$ and all $0 < \epsilon < \kappa$, there is*

$$\tau \equiv \tau(y_0, \epsilon) = O(y_0^{-2} \epsilon^{-3c/(c-a)}) \tag{262}$$

*so that for all $t \geq \tau$, one has*

$$\frac{x_\tau}{\sqrt{1 + 3a(\kappa + \epsilon)^2 x_\tau^2 (t - \tau)}} \leq x_t \leq \frac{x_\tau}{\sqrt{1 + 2a(\kappa - \epsilon)^2 x_\tau^2 (t - \tau)}} \tag{263}$$

*and*

$$(\kappa - \epsilon/2)x_t \leq y_t \leq (\kappa + \epsilon/2)x_t. \tag{264}$$

*In particular, $x_t, y_t = \Theta(t^{-1/2})$.*

*Proof.* Using Lemma D.7, we may take $(x_0, y_0)$ sufficiently small that the quantity $\widetilde{\Delta}_t := \widetilde{\Delta}(x_t, y_t)$ of Lemma D.6 satisfies the recursion

$$|\widetilde{\Delta}_{t+1}| \leq |\widetilde{\Delta}_t| \left(1 - \frac{c - a}{2} y_t^2\right) \tag{265}$$

for all $t \in \mathbb{N}$. By taking $(x_0, y_0)$ yet smaller if necessary we may also assume that $y_t$ satisfies the lower-bound

$$y_{t+1} \geq y_t(1 - 2cy_t^2). \tag{266}$$

Squaring and taking the reciprocal yields

$$\frac{1}{y_{t+1}^2} \leq \frac{1}{y_t^2} \frac{1}{1 - 2cy_t^2}, \tag{267}$$

and, taking $(x_0, y_0)$ yet smaller if necessary so that $1/(1 - 2cy_t^2) \leq 1 + 6y_t^2$ for all $t$, one obtains the recursive estimate

$$\frac{1}{y_{t+1}^2} - \frac{1}{y_t^2} \leq 6c, \tag{268}$$

from which one obtains the bound

$$y_t^2 \geq \frac{y_0^2}{1 + 6cy_0^2 t} \tag{269}$$

for all $t \in \mathbb{N}$ by telescoping. Plugging this into the recursion for $\widetilde{\Delta}$ and taking logarithms, one has

$$\ln|\widetilde{\Delta}_{t+1}| - \ln|\widetilde{\Delta}_t| \leq -\frac{c - a}{2} \frac{y_0^2}{1 + 6cy_0^2 t}. \tag{270}$$

Taking a telescoping sum yields

$$\ln|\widetilde{\Delta}_t| - \ln|\widetilde{\Delta}_0| \leq -\frac{c - a}{12c} \sum_{s=0}^{t-1} \frac{1}{\frac{1}{6cy_0^2} + s} \leq -\frac{c - a}{12c} \ln(1 + 6cy_0^2 t) \tag{271}$$

by Lemma E.4. Thus

$$|\widetilde{\Delta}_t| \leq |\widetilde{\Delta}_0|(1 + 6cy_0^2 t)^{-(c-a)/(12c)}, \tag{272}$$

and for any $0 < \epsilon < \sqrt{b/(c-a)}$ one sees that for any $t$ greater than or equal to

$$\tau(y_0, \epsilon) := \left\lceil \frac{1}{6cy_0^2} \left(\left(\frac{\epsilon}{4|\Delta_0|}\right)^{-12c/(c-a)} - 1\right) \right\rceil \tag{273}$$

one has $|\widetilde{\Delta}_t| < \epsilon/4$. It follows from the definition of $\widetilde{\Delta}(x, y) := (y - \kappa x + O(x^2))/x$ that $t \geq \tau(y_0, \epsilon)$ implies

$$y_t \in [\kappa x - (\epsilon/4)x + O(x^2), \kappa x + (\epsilon/4)x + O(x^2)] \subset [(\kappa - \epsilon/2)x, (\kappa + \epsilon/2)x], \tag{274}$$

where, shrinking $x_0, y_0$ if necessary, the final containment is obtained using Lemma D.7. With $t \geq \tau(y_0, \epsilon)$ and the bounds $(\kappa + \epsilon/2)x_t \geq y_t \geq (\kappa - \epsilon/2)x_t$ in hand, one obtains

$$(1 - a(\kappa + \epsilon)^2 x_t^2)x_t \leq x_{t+1} \leq (1 - a(\kappa - \epsilon)^2 x_t^2)x_t, \tag{275}$$

again shrinking $x_0, y_0$ and invoking Lemma D.7 if necessary to deal with the higher-order terms. Inverting and squaring yield the bounds

$$\frac{1}{x_{t+1}^2} \leq \frac{1}{x_t^2}\frac{1}{(1 - a(\kappa + \epsilon)^2 x_t^2)^2} \leq \frac{1}{x_t^2}(1 + 3a(\kappa + \epsilon)^2 x_t^2) \tag{276}$$

and

$$\frac{1}{x_{t+1}^2} \geq \frac{1}{x_t^2}\frac{1}{(1 - a(\kappa - \epsilon)^2 x_t^2)^2} \geq \frac{1}{x_t^2}(1 + 2a(\kappa - \epsilon)^2 x_t^2), \tag{277}$$

again shrinking $x_0, y_0$ and invoking Lemma D.7 if necessary to obtain the final upper bound in (276), so that

$$3a(\kappa + \epsilon)^2 \geq \frac{1}{x_{t+1}^2} - \frac{1}{x_t^2} \geq 2a(\kappa - \epsilon)^2, \tag{278}$$

from which telescoping summation yields the bounds (263). The corresponding bounds for $y_t$ now follow from the limits of (274). This completes the proof. $\qquad\square$

Finally, we may prove our convergence theorem in the critical regime using Theorem D.8.

**Theorem D.9.** *Assume that $\eta = 2/\lambda(\theta_*^{\|})$. Assume in addition that $\nu/(c(2/\lambda(\theta_*^{\|}), \theta_*^{\|})\lambda(\theta_*^{\|})) < 1$, where $\nu$ is defined in Assumption 3.4 and $c$ is defined in Assumption 3.3. Then for all $\theta_0^{\|}$ sufficiently close to $\theta_*^{\|}$ and all $\theta^{\perp} \neq 0$ sufficiently small, one has $d_M(\theta_t^{\|}, \theta_*^{\|}) = \Theta(t^{-1/2})$ and $|\theta_t^{\perp}| = \Theta(t^{-1/2})$.*

*Proof.* Denote $\lambda_* := \lambda(\theta_*^{\|})$ and $c_* := c(2/\lambda_*, \theta_*^{\|})$ for notational convenience. By Theorem 4.1, for all $(\theta^{\|}, \theta^{\perp})$ sufficiently close to $(\theta_*^{\|}, 0)$, GD has the coordinate expression

$$\begin{pmatrix} \mathrm{GD}^{\|}(\eta, \theta^{\|}, \theta^{\perp}) \\ \mathrm{GD}^{\perp}(\eta, \theta^{\|}, \theta^{\perp}) \end{pmatrix} = \begin{pmatrix} \exp_{\theta^{\|}}\left(-\frac{(\theta^{\perp})^2}{c(2/\lambda(\theta^{\|}))\lambda_*}\nabla_M\lambda(\theta^{\|})\right) + O((\theta^{\perp})^3 d_M(\theta^{\|}, \theta_*^{\|})) \\ \left(1 - \frac{2\lambda(\theta^{\|})}{\lambda_*}\right)\theta^{\perp} + (\theta^{\perp})^3 + O((\theta^{\perp})^4) \end{pmatrix}. \tag{279}$$

We first derive a formula for $d_M\big(\mathrm{GD}^{\|}(\eta, \theta^{\|}, \theta^{\perp}), \theta_*^{\|}\big) = \|\log_{\theta_*^{\|}}(\mathrm{GD}^{\|}(\eta, \theta^{\|}, \theta^{\perp}))\|$. Using a covariant Taylor expansion (29), observe that:

$$\nabla_M\lambda(\theta^{\|}) = \Pi_{\theta_*^{\|} \to \theta^{\|}}\big(\nabla_M\lambda(\theta_*^{\|}) + \nabla_M^2\lambda(\theta_*^{\|})[\log_{\theta_*^{\|}}(\theta^{\|})] + O\big(d_M(\theta^{\|}, \theta_*^{\|})^2\big)\big) \tag{280}$$

$$= \Pi_{\theta_*^{\|} \to \theta^{\|}}\big(\nu \log_{\theta_*^{\|}}(\theta^{\|}) + O(d_M(\theta^{\|}, \theta_*^{\|})^2)\big). \tag{281}$$

by Assumption 3.4, where $\Pi_{\theta_*^{\|} \to \theta^{\|}}$ is parallel transport from $\theta_*^{\|}$ to $\theta^{\|}$. Similarly, observe that

$$\frac{(\theta^{\perp})^2}{c(2/\lambda(\theta^{\|}))\lambda_*} = \frac{(\theta^{\perp})^2}{c_*\lambda_*} + O\big((\theta^{\perp})^2 d_M(\theta^{\|}, \theta_*^{\|})\big). \tag{282}$$

We may thus write

$$\log_{\theta_*^{\|}}\big(\mathrm{GD}^{\|}(\eta, \theta^{\|}, \theta^{\perp})\big) = \log_{\theta_*^{\|}}\left(\exp_{\theta^{\|}}\left(-\frac{\nu(\theta^{\perp})^2}{c_*\lambda_*}\Pi_{\theta_*^{\|} \to \theta^{\|}}\big(\log_{\theta_*^{\|}}(\theta^{\|})\big)\right)\right) \tag{283}$$

$$+ O\big((\theta^{\perp})^3 d_M(\theta^{\|}, \theta_*^{\|}), (\theta^{\perp})^2 d_M(\theta^{\|}, \theta_*^{\|})^2\big) \tag{284}$$

$$= \left(1 - \frac{\nu}{c_*\lambda_*}(\theta^{\perp})^2\right)\log_{\theta_*^{\|}}(\theta^{\|}) \tag{285}$$

$$+ O\big((\theta^{\perp})^3 d_M(\theta^{\|}, \theta_*^{\|}), (\theta^{\perp})^2 d_M(\theta^{\|}, \theta_*^{\|})^2\big). \tag{286}$$

Hence:

$$d_M\big(\mathrm{GD}^{\parallel}(\eta,\theta^{\parallel},\theta^{\perp}),\theta_*^{\parallel}\big) = \big\|\log_{\theta_*^{\parallel}}(\mathrm{GD}^{\parallel}(\eta,\theta^{\parallel},\theta^{\perp}))\big\| \tag{287}$$

$$= \Big(1 - \frac{\nu}{c_*\lambda_*}(\theta^{\perp})^2 + O\big((\theta^{\perp})^3 + (\theta^{\perp})^2 d_M(\theta^{\parallel},\theta_*^{\parallel})\big)\Big) d_M(\theta^{\parallel},\theta_*^{\parallel}). \tag{288}$$

We now derive a formula for $|\mathrm{GD}^{\perp}(\eta,\theta^{\parallel},\theta^{\perp})|$. Perform a covariant Taylor expansion (29) on $\lambda(\theta^{\parallel})$ to obtain

$$\lambda(\theta^{\parallel}) = \lambda_* + \langle\nabla_M\lambda(\theta_*^{\parallel}),\log_{\theta_*^{\parallel}}(\theta^{\parallel})\rangle + \frac{1}{2}\nabla_M^2\lambda(\theta_*^{\parallel})[\log_{\theta_*^{\parallel}}(\theta^{\parallel})^{\otimes 2}] + O\big(d_M(\theta^{\parallel},\theta_*^{\parallel})^3\big) \tag{289}$$

$$= \lambda_* + \frac{1}{2}\nu\, d_M(\theta^{\parallel},\theta_*^{\parallel})^2 + O\big(d_M(\theta^{\parallel},\theta_*^{\parallel})^3\big) \tag{290}$$

by Assumption 3.4. Then one sees that

$$\mathrm{GD}^{\perp}(\eta,\theta^{\parallel},\theta^{\perp}) = -\Big(1 + \frac{\nu}{\lambda_*}d_M(\theta^{\parallel},\theta_*^{\parallel})^2 + O\big(d_M(\theta^{\parallel},\theta_*^{\parallel})^3\big)\Big)\theta^{\perp} + (\theta^{\perp})^3 + O\big((\theta^{\perp})^4\big) \tag{291}$$

$$= -\Big(1 + \frac{\nu}{\lambda_*}d_M(\theta^{\parallel},\theta_*^{\parallel})^2 - (\theta^{\perp})^2 + O\big(d_M(\theta^{\parallel},\theta_*^{\parallel})^3,(\theta^{\perp})^3\big)\Big)\theta^{\perp}. \tag{292}$$

so that

$$|\mathrm{GD}^{\perp}(\eta,\theta^{\parallel},\theta^{\perp})| = \Big(1 + \frac{\nu}{\lambda_*}d_M(\theta^{\parallel},\theta_*^{\parallel})^2 - (\theta^{\perp})^2 + O\big(d_M(\theta^{\parallel},\theta_*^{\parallel})^3,(\theta^{\perp})^3\big)\Big)|\theta^{\perp}| \tag{293}$$

provided $d_M(\theta^{\parallel},\theta_*^{\parallel})$ and $|\theta^{\perp}|$ are sufficiently small. The result now follows by invoking Theorem D.8. $\qquad\square$

## D.3 Supercritical regime

Our convergence theorem in the supercritical regime is the easiest of the three. The orbit to which the iterates are attracted is hyperbolic unlike in the critical case, and the transient phenomena are relatively simple compared with those of the subcritical case.

**Theorem D.10.** *Assume that $\eta > 2/\lambda(\theta_*^{\parallel})$ is sufficiently small. Then there are positive constants $C_1, C_2, C_3$ and a stable orbit of period two of the form $\big(\theta_*^{\parallel}, \pm(\eta\lambda(\theta_*^{\parallel}) - 2)^{1/2} + O(\eta\lambda(\theta_*^{\parallel}) - 2)\big)$ to which the iterates $(\theta_t^{\parallel},\theta_t^{\perp})$ starting from any $(\theta_0^{\parallel},\theta_0^{\perp})$ satisfying $0 < |\theta_0^{\perp}| \le C_1(\eta\lambda(\theta_*^{\parallel}) - 2)^{1/2}$ and $d_M(\theta_0^{\parallel},\theta_*^{\parallel}) \le C_2(\eta\lambda(\theta_*^{\parallel}) - 2)^{1/2}$ converge with rate $O\big((1 - C_3(\eta\lambda(\theta_*^{\parallel}) - 2))^t\big)$.*

*Proof.* Arguing using covariant Taylor expansions as in the proof of Theorem D.9, one sees that

$$d_M\big(\mathrm{GD}^{\parallel}(\eta,\theta^{\parallel},\theta^{\perp}),\theta_*^{\parallel}\big) = \Big(1 - \frac{\eta\nu}{2c(\eta,\theta_*^{\parallel})}(\theta^{\perp})^2 + O\big((\theta^{\perp})^3,(\theta^{\perp})^2 d_M(\theta^{\parallel},\theta_*^{\parallel})\big)\Big) d_M(\theta^{\parallel},\theta_*^{\parallel}) \tag{294}$$

and

$$\mathrm{GD}^{\perp}(\eta,\theta^{\parallel},\theta^{\perp}) = -\Big(1 + (\eta\lambda(\theta_*^{\parallel}) - 2) + \frac{\eta\nu}{2}d_M(\theta^{\parallel},\theta_*^{\parallel})^2 - (\theta^{\perp})^2 + O\big(d_M(\theta^{\parallel},\theta_*^{\parallel})^3,(\theta^{\perp})^3\big)\Big)\theta^{\perp}. \tag{295}$$

Consequently, the result follows from Theorem D.11 below. $\qquad\square$

**Theorem D.11.** *Given $\alpha > 0$ and constants $a > 0$ and $b > 0$, consider the functions*

$$F_x(x,y) := (1 - ay^2 + O(y^3, xy^2))x, \qquad F_y(x,y) = -(1 + \alpha + bx^2 - y^2 + O(x^3,y^3))y. \tag{296}$$

*Then, there are constants $C_1, C_2, C_3 > 0$ such that for all $\alpha$ sufficiently small, the map $F = (F_x, F_y)$ admits a stable, period-2 orbit $\{(0,\xi_+),(0,\xi_-)\}$ with $\xi_{\pm} = \pm\sqrt{\alpha} + O(\alpha)$, and the iterates $(x_{t+1},y_{t+1}) := F(x_t,y_t)$ starting from any $(x_0,y_0)$ with $0 < |y_0| \le C_1\sqrt{\alpha}$ and $|x_0| \le C_2\sqrt{\alpha}$ converge to this orbit at a rate of $O\big((1 - C_3\alpha)^t\big)$.*

*Proof.* The line $x = 0$ is preserved by $F$, and thereon one sees that $F_y$ takes the form

$$F_y(0, \cdot) : y \mapsto -(1 + \alpha)y + y^3 + O(y^4) \tag{297}$$

which, by Lemma D.2, admits a period-two orbit $\{\xi_+, \xi_-\}$ of the form $\xi_\pm = \pm\sqrt{\alpha} + O(\alpha)$. We will show that the iterates of the square $F^2$ of $F$ converge to $(0, \xi_\pm)$ for all initial points $(x_0, y_0)$ sufficiently small, $y_0 \neq 0$. This will be achieved by showing that $\|DF^2(x, y)\| < 1$ uniformly over a neighbourhood of $(0, \xi_\pm)$, followed by proving a guarantee of convergence to this neighbourhood in finite time.

Via a routine calculation one sees that

$$F^2(x, y) = \begin{pmatrix} x\big(1 - a(1 + (1 + \alpha)^2)y^2 + O(xy^2, y^3)\big) \\ y\big((1 + \alpha)^2 + 2(1 + \alpha)bx^2 - (1 + \alpha)(1 + (1 + \alpha)^2)y^2 + O(x^3, x^2y, xy^2, y^3)\big) \end{pmatrix}. \tag{298}$$

The derivative of $F^2$ is then given by

$$DF^2(x, y) = \begin{pmatrix} 1 - a(1 + (1 + \alpha)^2)y^2 & -2a(1 + (1 + \alpha)^2)xy \\ 4(1 + \alpha)bxy & (1 + \alpha)^2 + 2(1 + \alpha)bx^2 - 3(1 + \alpha)(1 + (1 + \alpha)^2)y^2 \end{pmatrix} \tag{299}$$

$$+ O(x^3, x^2y, xy^2, y^3). \tag{300}$$

In particular, for all $y$ satisfying

$$\frac{\alpha}{(1 + \alpha)(1 + (1 + \alpha)^2)} \leq y^2 \leq 2\alpha,, \tag{301}$$

the matrix $DF^2(x, y)$ has entries with magnitudes upper-bounded by

$$\begin{pmatrix} 1 - \frac{a\alpha}{1+\alpha} & O(x\sqrt{\alpha}) \\ O(x\sqrt{\alpha}) & 1 - (1/2)\alpha + O(x^2) \end{pmatrix} + O(x^3, \sqrt{\alpha}x^2, \alpha x, \alpha^{3/2}) \tag{302}$$

for all $\alpha$ sufficiently small. Using the bound $\|A\|_2 \leq \sqrt{\|A\|_1 \|A\|_\infty}$ for a matrix $A$, where $\|\cdot\|_1$ and $\|\cdot\|_\infty$ are the maximum column 1-norm and maximum row 1-norm respectively, one sees then that there are $C_2, C_3 > 0$ such that for all $\alpha$ sufficiently small, all $x$ satisfying

$$|x| \leq C_2\sqrt{\alpha}. \tag{303}$$

and all $y$ satisfying (301), one has $\|DF^2(x, y)\|_2 \leq 1 - C_3\alpha$.

Letting $V$ be the neighbourhood of the two-point set $\{(0, \xi_\pm)\}$ enclosed by the bounds (301) and (303), one sees that the iterates $(x_{2t}, y_{2t})$ of $F^2$ starting at any point $(x_0, y_0) \in V$ converge toward $(0, \xi_{\text{sign}(y_0)})$ with rate $\|(x_{2t}, y_{2t} - \xi_\pm)\| \leq (1 - C_3\alpha)^t \|(x_0, y_0 - \xi_{\text{sign}(y_0)})\|$.

We complete the proof by showing that the iterates starting from any $x_0$ satisfying (303) and any $y_0$ satisfying

$$0 < y_0^2 \leq \frac{\alpha}{(1 + \alpha)(1 + (1 + \alpha)^2)} \tag{304}$$

are eventually drawn into $V$. This, however, is easy: for any $x$ satisfying (303) and any $y$ satisfying $y^2 \leq \alpha/\big((1 + \alpha)(1 + (1 + \alpha)^2)\big)$, one has

$$|(F^2)_y(x, y)| \geq |y|\big((1 + \alpha)^2 - (1 + \alpha)(1 + (1 + \alpha)^2)y^2 + O(\alpha^{3/2})\big) \tag{305}$$

$$\geq |y|\big(1 + (1/2)\alpha\big) \tag{306}$$

for all $\alpha$ sufficiently small. It follows that the iterates $y_{2t} = (F^{2t})_y(x_0, y_0)$ satisfy the recursion $|y_{2(t+1)}| \geq \big(1 + (1/2)\alpha\big)|y_{2t}|$ as long as $y_{2t}^2 \leq \alpha/\big((1 + \alpha)(1 + (1 + \alpha)^2)\big)$. We claim now that there exists $\tau \in \mathbb{N}$ satisfying

$$\tau \leq \frac{1}{2} \left\lceil \ln\left(\frac{\alpha}{y_0^2(1 + \alpha)(1 + (1 + \alpha)^2)}\right) \ln\left(1 + \frac{1}{2}\alpha\right)^{-1} \right\rceil \tag{307}$$

such that $y_{2\tau} \geq \alpha/\big((1 + \alpha)(1 + (1 + \alpha)^2)\big)$. Suppose for a contradiction that this were not the case; then

$$\frac{\alpha}{(1 + \alpha)(1 + (1 + \alpha)^2)} > y_0^2(1 + (1/2)\alpha)^{2\tau} \geq \frac{\alpha}{(1 + \alpha)(1 + (1 + \alpha)^2)}, \tag{308}$$

which is a contradiction. This proves the result. $\qquad \square$

# E  Elementary lemmas

In this section, for ease of reference, we collect several well-known facts and their elementary proofs. Recall that a subset $M'$ of a Riemannian manifold $M$ is *geodesically convex* if any two points of $M'$ can be connected by a unique geodesic contained entirely in $M'$.

**Lemma E.1.** *Let $\lambda : M' \to \mathbb{R}$ be a $C^2$ function on a geodesically convex subset $M'$ of a Riemannian manifold $M$. If $\lambda$ is $\mu$-geodesically strongly convex in the sense that $\nabla_M^2 \lambda \succeq \mu I_{TM}$ at all points on $M'$, then*

$$\lambda(y) \geq \lambda(x) + \langle \nabla_M \lambda(x), \log_x(y) \rangle + \frac{\mu}{2} d_M(x, y)^2 \tag{309}$$

*for all $x, y \in M'$.*

*Proof.* Fix $x, y \in M'$. Since $M'$ is geodesically convex, there exists a unique geodesic $\gamma : [0, 1] \to M'$ such that $\gamma(0) = x$ and $\gamma(1) = y$. This $\gamma$ satisfies

$$\dot{\gamma}(t) = \Pi(\gamma)_t \dot{\gamma}(0) = \Pi(\gamma)_t \log_x(y), \qquad \|\dot{\gamma}(t)\| = d_M(x, y) \tag{310}$$

for all $t \in [0, 1]$, where $\Pi(\gamma)_t : T_{\gamma(0)} M \to T_{\gamma(t)} M$ is parallel transport. Define $\varphi(t) := \lambda(\gamma(t))$. Then using (23) and the chain rule,

$$\dot{\varphi}(t) = \langle \nabla_M \lambda(\gamma(t)), \dot{\gamma}(t) \rangle, \qquad \ddot{\varphi}(t) = \langle \dot{\gamma}(t), \nabla_M^2 \lambda(\gamma(t)) \dot{\gamma}(t) \rangle \tag{311}$$

for all $t \in [0, 1]$. Thus, by Taylor's theorem, there is $s \in [0, 1]$ such that

$$\lambda(y) = \varphi(1) = \varphi(0) + \dot{\varphi}(0) + \frac{1}{2} \ddot{\varphi}(s) \tag{312}$$

$$= \lambda(x) + \langle \nabla_M \lambda(x), \log_x(y) \rangle + \frac{1}{2} \langle \dot{\gamma}(s), \nabla_M^2 \lambda(\gamma(s)) \dot{\gamma}(s) \rangle \tag{313}$$

$$\geq \lambda(x) + \langle \nabla_M \lambda(x), \log_x(y) \rangle + \frac{\mu}{2} \|\dot{\gamma}(s)\|^2 \tag{314}$$

$$= \lambda(x) + \langle \nabla_M \lambda(x), \log_x(y) \rangle + \frac{\mu}{2} d_M(x, y)^2 \tag{315}$$

as claimed. $\square$

**Lemma E.2.** *Let $\lambda : M' \to \mathbb{R}$ be a $C^2$ function on a geodesically convex subset $M'$ of a Riemannian manifold $M$. If $\lambda$ is $\mu$-geodesically strongly convex in the sense that $\nabla_M^2 \lambda \succeq \mu I_{TM}$ at all points on $M'$, then*

$$\langle \Pi_{x \to y} \nabla_M \lambda(x) - \nabla_M \lambda(y), \log_y(x) \rangle \geq \mu \, d_M(x, y)^2 \tag{316}$$

*for all $x, y \in M'$, where $\Pi_{x \to y} : T_x M \to T_y M$ is parallel transport.*

*Proof.* Fix $x, y \in M'$. From Lemma E.1, one has the estimates

$$\lambda(y) \geq \lambda(x) + \langle \nabla_M \lambda(x), \log_x(y) \rangle + \frac{\mu}{2} d_M(x, y)^2, \tag{317}$$

$$\lambda(x) \geq \lambda(y) + \langle \nabla_M \lambda(y), \log_y(x) \rangle + \frac{\mu}{2} d_M(x, y)^2. \tag{318}$$

Since parallel transport is an orthogonal transformation, one has

$$\langle \nabla_M \lambda(x), \log_x(y) \rangle = -\langle \nabla_M \lambda(x), \Pi_{y \to x} \log_y(x) \rangle = -\langle \Pi_{x \to y} \nabla_M \lambda(x), \log_y(x) \rangle. \tag{319}$$

Substituting this into (317) and rearranging yields

$$\langle \Pi_{x \to y} \nabla_M \lambda(x), \log_y(x) \rangle \geq \frac{\mu}{2} d_M(x, y)^2 + \lambda(x) - \lambda(y) \tag{320}$$

while (318) may be rearranged to give

$$-\langle \nabla_M \lambda(y), \log_y(x) \rangle \geq \frac{\mu}{2} d_M(x, y)^2 + \lambda(y) - \lambda(x). \tag{321}$$

Adding these estimates then yields the result. $\square$

**Lemma E.3.** *Let* $\lambda : M' \to \mathbb{R}$ *be a* $C^2$ *function on a geodesically convex subset* $M'$ *of a Riemannian manifold* $M$. *If* $\lambda$ *is* $\mu$-*geodesically strongly convex in the sense that* $\nabla_M^2 \lambda \succeq \mu I_{TM}$ *at all points on* $M'$ *and admits a critical point* $x_*$ *in* $M'$, *then* $x_*$ *is the unique global minimum of* $\lambda$ *in* $M'$ *and*

$$\|\nabla_M \lambda(x)\|^2 \geq 2\mu\big(\lambda(x) - \lambda(x_*)\big) \tag{322}$$

*for all* $x \in M'$.

*Proof.* That the global minimum $x_*$ is unique follows from Lemma E.1: for any $x \neq x_*$ in $M$, one has

$$\lambda(x) \geq \lambda(x_*) + \frac{\mu}{2} d_M(x, x_*)^2 > \lambda(x_*), \tag{323}$$

thus proving the first claim. To prove the second claim, fix $x \in M'$. Lemma E.1 then applies to yield

$$\lambda(x) - \lambda(x_*) \leq -\langle \nabla_M \lambda(x), \log_x(x_*) \rangle - \frac{\mu}{2} \|\log_x(x_*)\|^2 \tag{324}$$

$$= -\frac{\mu}{2} \left\| \frac{1}{\mu} \nabla_M \lambda(x) + \log_x(x_*) \right\|^2 + \frac{1}{2\mu} \|\nabla_M \lambda(x)\|^2 \tag{325}$$

$$\leq \frac{1}{2\mu} \|\nabla_M \lambda(x)\|^2, \tag{326}$$

from which the result follows. $\qquad\square$

**Lemma E.4.** *For any* $c > 0$, *and any integers* $0 \leq s < t$, *one has*

$$\ln\left(\frac{c+t+1}{c+s}\right) \leq \sum_{k=s}^{t} \frac{1}{c+k} \leq \frac{1}{c+s} + \ln\left(\frac{c+t}{c+s}\right). \tag{327}$$

*Proof.* Since $x \mapsto \frac{1}{(c+x)}$ is monotonically decreasing along the positive reals, one has

$$\int_n^{n+1} f(x)dx \leq \int_n^{n+1} f(n)dx = f(n) = \int_{n-1}^n f(n)dx \leq \int_{n-1}^n f(x)dx \tag{328}$$

for any natural number $n$, from which it follows that

$$\int_s^{t+1} \frac{1}{(c+x)}dx \leq \sum_{k=s}^{t} \frac{1}{(c+k)} = \frac{1}{(c+s)} + \sum_{k=s+1}^{t} \frac{1}{(c+k)} \tag{329}$$

$$\leq \frac{1}{(c+s)} + \int_s^t \frac{1}{(c+x)}dx. \tag{330}$$

Integrating both sides now yields the result. $\qquad\square$

# F Experimental supplement

Our experiments were conducted for a depth 5, width 1 scalar factorisation problem, corresponding to the function $f : \mathbb{R}^p \to \mathbb{R}$ defined by

$$f(\theta_1, \ldots, \theta_p) := \theta_p \cdots \theta_1, \qquad (\theta_1, \ldots, \theta_p) \in \mathbb{R}^p, \tag{331}$$

with target value $y = 1$. Our code[3] runs gradient descent on this problem, and can be executed in seconds on a single cpu. At each iterate, the projection onto the solution manifold $M := f^{-1}\{y\}$ is computed as follows.

The KKT conditions for the constrained optimisation problem

$$\min_{\theta^\| \geq 0} \frac{1}{2} \|\theta^\| - \theta\|^2 \quad \text{such that} \quad \prod_{i=1}^p \theta_i^\| = y \tag{332}$$

---

[3]Available at `https://github.com/lemacdonald/eos-convergence-rates-codimension-1`

are given by

$$\theta_i^{\|} - \theta_i^{\|}\theta_i + \alpha\frac{y}{\theta_i^{\|}} = 0, \qquad i = 1, \ldots, p, \tag{333}$$

where $\alpha$ is the constraint parameter. These equations admit the quadratic solutions

$$\theta_i^{\|}(\alpha) = \frac{\theta_i \pm \sqrt{\theta_i^2 - 4\alpha y}}{2}, \qquad i = 1, \ldots, p. \tag{334}$$

It is easily verified $\alpha \mapsto \theta_i^{\|}(\alpha)$, is monotonically decreasing (when (334) is taken with the $+$ sign) or monotonically increasing (when (334) is taken with the $-$ sign); consequently, the map $\phi : \alpha \mapsto \prod_{i=1}^{p} \theta_i^{\|}(\alpha)$ is also monotonic provided the signs are consistent among the $\theta_i^{\|}(\alpha)$.

The sign conventions we use are as follows. If $\prod_{i=1}^{p} \theta_i < y$, then we must take the $+$ sign for all $\theta_i^{\|}(\alpha)$. If $\prod_{i=1}^{p} \theta_i \geq y$, then, noting that $2^{-p} \prod_{i=1}^{p} \theta_i$ is the minimum value possible for $\prod_{i=1}^{p} \theta_i^{\|}(\alpha)$ when all $\theta_i^{\|}(\alpha)$ are taken with the positive sign in (334), either:

1. $y \geq 2^{-p} \prod_{i=1}^{p} \theta_i$, in which case we take the positive sign in (334) for all $i$.
2. $y < 2^{-p} \prod_{i=1}^{p} \theta_i$, in which case we take the negative sign in (334) for all $i$.

Having determined the sign to use uniformly over all $i = 1, \ldots, p$ in (334), we are assured that $\alpha \mapsto \phi(\alpha) = \prod_{i=1}^{p} \theta_i^{\|}(\alpha)$ is monotonic, hence that $\phi(\alpha) = y$ admits a unique solution which we obtain using the bisection method.

