# OpenReview forum: "Convergence Rates for Gradient Descent on the Edge of Stability for Overparametrised Least Squares"
_NeurIPS.cc/2025/Conference — NeurIPS 2025 poster_

### Official Review · Reviewer_QFWU · 2025-06-27

**Clarity:** 3
**Significance:** 2
**Originality:** 2
**Rating:** 4
**Confidence:** 4

**Summary:**

This paper studies the effect of step sizes in gradient descent through the phenomenon named edge of stability. The authors analyze three regimes: subcritical, critical, and supercritical. The analysis is based on decomposing the training dynamics into the parallel and orthogonal directions with respect to the solution manifold. Under the assumption of certain symmetry, isometry, and convexity, the authors prove the convergence results in each regime.

**Questions:**

The questions follow from "Weakness":
1. Could you give examples where all your assumptions are satisfied?
2. Could you discuss why those assumptions are necessary (not just sufficient) and provide examples demonstrating such necessity?
3. Could you conduct experiments that are more closely related to machine learning in practice?

**Ethical Concerns:**

["NO or VERY MINOR ethics concerns only"]

**Final Justification:**

I appreciate the fact that authors are honest with the limitation of their work, and upon critique they improve their results by loosening the assumptions. I still believe that the simplification of the model and the strong assumptions are strong limitations, which are also pointed out by other reviewers, but at the same time I agree that theory on edge of stability is so sparse that every progress counts.

**Limitations:**

Yes.

**Paper Formatting Concerns:**

I do not notice any major formatting issue.

**Quality:**

2

**Strengths And Weaknesses:**

Strength:

1. The intuitions are well explained.
2. The authors state the assumptions that the theorems are based on.
3. The proof of theoretical results are given coherently in the appendix.

Weakness:

1. The assumptions are so strong (symmetry, isometry, strong convexity) that it is difficult to find a reasonable setting where all assumptions are satisfied.
2. The experiments are based on simple settings such as scalar network (depth-5, width-1) and shallow linear network (depth-2, width-5). It is not convincing that that this will be a general behaviors for machine learning models.

---

> ### Author Rebuttal · Authors · 2025-07-30
>
> Thank you for your time reviewing and critiquing our paper. In summary, it seems your main critiques concern the strong assumptions and whether they are necessary. While our assumptions are strong and our examples are toy models only, they are no stronger than those of comparable works in the literature (which also deal only with toy models), while our results are much stronger. We are able to both prove and empirically demonstrate that one of our assumptions (suitably weakened) is *necessary* for our supercritical convergence theorem. See below for details.
>
> **Weaknesses**
>
> > The assumptions are so strong (symmetry, isometry, strong convexity) that it is difficult to find a reasonable setting where all assumptions are satisfied.
>
> Please note that we only insist on convexity of *the sharpness along the solution manifold*; we do not insist on convexity of the loss itself. We give two classes of neural network examples, with *non-convex loss landscapes*, where all our assumptions are satisfied (namely, factorization of a scalar either by other scalars or by two vectors, Lines 175-191). While we agree these are toy models, the present state of understanding of edge of stability dynamics is so limited that theoretical analysis is always restricted to simple models such as scalar linear networks [1] and two layer linear networks [2]. Unlike these other works, however, our assumptions are formulated in general mathematical terms which do not rule out possibly more complex models in the future and are suggestive of the mathematical features that should be considered when studying more complex models. Thus, even though our assumptions are restrictive, they are still an advance over prior works.
>
> > The experiments are based on simple settings such as scalar network (depth-5, width-1) and shallow linear network (depth-2, width-5). It is not convincing that that this will be a general behaviors for machine learning models.
>
> We make no claim that the precise assumptions we make, or the empirical observations we make, would extend to practical models like ResNets or Transformers. However, the *theory* of edge of stability is still primarily restricted to toy models [1,2], which must be understood before more complex models can be understood. By formulating our theory in general mathematical terms rather than in terms of specific toy architectures like in prior works, our theory is still an advance over prior works and a step towards understanding more complex models.
>
> **Questions**
>
> > Could you give examples where all your assumptions are satisfied?
>
> Please see Lines 175-191 for examples satisfying the assumptions we make (namely, factorization of a scalar either by other scalars or by two vectors). While we agree these are toy examples, we emphasise again that the theory of edge of stability is still so undeveloped that it has not progressed past such toy examples [1,2]. We cannot expect to have a theory of practical models without first having a theory of toy models, but by formulating our theory in general mathematical terms our paper is a step beyond prior works and a step toward bridging this gap.
>
> > Could you discuss why those assumptions are necessary (not just sufficient) and provide examples demonstrating such necessity?
>
> Great question.
>
> (a) We have discovered since submission that the symmetry assumption (Assumption 3.1, Lines 151-152) can be weakened as follows. Recall that we use $\theta_*^{\parallel}$ to denote the minimiser of the sharpness along the solution manifold $M$, and $n$ to denote the unit normal vector field along $M$. Then Assumption 3.1 can be weakened to the assumption that at each point $\theta:=\theta_*^{\parallel}+\alpha n(\theta_*^{\parallel})$ orthogonal to $M$ at $\theta_*^{\parallel}$, the loss gradient $\nabla\ell(\theta)$ is also orthogonal to $M$ at $\theta_*^{\parallel}$. The examples we cite satisfy this weakened assumption, and this weakened assumption *is* necessary for convergence to a stable period-2 orbit orthogonal to $M$ at the point $\theta_*^{\parallel}$. The proof is immediate: if gradient descent oscillates between two points on a line orthogonal to $M$ at $\theta_*^{\parallel}$, then necessarily the loss gradient must be parallel to this line, hence orthogonal to $M$. We are able to produce an empirical demonstration of this necessity, comparing the iterates of gradient descent (after approximate convergence to a stable orbit) for the 2 layer scalar linear network $f(\theta_1,\theta_2) = \theta_2\theta_1$ with a 2 layer scalar ReLU network $f(\theta_1,\theta_2) = \theta_2 ReLU(\theta_1)$ and a 2 layer scalar $\tanh$ network $f(\theta_1,\theta_2) = \theta_2\tanh(\theta_1)$ to fit the target value $4$. In the linear and ReLU cases, the assumption is satisfied, leading to oscillations orthogonal to $M$ at exactly the point of minimum sharpness $\theta_*^{\parallel}$; in the $\tanh$ case, the assumption is not satisfied, and it is clear from the plot that the oscillations do *not* occur about the point of minimum sharpness $\theta_*^{\parallel}$. This is a novel finding, suggesting that the properties of the nonlinearity play a key role in determining the implicit bias of edge of stability dynamics.
>
> (b) We can demonstrate empirically that Theorems 4.3 and 4.4 do not hold if $d_0:=d(\theta_0^{\parallel},\theta_*^{\parallel})$ is too large. Specifically, for a two layer scalar linear network, if $\eta\geq 2/\lambda(\theta_*^{\parallel})$ and $d_0$ is too large, gradient descent starting from $\|\theta^{\perp}_0\|\neq0$ diverges, violating Theorem 4.3 and 4.4. While it may be possible to quantitatively weaken the assumptions on $d_0$ in these theorems, this experiment demonstrates that theorems are *necessarily* local. We are happy to provide code for these simple experiments upon request.
>
> > Could you conduct experiments that are more closely related to machine learning in practice?
>
> Edge of stability experiments for more practical models such as ResNets, VGGs and Transformers on more practical datasets such as CIFAR10 and WikiText have been conducted in prior work [3]. However, these experiments only plot the sharpness at each iterate $\theta_t$, rather than at the *projection* $\theta^{\parallel}_t$ of each iterate to the solution manifold as we require. To conduct experiments reflective of our theory on practical models, we would need a scalable way of projecting iterates to the solution manifold as well as a method for finding the minimum value of the sharpness along the solution manifold. To our knowledge neither of these calculations have been attempted in the literature before. While we agree that these experiments would be desirable, we do not know how to perform them and believe they would be sufficiently difficult to warrant their own paper.
>
> Please let us know if you would like any further clarifications.
>
> 1. Gradient Descent Monotonically Decreases the Sharpness of Gradient Flow Solutions in Scalar Networks and Beyond, Kreisler et al, ICML2023.
> 2. Universal Sharpness Dynamics in Neural Network Training: Fixed Point Analysis, Edge of Stability, and Route to Chaos, Kalra et al, ICLR2025.
> 3. Gradient Descent on Neural Networks Typically Occurs at the Edge of Stability, Cohen et al, ICLR2021.

---

> > ### Comment · Reviewer_QFWU · 2025-08-05
> >
> > Thank you for the rebuttal and the references. I appreciate the honesty with the limitation of the work, and upon critique the improvement in results by loosening the assumptions. Although the simplification of the model and the strong assumptions are indeed limitations, which are also pointed out by other reviewers, at the same time I agree that theory on edge of stability is so little that every progress counts.

---

> > > ### Author Response · Authors · 2025-08-05
> > >
> > > Thank you for your continued engagement and appreciation of our contribution.

---

### Official Review · Reviewer_CvCh · 2025-06-27

**Clarity:** 3
**Significance:** 3
**Originality:** 3
**Rating:** 4
**Confidence:** 4

**Summary:**

This paper proves a generalized convergence result for gradient descent on the edge of stability in some “overparametrized least-squares models” with some generic assumptions, which generalizes several previous papers on certain deep scalar factorization. Based on the assumptions, the paper can rigorously prove local convergence in sub-critical, critical, and supercritical regimes (characterized by the sharpness of the flattest minimizer) where the initialization and flattest convergence point $\theta_*^{||}$ is sufficiently close.

**Questions:**

Questions:
1. Is there any more general case satisfying the assumptions?
2. Is it possible to remove any part of the assumptions in the theorem, e.g. $d_M(\theta_0^{||},\theta_*^{||})$ sufficiently small or dynamics-related assumptions?
3. Why Theorem 4.2 need to assume the $d_M(\theta_0^{||},\theta_*^{||})$ is sufficiently small? The final convergence point is some sub-optimal solution with larger sharpness.
4. A minor question for the title: why stress overparameterization least squares? I am not sure where “overparameterization” is technically mentioned in the paper, though it might be a necessary condition for the assumptions to hold.

**Ethical Concerns:**

["NO or VERY MINOR ethics concerns only"]

**Final Justification:**

The authors settled most of my concerns in the rebuttal. Specifically, the authors managed to lift the strong assumptions (training dynamics related) in the main theorems. Though the scope is limited, I believe this paper makes its contribution in rigorous analysis within the EoS regime. Now I lean slightly toward acceptance if (1) Theorem 4.2 can be restated as the rebuttal claimed (2) the third-order term assumptions are lifted. I appreciate the authors' effort after submission, and I have increased the score to 4.

**Limitations:**

See weakness.

**Quality:**

3

**Strengths And Weaknesses:**

Strength:

- This paper generalizes some toy models in the previous works with generic assumptions on the solution manifold. It offers a more generalized local convergence view of deep scalar multiplication. The paper also unifies the different regimes of EoS by characterizing the sharpness of the flattest minima. The technical contribution is solid.

Weakness:
- The scope of the results are a bit limited. The Assumptions are generalized, but still very strong. For example, it cannot go beyond scalar-valued least squares.
- Theorem 4.2-4.4 are mostly local convergence with the conditions that $d_M(\theta_0^{||},\theta_*^{||})$ is sufficiently small, which weakens the result. Theorem 4.3 and 4.4 even additionally assume the dynamics have sufficiently small higher-order terms, which is a very strong assumption that needs to hold along the training trajectories.

---

> ### Author Rebuttal · Authors · 2025-07-30
>
> Thank you for your attention to our paper. In summary, it seems your main critiques regard our strong assumptions and local analysis. We have been able to weaken some of the assumptions. Moreover, the analysis for Theorem 4.2 is local only in a weak sense which holds also for state of the art results in Riemannian optimization generally, while for Theorems 4.3 and 4.4 we can demonstrate empirically that the locality is necessary. See below for details.
>
> **Weaknesses**
>
> > The scope of the results are a bit limited. The Assumptions are generalized, but still very strong. For example, it cannot go beyond scalar-valued least squares.
>
> We agree that our assumptions are strong. However, we request that the current state of knowledge be kept in mind. Edge of stability was only empirically identified in 2021, and theoretical understanding of it is still extremely limited. The phenomenon cannot be explained by classical optimisation theory and its theoretical explanation even in toy examples such as scalar nets is very much an open and active area of research [7,8]. Our work is a significant step forward compared with existing literature since it is the only one that can prove quantitative convergence theorems in the unstable regimes considered.
>
> > Theorem 4.2-4.4 are mostly local convergence with the conditions that $d(\theta_0^{\parallel},\theta_*^{\parallel})$ is sufficiently small, which weakens the result. Theorem 4.3 and 4.4 even additionally assume the dynamics have sufficiently small higher-order terms, which is a very strong assumption that needs to hold along the training trajectories
>
> (a) Concerning the locality of the analysis: please see the answer to your question below.
>
> (b) Concerning the "third order terms" assumption, your critique is entirely valid and we completely agree that this is too strong. Since submitting, we have discovered the assumption can be removed entirely using a "normal form" from bifurcation theory (Theorem 3.4, [6]), whose hypotheses are all satisfied by the examples we consider. Using this result, all our theorems continue to hold and retain identical quantitative rates.
>
> **Questions**
>
> > Is it possible to remove any part of the assumptions in the theorem, e.g. $d(\theta_0^{\parallel},\theta_*^{\parallel})$ sufficiently small or dynamics-related assumptions? Why Theorem 4.2 need to assume the $d(\theta_0^{\parallel},\theta_*^{\parallel})$ is sufficiently small? The final convergence point is some sub-optimal solution with larger sharpness.
>
> These are great questions.
>
> (a) Analysis of Riemannian gradient descent on a geodesically strongly convex objective on a curved manifold generally requires that the initial distance to the objective is bounded by some finite constant $D$; the constants appearing in the convergence analysis then depend on $D$ and grow arbitrarily large as $D$ grows large due to the curvature [2,3,4]. In this sense all Riemannian gradient descent convergence theorems require an "initial distance to optimum is sufficiently small" (i.e. $\leq D$) assumption, although $D$ can be taken as large as one likes (for the price of worsening convergence rate) provided that the objective remains geodesically strongly convex and the neighbourhood remains uniquely geodesic. Our own analysis, since it boils down to Riemannian gradient descent in the parallel direction, must also make this assumption. For Theorem 4.2 specifically, this is the *only* assumption required on $d_0:=d(\theta_0^{\parallel},\theta_*^{\parallel})$, so "sufficiently small" in Theorem 4.2 means $O(1)$ and can be taken as large as any finite-diameter, uniquely geodesic neighbourhood on which the strong convexity of $\lambda$ holds. Thus Theorem 4.2 is no more local than the state of the art results in Riemannian optimisation generally. We will make this point clearer in the next version.
>
> (b) Compared with Theorem 4.2, the assumptions on $d_0$ must be strengthened in Theorems 4.3 and 4.4 due to the fact that we *in addition* need to control the $\theta^{\perp}$-dynamics at each point relative to the limiting $\theta^{\perp}$-dynamics (which, unlike in Theorem 4.2, do not convergence exponentially to zero).  This requires $d_0$ to be small relative to these limiting $\theta^{\perp}$-dynamics (e.g. $d_0 = O((\eta\lambda(\theta_*^{\parallel})-2)^{3/2})$ in Theorem 4.4). Moreover, we can demonstrate empirically that this locality is *necessary*: even in the simplest case of a two layer scalar linear network, having learning rate $\eta\geq 2/\lambda(\theta_*^{\parallel})$ means that if $d_0$ is sufficiently large, so that $\eta>2/\lambda(\theta_0^{\parallel})$ is sufficiently large, optimisation diverges for any $\|\theta^{\perp}_0\|\neq 0$. Thus while it may be possible to quantitatively weaken the $d_0$ assumption in Theorems 4.3 and 4.4, it is not possible to remove the assumption entirely. We are happy to provide code for this experiment if requested.
>
> (c) We can remove the "third order terms sufficiently small" assumption as detailed above without changing the convergence theorems or the validity of our examples.
>
> (d) We can relax Assumption 3.1 to the assumption that at every point on the line orthogonal to $M$ at $\theta_*^{\parallel}$, the loss gradient is orthogonal to $M$ at $\theta_*^{\parallel}$. We can prove that this condition is *necessary* for Theorem 4.4 (the proof is easy: if gradient descent oscillates between two points on the line orthogonal to $M$ at $\theta_*^{\parallel}$, then the gradient itself, being parallel to this line, must also be orthogonal to $M$ at $\theta_*^{\parallel}$). The examples we cite in the paper all satisfy this relaxed assumption. We can also empirically demonstrate the necessity of this relaxed assumption (see the paragraph that follows).
>
> > Is there any more general case satisfying the assumptions?
>
> In place of ReLU on Line 186 (multilayer scalar factorisation of depth $L$) one may take any nonlinearity which is the identity on a neighbourhood of $y^{1/L}$, where $y>0$ is the target value. A nice *counterexample* is the 2 layer nonlinear scalar network $f(\theta_1,\theta_2) = \theta_2\tanh(\theta_1)$ when the target value $y>0$; this example does *not* satisfy the weakened assumption in (d) above (but satisfies all the rest of our assumptions), and it can be seen by numerically plotting the trajectory that, in the supercritical regime, convergence to a stable period-2 orbit centred on $\theta_*^{\parallel}$ *does not hold*; convergence is instead to a stable period-2 orbit centred on a *different* point, of *suboptimal* sharpness. This is to be contrasted with the linear and ReLU case, in which convergence to a stable period-2 orbit about the point of *optimal sharpness* $\theta_*^{\parallel}$ *does* occur, as predicted by our theory. We will cite this $\tanh$ counterexample as an additional limitation of the theory, however this counterexample is itself of interest: it suggests that the flatness-seeking implicit bias of edge of stability dynamics is *mediated* by the choice of nonlinearity in such a way that it can hold for ReLUs but not for other activations (such as $\tanh$). To our knowledge, this is itself a new observation that will be of great interest to the community studying the optimization of deep nets. We are happy to supply code to run this experiment if required.
>
> > A minor question for the title: why stress overparameterization least squares?
>
> This is a point worth clarifying. For a model $f$, overparametrisation is often used in the optimisation literature to guarantee that the neural tangent kernel $df df^T$ is positive definite [5], which is the same as requiring that $df$ is full rank. We insist on precisely this full-rank property at lines 132-134 by asking that $y$ be a regular value. Thus, we regard this assumption as an overparametrisation assumption. Without it, the solutions need not form a smooth manifold and our analysis need not apply. We will attempt to make this clearer in the next version.
>
> Please let us know if you would like any further clarifications.
>
> 1. A Convergence Analysis of Gradient Descent for Deep Linear Neural Networks, Arora et al, ICLR 2019.
> 2. First order methods for geodesically convex optimization, Zhang et al, COLT, 2016.
> 3. Convergence and Trade-Offs in Riemannian Gradient Descent and Riemannian Proximal Point, Rubio et al, ICML2024
> 4. On the Convergence of Gradient Descent for Finding the Riemannian Center of Mass, Afsari et al, SIAM Journal on Control and Optimization, Vol. 51,  Issue 3, 2013.
> 5. Memorization and Optimization in Deep Neural Networks with Minimum Over-parameterization, Bombari et al, NeurIPS2022.
> 6. Normal Forms, Differentiable Conjugacies, and Elementary Bifurcations of Maps, Glendinning et al, SIAM Journal on Applied Mathematics, Volume 83, Issue 2, 2023.
> 7. Gradient Descent Monotonically Decreases the Sharpness of Gradient Flow Solutions in Scalar Networks and Beyond, Kreisler et al, ICML 2023.
> 8. Beyond the Edge of Stability via Two-step Gradient Updates, Chen et al, ICML 2023.

---

> ### Comment · Reviewer_CvCh · 2025-08-05
>
> The authors settled most of my concerns in the rebuttal. Specifically, the authors managed to lift the strong assumptions (training dynamics related) in the main theorems. Though the scope is limited, I believe this paper makes its contribution in rigorous analysis within the EoS regime. Now I lean slightly toward acceptance if (1) Theorem 4.2 can be restated as the rebuttal claimed (2) the third-order term assumptions are lifted in the later versions. I appreciate the authors' effort after submission, and I have increased the score to 4.

---

> > ### Author Response · Authors · 2025-08-05
> >
> > Thank you for your continued engagement. We are happy to make the changes you have specified for the next version.

---

### Official Review · Reviewer_SHZE · 2025-07-06

**Clarity:** 3
**Significance:** 3
**Originality:** 3
**Rating:** 4
**Confidence:** 3

**Summary:**

The paper investigates the edge of stability behavior of large step gradient descent (GD) in a codimension one overparameterized least squares problem (with a single input data point). As the authors argue, the setting strikes a balance between being overly minimal and overly general, allowing for clear insights into the dynamics. Moreover, unlike several prior works that focus on logistic regression, which show that GD iterates eventually enter a stable regime where the loss decreases monotonically, this work presents a case that the iterates never reach such a stable regime. This observation could be of independent interest.

Under this paper's setting, the set of global minimizers forms a Riemannian manifold, enabling a decomposition of the GD dynamics. The authors show that the sharpness $\lambda$ is geodesically strongly convex along the manifold for two instances. Assuming that the manifold is symmetric with respect to a reference point $\theta_*^{\|\|}$, they analyze the convergence behavior of GD when the iterates remain within a neighborhood of $\theta_*^{\|\|}$. In particular, the authors identify three regimes exhibiting distinct behaviors depending on whether the step size is smaller than, equal to, or larger than $2/\lambda(\theta_*^{\|\|})$.

**Questions:**

- Lines 39-40: Unlike [31,30,6], which shows that large step GD eventually enters a regime where the logistic loss decreases monotonically, the paper [21] focuses on the least squares loss and does not appear to provide a similar analysis. Could you clarify this distinction?
- Line 51: I suggest clarifying that the GD update approximations in equation (2) will be formally derived later.
- Line 71: Could you cite relevant papers to support this point?
- Lines 112-113: Where was the oscillatory implicit bias observed? How does it align with the technical findings in this paper?
- Line 147: I recommend rephrasing to make it clear that this is an assumption, later shown to be satisfied by several examples. As written, the statement may initially make the assumption seem artificial, even though justifications follow soon after.
- Lines 183-184: Could you clarify why this is the case?
- Line 220: I suggest stating Assumption 3.1 explicitly here for clarity.
- Line 239: Why is the sharpness $\lambda$ characterized through the geodesic strong convexity of itself?
- Line 245-246: It would be helpful to briefly summarize the main message of prior works involving bifurcation diagrams here.
- Line 248: I do not fully follow why the observations above motivate the consideration of the nominal dynamical system described in equations (15-16). A clearer explanation would be appreciated.
- Line 281: $\lambda(\theta_0^{\|\|}) > \lambda(\theta_*^{\|\|})$ appears to implicitly assumed here. It would be helpful if the authors could make this explicit and provide some explanation or justification. Additionally, is this assumption also made in the analysis of the other two regimes?
- Line 299: This line seems to treat $\theta_*^{\|\|}$ as the optimally flat optimal minimizer. If that is indeed the case, it would be beneficial to elaborate on this interpretation earlier in the paper.

**Ethical Concerns:**

["NO or VERY MINOR ethics concerns only"]

**Final Justification:**

Many of my questions were requests for clarification, and the authors responded well, which helped me better position this paper.
While the local nature of the results was initially a concern, I now understand that only the initialization needs to be close to the reference point—not all iterates—which makes the assumption less restrictive.
Other reviewers also raised concerns about this local restriction, but like them, I do not see it as a critical limitation. Overall, I believe this paper may offer valuable insight into understanding the EoS phenomenon

**Limitations:**

The paper clearly list several limitations at the end. However, I believe the assumption that the iterates remain near the reference point $\theta_*^{\|\|}$ constitutes an additional non-negligible limitation that deserves explicit discussion.

**Paper Formatting Concerns:**

.

**Quality:**

3

**Strengths And Weaknesses:**

Strengths
- The decomposition of GD dynamics into components tangential and orthogonal to the manifold, as presented in Theorem 4.1, and their characterization in terms of the sharpness $\lambda$ is novel and insightful. (While I did not have sufficient time or expertise to fully verify the proof of this theorem, the proof sketch appears sound.)
- Demonstrating that Assumptions 3.1 and 3.2, which initially seemed quite strong, hold in settings such as multilayer scalar factorization and scalar factorization by two vectors is interesting. This helps illuminate the structure of the manifold and the property of the sharpness $\lambda$ along it.

Weaknesses
- Although the study of edge of stability in least squares problems is valuable, the paper does not clearly motivate why it is important to extend this analysis beyond the logistic loss setting.
- The main conclusion relies on the assumption that the iterates remain within a neighborhood of the reference point  $\theta_*^{\|\|}$, which may limit the generalizability of the results to broader regions of the manifold.
- This paper is technically dense, which makes it challenging to extract the main message. A clearer high-level overview of intuitive summary (especially, of Sections 4.2-4.4) would improve accessibility for a broader audience,

---

> ### Author Rebuttal · Authors · 2025-07-30
>
> Thank you for your detailed reading and critique. In summary, it seems your primary critiques concern lack of clarity and an assumption that the iterates remain near the fixed point. We have attempted to clarify the points you raised, as well as clarify that we only assume the *initialisation* is near the fixed point, and explain why this assumption is necessary. See below for details.
>
> **Weaknesses**
>
> > the paper does not clearly motivate why it is important to extend this analysis beyond the logistic loss setting
>
> The square loss is frequently used in applications to scene synthesis [1] and has been shown to be equally performant to logistic losses for classification [2]. Whether one uses the square loss or logistic loss, one observes edge of stability convergence with large learning rates [3]. The logistic loss setting is just one side of a more general set of phenomena which are still poorly understood; our work is a step towards a more complete understanding.
>
> > The main conclusion relies on the assumption that the iterates remain within a neighborhood of the reference point
>
> We only assume that the *initial* iterate is in a neighborhood of the reference point; our theorems then *prove* that all further iterates remain in the neighborhood (see Lines 261 through 272 where this is identified explicitly). Moreover:
>
> (a) Concerning assumptions on $d_0:=d(\theta_0^{\parallel},\theta_*^{\parallel})$: analysis of Riemannian gradient descent on a geodesically strongly convex objective on a curved manifold generally requires that the initial distance to the objective is bounded by some finite constant $D$; the constants appearing in the convergence analysis then depend on $D$ and grow arbitrarily large as $D$ grows large due to the curvature [4]. In this sense all Riemannian gradient descent convergence theorems require an "initial distance to optimum is sufficiently small" (i.e. $\leq D$) assumption, although $D$ can be taken as large as one likes (for the price of worsening convergence rate) provided that the objective remains geodesically strongly convex and the neighborhood remains uniquely geodesic. Our own analysis, since it boils down to Riemannian gradient descent in the parallel direction, must also make this assumption. For Theorem 4.2 specifically, this is the *only* assumption required on $d_0:=d(\theta_0^{\parallel},\theta_*^{\parallel})$, so "sufficiently small" in Theorem 4.2 means $O(1)$ and can be taken as large as any finite-diameter, uniquely geodesic neighborhood on which the strong convexity of $\lambda$ holds. Thus Theorem 4.2 is no more local than state of the art results in Riemannian optimisation generally. We will make this point clearer in the next version.
>
> (b) Compared with Theorem 4.2, the assumptions on $d_0$ must be strengthened in Theorems 4.3 and 4.4 due to the fact that we *in addition* need to control the $\theta^{\perp}$-dynamics at each point relative to the limiting $\theta^{\perp}$-dynamics (which, unlike in Theorem 4.2, do not convergence exponentially to zero).  This requires $d_0$ to be small relative to these limiting $\theta^{\perp}$-dynamics (e.g. $d_0 = O((\eta\lambda(\theta_*^{\parallel})-2)^{3/2})$ in Theorem 4.4). Moreover, we can demonstrate empirically that this locality is *necessary*: even in the simplest case of a two layer scalar linear network, having learning rate $\eta\geq 2/\lambda(\theta_*^{\parallel})$ means that if $d_0$ is sufficiently large, so that $\eta>2/\lambda(\theta_0^{\parallel})$ is sufficiently large, optimisation diverges for any $\|\theta^{\perp}_0\|\neq 0$. We are happy to provide code for this experiment if requested.
>
> > This paper is technically dense...
>
> This is a fair critique. We will include a higher-level overview of these sections at the beginning of Section 4 in the next version.
>
> **Questions**
>
> We are happy to make the changes you have suggested. Concerning your questions:
>
> > Lines 39-40...Could you clarify this distinction?
>
> The logistic loss in the papers [30,31,6] has sharpness tending to zero as one approaches infinity in parameter space in the direction of perfect classification. The analysis in these papers demonstrates an implicit bias in this direction implying that eventually, *irrespective of how large the learning rate $\eta$ is*, the algorithm will find a point at which the sharpness is $<2/\eta$ (stable regime). For the square loss considered in [21] the sharpness is nonvanishing, having minimal value $\lambda_*>0$, and the learning rate $\eta$ *must be taken small enough* that $\eta<2/\lambda_*$ to ensure eventual stability. The setting in [21] thus corresponds to the subcritical regime in our paper (Theorem 4.2). We also consider the square loss in our paper, but consider regimes (the critical and supercritical regime) in which $\eta\geq 2/\lambda_*$, which are not covered by [21].
>
>
> > Line 71: Could you cite relevant papers to support this point?
>
> Yes: the subcritical regime (our Theorem 4.2) has been observed empirically and analysed theoretically in [5]. Their convergence theorem is similar to ours, but they work with a specific (width 2, scalar output) model rather than a *class* of models as we do. The supercritical regime (our Theorem 4.4) has been observed empirically in [6] for 2 layer linear neural nets and in [7] for deep scalar linear neural nets, but neither work supplies a convergence theorem comparable to ours. The critical regime (our Theorem 4.3) has not been identified or analysed in any other works to our knowledge.
>
> > Lines 112-113: Where was the oscillatory implicit bias observed?
>
> The oscillatory implicit bias has been observed in [6], where it is shown that the sharpness itself oscillates during training and exhibits a bifurcation diagram as the learning rate is increased. Our theory demonstrates theoretically that these oscillations are a consequence of the quadratic dynamical system on $\theta^{\perp}$ (Eq. (12)); such systems are well-known to exhibit such oscillations and bifurcation diagrams [9]. Our theory covers the first "fork" in the bifurcation diagram. We will attempt to make this point clearer in the final version.
>
> > Lines 183-184: Could you clarify why this is the case?
>
> The reason is that ReLUs do not change the loss landscape in a neighborhood of the solution manifold. This is most easily seen in the depth 2 case. The solutions of $\theta_2\theta_1 = 1$ form a hyperbola admitting two connected components: one in the $\theta_1,\theta_2>0$ quadrant and one in the $\theta_1,\theta_2<0$ quadrant. The solutions of $\theta_2\phi(\theta_1) = 1$ form *exactly the same* hyperbola component in the $\theta_1,\theta_2>0$ quadrant, with no solutions occurring in the $\theta_1,\theta_2<0$ quadrant. Since we restrict our analysis to only the component in the positive quadrant, the solution manifold is the same for our analysis in each case.
>
> > Line 239: Why is the sharpness $\lambda$ characterized through the geodesic strong convexity of itself?
>
> We need to word this more clearly. What we intend to say is that the geodesic strong convexity can be used to control the sharpness of the solution to which one converges: in the subcritical regime, the $\theta^{\parallel}$ dynamics are Riemannian gradient descent on $\lambda$ with exponentially decreasing stepsize, so they converge to a suboptimally flat minimum. In the critical and supercritical regimes, the $\theta^{\parallel}$ dynamics are Riemannian gradient descent on $\lambda$ with polynomially decreasing and constant step size respectively, which, since $\lambda$ is geodesically strongly convex, is sufficient to guarantee convergence to the minimiser of $\lambda$.
>
> > Line 248: I do not fully follow...
>
> The nominal systems are motivated by looking at Equations (11) and (12). Equation (11) looks like gradient descent on $\lambda$ with time-varying step size, but since it is supposed to occur along the solution manifold $M$, we are motivated to compare it with Riemannian gradient descent on $\lambda$, which is the natural generalisation of gradient descent to manifolds. Equation (12) is a quadratic dynamical system but with time-varying coefficients; the nominal system just fixes the coefficients to make the system easier to study. We will attempt to be clearer in the final version.
>
> > Line 281: $\lambda(\theta_0^{\parallel})>\lambda(\theta_*^{\parallel})$ appears to implicitly assumed here
>
> Yes, $\lambda(\theta_0^{\parallel})>\lambda(\theta_*^{\parallel})$ is implicitly assumed throughout because (as you have identified) $\theta_*^{\parallel}$ is indeed the optimally flat optimal minimzer. We have made this explicit at Line 164, however we will attempt to make this clearer in the next version.
>
> Please let us know if you would like any further clarification.
>
> 1. NeRF: Representing Scenes as Neural Radiance Fields for View Synthesis, Mildenhall et al, ECCV2020.
> 2. Evaluation of Neural Architectures Trained with Square Loss vs Cross-Entropy in Classification Tasks, Hui et al, ICLR2021.
> 3. Gradient Descent on Neural Networks Typically Occurs at the Edge of Stability, Cohen et al, ICLR2021
> 4. Convergence and Trade-Offs in Riemannian Gradient Descent and Riemannian Proximal Point, Rubio et al, ICML2024
> 5. A Minimalist Example of Edge-of-Stability and Progressive Sharpening, Liu et al, arXiv, 2025.
> 6. Universal Sharpness Dynamics in Neural Network Training: Fixed Point Analysis, Edge of Stability, and Route to Chaos, Kalra et al, ICLR2025.
> 7. Gradient Descent Monotonically Decreases the Sharpness of Gradient Flow Solutions in Scalar Networks and Beyond, Kreisler et al, ICML2023.
> 8. Nonlinear Dynamics and Chaos: With applications to Physics, Biology, Chemistry and Engineering, Steven Strogatz, CRC Press, 2015.

---

### Official Review · Reviewer_Qn8i · 2025-07-06

**Clarity:** 3
**Significance:** 3
**Originality:** 3
**Rating:** 5
**Confidence:** 4

**Summary:**

This paper studies the convergence of gradient descent in the "edge-of-stability," or large step size, regime for a single data point and the square loss. The authors consider three learning rate regimes -- the subcritical regime, where $\eta$ is small and the model converges to a minimizer with suboptimal sharpness; the supercritical regime, where $\eta$ is large and the model converges linearly to the flattest minimizer, and a critical regime. The proof relies on the observation that GD on the loss projected onto the tangent space approximates reimannian GD on the sharpness, along with an assumption that the sharpness is strongly convex.

**Questions:**

- Theorem 4.4 further assumes that $\eta$ is only very slightly larger than $2/\lambda(\theta_*^\parallel)$. This also seems like a potentially restrictive assumption. Can you comment on what happens for larger $\eta$?
- What is meant by "Given a tangent vector $v \in T_{\theta}\mathbb{R}^p$, let us also denote by $v^\parallel$ and $v^\perp$ the tangent and orthogonal components of $v$, respectively" in line 217? Isn't $v$ already a tangent vector?
- There appears to be a typo in Theorem 4.2 (line 290) for the conditions on $\eta$.

**Ethical Concerns:**

["NO or VERY MINOR ethics concerns only"]

**Final Justification:**

I believe that this paper makes valuable contributions towards understanding the dynamics of gradient descent at the edge of stability. The authors responded to my questions on the local nature of the analysis, and will provide more clarification on the specific conditions required in the next revision. I thus recommend acceptance for this paper.

**Limitations:**

Yes

**Quality:**

3

**Strengths And Weaknesses:**

## Strengths
- It is of much interest in the community to better understand the edge-of-stability phenomenon. Prior works with convergence guarantees often rely on specific models/assumptions. This paper presents a very generic analysis framework for proving convergence to the flattest minimizer, which is to my knowledge novel.
- The introduction of strong convexity of sharpness as an assumption to prove convergence, and observation that common toy models have such strong convexity, is also interesting.
- The paper is well written and easy to understand. I have skimmed the proofs and they appear to be sound.

## Weaknesses

- The main limitation of this paper is that the analysis is "local," i.e the main theorems require $\theta_0$ to be sufficiently close to the manifold of minimizers, and in the case of Theorem 4.4, $\theta_0^\parallel$ to also be sufficiently close to the flattest minimizer $\theta_*^\parallel$. This is a similar style of analysis to prior works on label noise SGD [1,2,3], with the difference being that the oscillations from the edge of stability, rather than label noise, drive down the sharpness. As such, I find the claims that the paper analyzes the non-monotonic decrease of GD to be slightly overstated, as the theorems can only handle the case where the loss is already small. I also recommend the authors to add discussion on these prior works.

- As the authors indeed mention, the analysis is limited to one training example (codimension 1)

[1] Implicit regularization for deep neural networks driven by an Ornstein-Uhlenbeck like process. Guy Blanc, Neha Gupta, Gregory Valiant, Paul Valiant. COLT 2020.
[2] Label noise SGD provably prefers flat global minimizers. Alex Damian, Tengyu Ma, Jason D. Lee. NeurIPS 2021.
[3] What happens after SGD reaches zero loss? –a mathematical framework. Zhiyuan Li, Tianhao Wang, Sanjeev Arora. ICLR 2022.

---

> ### Author Rebuttal · Authors · 2025-07-30
>
> Thank you for your considered reading and critique of our work. In summary it seems that your primary critique is about the local nature of the analysis. Our analysis for Theorem 4.2 is local only in a weak sense that is also true for state of the art results in Riemannian optimization more generally; moreover, we can demonstrate empirically that a much stronger kind of locality is *necessary* for convergence in Theorems 4.3 and 4.4. More details below.
>
> **Weaknesses**
>
> We are happy to include discussion of the works you mentioned in the next version.
>
> > The main limitation of this paper is that the analysis is "local," i.e the main theorems require $\theta_0$ to be sufficiently close to the manifold of minimizers and in the case of Theorem 4.4, $\theta_0^{\parallel}$ to also be sufficiently close to the flattest minimizer
>
> (a) Concerning closeness of $\theta_0$ to the manifold of minimizers: this is indeed a limitation in the broader context of optimization, but in the context solely of edge of stability (EOS) dynamics (which is our concern in this paper) we do not believe this to be a limitation since EOS dynamics typically don't begin until the loss has already decreased substantially from initialisation [1].
>
> (b) Concerning assumptions on $d_0:=d(\theta_0^{\parallel},\theta_*^{\parallel})$: analysis of Riemannian gradient descent on a geodesically strongly convex objective on a curved manifold generally requires that the initial distance to the objective is bounded by some finite constant $D$; the constants appearing in the convergence analysis then depend on $D$ and grow arbitrarily large as $D$ grows large due to the curvature [2,3,4]. In this sense all Riemannian gradient descent convergence theorems require an "initial distance to optimum is sufficiently small" (i.e. $\leq D$) assumption, although $D$ can be taken as large as one likes (for the price of worsening convergence rate) provided that the objective remains geodesically strongly convex and the neighbourhood remains uniquely geodesic. Our own analysis, since it boils down to Riemannian gradient descent in the parallel direction, must also make this assumption. For Theorem 4.2 specifically, this is the *only* assumption required on $d_0:=d(\theta_0^{\parallel},\theta_*^{\parallel})$, so "sufficiently small" in Theorem 4.2 means $O(1)$ and can be taken as large as any finite-diameter, uniquely geodesic neighbourhood on which the strong convexity of $\lambda$ holds. Thus Theorem 4.2 is no more local than state of the art results in Riemannian optimisation generally. We will make this point clearer in the next version.
>
> (c) Compared with Theorem 4.2, the assumptions on $d_0$ must be strengthened in Theorems 4.3 and 4.4 due to the fact that we *in addition* need to control the $\theta^{\perp}$-dynamics at each point relative to the limiting $\theta^{\perp}$-dynamics (which, unlike in Theorem 4.2, do not convergence exponentially to zero).  This requires $d_0$ to be small relative to these limiting $\theta^{\perp}$-dynamics (e.g. $d_0 = O((\eta\lambda(\theta_*^{\parallel})-2)^{3/2})$ in Theorem 4.4). Moreover, we can demonstrate empirically that this locality is *necessary*: even in the simplest case of a two layer scalar linear network, having learning rate $\eta\geq 2/\lambda(\theta_*^{\parallel})$ means that if $d_0$ is sufficiently large, so that $\eta>2/\lambda(\theta_0^{\parallel})$ is sufficiently large, optimisation diverges for any $\|\theta^{\perp}_0\|\neq 0$. Thus while it may be possible to quantitatively weaken the $d_0$ assumption in Theorems 4.3 and 4.4, it is not possible to remove the assumption entirely. We are happy to provide code for this experiment if requested.
>
>
> **Questions**
>
> >Theorem 4.4 further assumes that $\eta$ is only very slightly larger than $2/\lambda(\theta_*^{\parallel})$. This also seems like a potentially restrictive assumption. Can you comment on what happens for larger $\eta$?
>
> Great question. For larger $\eta$, the periodicity of the orbit to which the dynamics converge increase in a "period-doubling" fashion (i.e., to an orbit of period 4, then to an orbit of period 8, and so on) before eventually becoming chaotic (see Lines 324-330). This "period doubling" phenomenon has already been observed empirically for edge of stability dynamics in prior work [5], and is well-known to occur for perturbed quadratic systems of the type we consider ([6], Section 10.3). We did not include analysis of this case because it would have required a much lengthier analysis of the quadratic dynamical system than what we have supplied, and it seemed to us that the analysis would have to be performed on a case-by-case basis (a separate theorem for each period, with determining the $2^n$th-periodic oscillation requiring the solution of a $2^n$th degree polynomial followed by a perturbation analysis). With the paper already being quite long and technical, we felt it best to leave investigation of this question to future work.
>
> >What is meant by "Given a tangent vector $v\in T_{\theta}\mathbb{R}^p$, let us also denote by $v^{\parallel}$ and $v^{\perp}$ the tangent and orthogonal components of $v$, respectively" in line 217? Isn't already a tangent vector?
>
> Thanks for picking up on this, we should have been clearer. $v\in T_{\theta}\mathbb{R}^p$ is indeed a tangent vector *to Euclidean space*, but $v^{\parallel}$ is its projection along *the tangent space to $M$*, while $v^{\perp}$ is its projection *orthogonal* to $M$. We will clarify this in the final version.
>
> >There appears to be a typo in Theorem 4.2 (line 290) for the conditions on $\eta$.
>
> Thank you for spotting this, you are correct: the condition should read $1/\lambda(\theta_*^{\parallel})<\eta<2/\lambda(\theta_*^{\parallel})$ rather than $1/\lambda(\theta_*^{\parallel})<\eta<2/\lambda(\theta_0^{\parallel})$.
>
> Please let us know if you would like any further clarifications.
>
> 1. Gradient Descent on Neural Networks Typically Occurs at the Edge of Stability, Cohen et al, ICLR2021
> 2. First order methods for geodesically convex optimization, Zhang et al, COLT, 2016
> 3. Convergence and Trade-Offs in Riemannian Gradient Descent and Riemannian Proximal Point, Rubio et al, ICML2024
> 4. On the Convergence of Gradient Descent for Finding the Riemannian Center of Mass, Afsari et al, SIAM Journal on Control and Optimization, Vol. 51,  Issue 3, 2013
> 5. Universal Sharpness Dynamics in Neural Network Training: Fixed Point Analysis, Edge of Stability, and Route to Chaos, Kalra et al, ICLR2025
> 6. Nonlinear Dynamics and Chaos: With applications to Physics, Biology, Chemistry and Engineering, Steven Strogatz, CRC Press, 2015
> 7. Normal Forms, Differentiable Conjugacies, and Elementary Bifurcations of Maps, Glendinning et al, SIAM Journal on Applied Mathematics, Volume 83, Issue 2, 2023

---

> > ### Comment · Reviewer_Qn8i · 2025-08-03
> >
> > Thank you to the authors for your response to my questions.
> >
> > I think that the clarity of the paper could benefit by explicitly specifying how "sufficiently small" various quantities, such as $\||\theta\_0^\perp\||$, need to be in order for the proof to go through. As written, it isn't clear whether these can be O(1) quantities.
> >
> > Re point (a): It is indeed possible (and observed in practice) that EOS behavior can arise when the loss is $\Theta(1)$. In fact, if one chooses $\eta$ to be slightly larger than 2/sharpness at initialization, then EOS behavior will be observed immediately at the start of training. And so I do still think that requiring $\theta_0$ to be sufficiently close to the manifold of minimizers is a limitation.
> >
> > Nevertheless, I think that these are minor points, and I would like to maintain my accept recommendation for this paper.

---

> > > ### Author Response · Authors · 2025-08-05
> > >
> > > Thank you for your continued engagement and your appreciation of our paper. We will make sure to clarify what "sufficiently small" means for each of the theorems in the next version.

---

### Decision · Program_Chairs · 2025-09-17

**Decision:**

Accept (poster)

**Comment:**

The paper investigates the edge-of-stability phenomenon of gradient descent under large stepsize. By focusing on the overparametrized least square problem, the paper provides a detailed analysis of the phenomenon. Based on the local characterization of the solution set which is a Riemannian manifold, the authors manage to decompose the gradient descent dynamics, which further allows to  identify three regimes exhibiting distinct behaviors depending on  the stepsize: subcritical, critical, and supercritical.


**Strength:**
 - The paper provides an insightful perspective to understand the edge-of-stability phenomenon.
 - The paper is well presented.

**Weakness:**
 - The analysis is only for "local", and limited on one training example.
 - Relative strong assumptions, which limits the applicability of the result.

**Reason of decision:**
The topic of the paper is important and has broad interests, the obtained result delivers an interesting and insightful perspective to understand the edge-of-stability phenomenon.

**Discussion summary:** The authors are suggested to adopt the reviewers' suggestions to revise the final version of the paper, not limited to the following points
 - Better justification and discussion on the result only for "local" analysis.
 - Re-organization of the paper for better presentation.
 - Modification of Theorem 4.2, and weaken assumption by lifting third-order term assumption.